# THE DIFFERENCES BETWEEN DIRECT ALIGNMENT ALGORITHMS ARE A BLUR

## ABSTRACT

Direct Alignment Algorithms (DAAs) simplify LLM alignment by directly optimizing policies, bypassing reward modeling and RL. While DAAs differ in their use of SFT (one-stage vs. two-stage) and the scalar score they optimize (likelihood vs. odds ratios), the key performance drivers remain underexplored. We present a systematic comparison and analyze a previously overlooked axis - the ranking objective (pairwise vs. pointwise). To isolate this factor, we propose a unified training framework across DAAs by (i) converting one-stage methods (ORPO, ASFT) into a two-stage pipeline with an explicit SFT phase and (ii) introducing a $\beta$ parameter that places all methods in the same hyperparameter space and improves the quality of odds-ratio DAAs (ORPO, ASFT). Under this setup, the ranking objective emerges as the primary determinant of alignment quality, whereas the particular scalar score (policy–reference ratio vs. odds ratio) is secondary. We corroborate this on instruction-following tasks and further confirm it on math-reasoning benchmarks across model scales. Evidence suggests that this stems from how these objectives interact with prompt-specific biases, supported both by strictly controlled experiments and by observations on real data. Our findings underscore the need for nuanced evaluations in DAA research to avoid oversimplified claims of superiority.

## 1 INTRODUCTION

Direct Preference Optimization (DPO) (Rafailov et al., 2023), rooted in RLHF Ouyang et al. (2022); Stiennon et al. (2020), has led to a proliferation of Direct Alignment Algorithms (DAAs) (Meng et al., 2024; Azar et al., 2024; Chen et al., 2024). These methods differ in design: most adopt DPO's two-stage paradigm, modifying the loss function and retaining a policy, reference ratio and temperature parameter $\beta$ (Xiao et al., 2024; D'Oosterlinck et al., 2025), while others, such as ORPO and ASFT (Hong et al., 2024; Wang et al., 2024), unify alignment and supervised fine-tuning (SFT) in a single stage using an odds-ratio objective without a reference policy. This variety has resulted in a fragmented literature, making it difficult to isolate which design choices actually drive improvements in alignment quality.

In this work, to enable controlled comparison of algorithmic factors, we restrict our analysis to the offline setting with static datasets of binary preferences, avoiding confounding effects from online data collection or more complex feedback structures. We systematically analyze one-stage DAAs and provide a detailed motivation for converting them into a two-stage pipeline with an explicit SFT phase. Crucially, we show that introducing a $\beta$ parameter, typically absent in one-stage odds-ratio methods, serves as an effective tempering mechanism and is essential for unlocking their full performance. By unifying all methods under this protocol, we place single- and two-stage DAAs in a common hyperparameter space and enable controlled comparison. Within this framework, we conduct comprehensive empirical studies on instruction-following and math-reasoning benchmarks using Llama 3 (3B, 8B) and Qwen 2.5 (7B, 14B) models, and systematically examine the data efficiency of DAAs with respect to SFT data volume.

Our main contributions are: (i) We establish a unified training protocol for DAAs, demonstrating that moving the SFT term out of originally one-stage losses into a separate SFT phase and using an alignment-only loss with a proposed $\beta$ parameter is essential for maximizing performance, even for odds-ratio objectives. (ii) Within this unified setting, we find that previously reported advantages

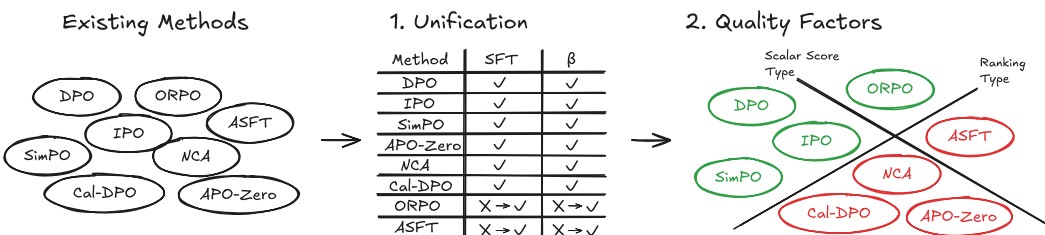

Figure 1: **Overview of our work and main finding. Left:** Existing DAA methods differ in use of SFT and $\beta$. **Center:** Our unified protocol makes SFT and $\beta$ explicit for all, bringing ORPO/ASFT into the same framework. **Right:** We compare DAAs along two axes (*scalar score type* and *ranking type*) and find that **ranking type** (pairwise, green vs. pointwise, red) is the main determinant of alignment quality after unification.

of various DAAs often disappear: after tuning, all methods perform similarly or worse than DPO. Our results indicate that *ranking type* (pairwise vs. pointwise), rather than scalar-score choice or heuristic loss design, is the primary determinant of alignment quality, with both score types yielding comparable results. (iii) We provide evidence that observed performance gaps arise from the interaction between each objective and prompt-specific data biases, explaining why differences among DAAs emerge primarily at intermediate task difficulty; outside this regime, the distinctions between DAAs is a *blur*.

Among our findings, we observe that most methods are highly data-efficient: with 5–10% SFT, models reach $\geq 95\%$ of their full-data score. Our findings challenge claims of algorithmic superiority in the DAA literature and underscore the importance of systematic, controlled evaluation.

## 2 PRELIMINARIES

### 2.1 MODELING SEQUENCES

Given a sequence $y$ of length $|y|$, the log-probability can be written as $\log p(y) = \sum_{i=1}^{|y|} \log p(y_i \mid y_{<i})$, which may also be conditioned on another sequence $x$. In practice, optimizing normalized log-probability $\frac{1}{|y|} \log p(y) = \log\left(p(y)^{\frac{1}{|y|}}\right)$ often improves numerical stability and leads to better training. However, once normalized, the resulting quantity is no longer a strict probability measure. Throughout this paper, whenever we write $p(y)$, we refer to this normalized version $p(y)^{\frac{1}{|y|}}$. Whenever a method does not apply this normalization, we indicate it explicitly.

Welleck et al. (2019) introduced a log-unlikelihood term that reduces the probability of certain undesirable tokens: $\log\left(1-p(c|y_{<i})\right)$ for $c \in \mathcal{C}$. It can be extended to an entire sequence as $\log\left(1-p(y)\right)$.

### 2.2 REINFORCEMENT LEARNING FROM HUMAN FEEDBACK

Reinforcement Learning from Human Feedback (RLHF) (Ouyang et al., 2022; Stiennon et al., 2020) is a prominent approach to aligning language models. It generally has three stages:

- **Supervised Fine-Tuning (SFT).** During the SFT stage, the model $\pi_\theta$ is trained to follow instructions by maximizing the probability of correct output $y$ given input $x$. For a single training pair $(x, y)$, we define the per-sample SFT loss as $\mathcal{L}_{\text{SFT}}(\pi_\theta, x, y) = -\log \pi_\theta(y \mid x)$. During fine-tuning, we minimize the expectation of this per-sample loss over the training dataset $\mathcal{D}$:
$\mathbb{E}_{(x,y) \sim \mathcal{D}}\left[\mathcal{L}_{\text{SFT}}(\pi_\theta, x, y)\right]$.

- **Reward Modeling (RM).** A reward model $r_\psi(x, y)$ produces a satisfaction score. It is trained on preference pairs using the Bradley-Terry model (Bradley & Terry, 1952): $\mathcal{L}_{\text{RM}}(r_\psi) = -\mathbb{E}_{(x,y_w,y_l) \sim \mathcal{D}}\left[\log \sigma\left(r_\psi(x, y_w) - r_\psi(x, y_l)\right)\right]$, where $y_w$ is the preferred response and $y_l$ is the less preferred one.

- **Reward Maximization.** The objective is to generate responses that maximize the learned reward, with a KL penalty to prevent reward hacking: $\max_{\pi_\theta} \mathbb{E}_{x \sim \mathcal{D}, y \sim \pi_\theta(y|x)}\left[r_\phi(x, y)\right]\quad -$

$\beta \, \mathbb{D}_{\mathrm{KL}}\big[\pi_\theta(x, y) \,\|\, \pi_{\mathrm{ref}}(x, y)\big]$. Reinforcement learning (RL) algorithms are commonly used to optimize this objective (Schulman et al., 2017; Ouyang et al., 2022).

## 2.3 DIRECT ALIGNMENT ALGORITHMS

Direct alignment algorithms replace the reward modeling and RL stages (but keep the SFT phase) with a single alignment step. Various preference-optimization loss functions have been proposed, employing these core components:

- $r_\theta^{\mathrm{ref}}(y, x) = \log\big(\frac{\pi_\theta(y|x)}{\pi_{\mathrm{ref}}(y|x)}\big)$ from DPO (Rafailov et al., 2023), which acts as an implicit reward $\beta \, r_\theta^{\mathrm{ref}}$. No length normalization is used.

- $r_\theta^{\mathrm{odds}}(y, x) = \log\big(\frac{\pi_\theta(y|x)}{1-\pi_\theta(y|x)}\big)$ utilized in ORPO (Hong et al., 2024), representing the odds of generating $y$ versus not generating it. While not directly derived from an RL objective in the same way as $r_\theta^{\mathrm{ref}}$, its empirical success in methods like ORPO and ASFT motivates its inclusion in our comparative analysis.

Several Direct Alignment Algorithms use these notations. Information on sequence probability normalization for these methods is presented in Appendix A.1. **Direct Preference Optimization (DPO)** (Rafailov et al., 2023): $\mathcal{L}_{\mathrm{DPO}} = -\log \sigma\big(\beta \, r_\theta^{\mathrm{ref}}(y_w, x) - \beta \, r_\theta^{\mathrm{ref}}(y_l, x)\big)$ (this method does not normalize probabilities by length);[2] **Identity Preference Optimization (IPO)** (Azar et al., 2024): $\mathcal{L}_{\mathrm{IPO}} = \big(r_\theta^{\mathrm{ref}}(y_w, x) - r_\theta^{\mathrm{ref}}(y_l, x) - \frac{1}{2\beta}\big)^2$; **Simple Preference Optimization (SimPO)** (Meng et al., 2024): $\mathcal{L}_{\mathrm{SimPO}} = -\log \sigma\big(\beta \, \log \pi_\theta(y_w, x) - \beta \, \log \pi_\theta(y_l, x) - \gamma\big)$; **Noise Contrastive Alignment (NCA)** (Chen et al., 2024): $\mathcal{L}_{\mathrm{NCA}} = -\log \sigma\big(\beta \, r_\theta^{\mathrm{ref}}(y_w, x)\big) - 0.5 \log \sigma\big(-\beta \, r_\theta^{\mathrm{ref}}(y_w, x)\big) - 0.5 \log \sigma\big(-\beta \, r_\theta^{\mathrm{ref}}(y_l, x)\big)$; **Calibrated Direct Preference Optimization (Cal-DPO)** (Xiao et al., 2024): $\mathcal{L}_{\mathrm{Cal-DPO}} = -\log \sigma\big(r_\theta^{\mathrm{ref}}(y_w, x) - r_\theta^{\mathrm{ref}}(y_l, x)\big) + \big(r_\theta^{\mathrm{ref}}(y_w, x) - \frac{1}{2\beta}\big)^2 + \big(r_\theta^{\mathrm{ref}}(y_l, x) + \frac{1}{2\beta}\big)^2$; **Anchored Preference Optimization Zero (APO-Zero)** (D'Oosterlinck et al., 2025): $\mathcal{L}_{\mathrm{APO-Zero}} = -\sigma\big(\beta \, r_\theta^{\mathrm{ref}}(y_w, x)\big) + \sigma\big(\beta \, r_\theta^{\mathrm{ref}}(y_l, x)\big)$.

## 2.4 ONE-STAGE ALIGNMENT METHODS

One-stage alignment (as a subset of DAA methods) merges SFT and direct alignment in one step by adding their losses: $\mathcal{L}_{\mathrm{Single}}(\pi_\theta) = \mathbb{E}_{(x, y_w, y_l) \sim \mathcal{D}}\big[\mathcal{L}_{\mathrm{SFT}}(\pi_\theta, x, y_w) + \lambda \, \mathcal{L}_{\mathrm{Align}}(\pi_\theta, x, y_w, y_l)\big]$, where $\lambda$ is a hyperparameter, and no reference policy $\pi_{\mathrm{ref}}$ is required.

One-stage methods using odds ratios include:

**Odds Ratio Preference Optimization (ORPO)** (Hong et al., 2024) is defined as: $\mathcal{L}_{\mathrm{ORPO}} = -\log \pi_\theta(y_w|x) - \lambda \underbrace{\log \sigma\big(r_\theta^{\mathrm{odds}}(y_w, x) - r_\theta^{\mathrm{odds}}(y_l, x)\big)}_{-\mathcal{L}_{\mathrm{ORPO}_{\mathrm{Align}}}}.$

**Aligned Supervised Fine-Tuning (ASFT)** (Wang et al., 2024) is defined as: $\mathcal{L}_{\mathrm{ASFT}} = -\log \pi_\theta(y_w|x) - \lambda \Big( \underbrace{\log \sigma\big(r_\theta^{\mathrm{odds}}(y_w, x)\big) + \log \sigma\big(-r_\theta^{\mathrm{odds}}(y_l, x)\big)}_{-\mathcal{L}_{\mathrm{ASFT}_{\mathrm{Align}}}} \Big).$

# 3 METHOD

## 3.1 GENERALIZING ASFT AND ORPO

Our goal in this paper is to characterize the differences among various DAAs. Before proceeding, we summarize the objectives of ASFT and ORPO. These approaches are referred to as *one-stage* methods because they perform alignment immediately after the base model is obtained, in contrast to methods that insert a separate SFT stage before alignment. Consequently, ASFT and ORPO omit the parameter $\beta$; as one-stage methods, the distance to a reference policy is not required. At first

---

[2]Unless otherwise noted, the expectation over $(x, y_w, y_l) \sim \mathcal{D}$ is taken.

glance, it may seem unnecessary to introduce $\beta$ into one-stage methods, yet we will demonstrate that neither the one-stage design nor the absence of $\beta$ is mandatory for ASFT and ORPO.

### 3.1.1 ORPO AND ASFT CAN OPERATE WITHOUT THE SFT LOSS TERM AND AS TWO-STAGE METHODS

First, note that $\mathcal{L}_{\text{ASFT}_{\text{Align}}} = -\log \pi_\theta(y_w|x) - \log\big(1 - \pi_\theta(y_l|x)\big)$, and thus $\mathcal{L}_{\text{ASFT}} = -(1 + \lambda)\log \pi_\theta(y_w|x) - \lambda \log\big(1 - \pi_\theta(y_l|x)\big)$; see Appendix C for a proof. Second, $\mathcal{L}_{\text{ORPO}} = \mathcal{L}_{\text{ASFT}} + \lambda \log\big(\pi_\theta(y_w|x)(1 - \pi_\theta(y_l|x)) + \pi_\theta(y_l|x)(1 - \pi_\theta(y_w|x))\big)$; see Appendix D for details.

From these equations it follows that $\mathcal{L}_{\text{ORPO}} \leq \mathcal{L}_{\text{ASFT}}$ and $\mathcal{L}_{\text{ORPO}_{\text{Align}}} \leq \mathcal{L}_{\text{ASFT}_{\text{Align}}}$ (see Appendix D.2).

These results lead to three observations: (i) $\mathcal{L}_{\text{ASFT}}$ upper-bounds $\mathcal{L}_{\text{ORPO}}$; therefore, minimizing the former automatically minimizes the latter. (ii) $\mathcal{L}_{\text{ASFT}_{\text{Align}}}$ can be regarded as the simplest DAA loss, mirroring the structure of BCE (see Appendix C.3); (iii) Most importantly, the alignment terms of ORPO and ASFT already include the NLL component ($-\log \pi_\theta(y_w|x)$), making the additional $\mathcal{L}_{\text{SFT}}$ term in one-stage formulations potentially redundant. Thus, we hypothesize that removing the explicit $\mathcal{L}_{\text{SFT}}$ term and instead using a separate SFT stage followed by alignment will improve performance, motivating our **RQ1**: *"Does converting ORPO and ASFT to a two-stage pipeline improve alignment quality?"* and experiments in Section 5.1, where we compare ASFT and ORPO both in their original one-stage form and in a two-stage variant that follows an explicit SFT phase.

### 3.1.2 TEMPERING ASFT AND ORPO

We now revisit the original one-stage methods from Section 2.4 and examine how the alignment terms $\mathcal{L}_{\text{ORPO}_{\text{Align}}}$ and $\mathcal{L}_{\text{ASFT}_{\text{Align}}}$ compare. These terms optimize preferences and, depending on the coefficient $\lambda$, can dominate or have a smaller impact on the final loss.

While $\mathcal{L}_{\text{ASFT}_{\text{Align}}}$ and $\mathcal{L}_{\text{ORPO}_{\text{Align}}}$ use $r_\theta^{\text{odds}}$, many DAAs incorporate a scaling parameter $\beta$. To enable a unified comparison and investigate the role of $\beta$, we introduce it to scale $r_\theta^{\text{odds}}$:

$$\mathcal{L}_{\text{ASFT}_{\text{Align}}}^\beta = -\log \sigma(\beta r_\theta^{\text{odds}}(y_w, x)) - \log \sigma(-\beta r_\theta^{\text{odds}}(y_l, x)), \tag{1}$$

$$\mathcal{L}_{\text{ORPO}_{\text{Align}}}^\beta = -\log \sigma(\beta r_\theta^{\text{odds}}(y_w, x) - \beta r_\theta^{\text{odds}}(y_l, x)). \tag{2}$$

Both $\mathcal{L}_{\text{ASFT}}^\beta$ and $\mathcal{L}_{\text{ORPO}}^\beta$ generalize their vanilla counterparts (recovering them when $\beta = 1$). As in DPO, $\beta$ can be viewed as a *temperature* or *scaling* parameter that regulates the intensity of the preference for "good" odds. See Appendix E for gradient formulations and more details on these methods.

This unification (introduced not as a proposal of new standalone methods, but to enable consistent evaluation) raises **RQ2**: *"Does the tempering factor enhance the alignment quality of ASFT and ORPO?"* in Section 5.2 and enables a direct comparison of all methods across different setups.

### 3.2 ON THE DIFFERENCE BETWEEN DIRECT ALIGNMENT ALGORITHMS

By unifying ORPO and ASFT within a common two-stage framework parameterized by $\beta$, we place all DAAs on comparable footing. In this view, two axes of variation become explicit: (i) the scalar score used in the objective ($r_\theta^{\text{ref}}$[1] vs. $r_\theta^{\text{odds}}$), and (ii) whether the loss is defined over pairwise preferences or pointwise scores. The first axis follows directly from how existing losses are written, but the second has rarely been highlighted in prior work despite being a fundamental design choice.

The distinction between $r_\theta^{\text{ref}}$ and $r_\theta^{\text{odds}}$ is structural. $r_\theta^{\text{ref}}$ originates in RLHF, whereas the $r_\theta^{\text{odds}}$ is derived from odds-ratio objective. Empirical evidence comparing these scores in a standardized setting is, to our knowledge, still scarce. In contrast, the difference between pointwise and pairwise methods is functional: pairwise methods (DPO, IPO, SimPO, ORPO) depend on relative reward differences between candidate texts, whereas pointwise methods (APO-Zero, NCA, Cal-DPO, ASFT)

---

[1]SimPO does not explicitly use a reference policy, but can be treated similarly if a uniform reference policy is assumed.

maximize the probability of chosen sequences and minimize that of rejected ones independently of their mutual gap. This echoes empirical findings in learning-to-rank (Liu et al., 2009; Burges et al., 2005; Li, 2011; Melnikov et al., 2016), where pairwise objectives often yield more robust ranking signals than pointwise ones, though the precise reasons and applicability to LLM alignment remain under active investigation.

The experiments reported in Section 5.3 examine our **RQ3**: *"What factors of DAAs affect alignment quality?"* – the scalar score ($r_\theta^{\text{ref}}$ vs. $r_\theta^{\text{odds}}$) or the ranking type (pairwise vs. pointwise) – has the greatest impact on DAA performance.

## 4 EXPERIMENTAL SETUP

We systematically compare and evaluate DAA methods using a standard training and instruction-following evaluation framework Tunstall et al. (2023); Meng et al. (2024); Gorbatovski et al. (2024). Our main experiments use the Llama 3.1 8B model AI@Meta (2024), trained on the UltraChat Ding et al. (2023) and UltraFeedback (UF) Cui et al. (2023) datasets, and evaluated on the AlpacaEval 2 Dubois et al. (2024); Li et al. (2023) and ArenaHard Li et al. (2024b) benchmarks. For the Reddit TL;DR Stiennon et al. (2020) task, we employ the Llama 3.2 3B model, comparing it side by side with the "golden" validation split Rafailov et al. (2023; 2024) using the prompt in Appendix K.

### 4.1 BASE VS SFT-INITIALIZED MODELS.

To investigate the impact of SFT and the applicability of one-stage loss $\mathcal{L}_{\text{Align}}$ component, we use the UF dataset for SFT (avoiding additional knowledge from UltraChat), and for pairwise preference optimization. We carefully tuned the hyperparameters to optimize each method's performance.

For the *Base-initialized* setup, we perform a grid search over learning rates $\{6 \times 10^{-6}, 8 \times 10^{-6}, 1 \times 10^{-5}\}$, inspired by values suggested in ORPO and ASFT, and explore $\lambda \in \{0.1, 0.2, 0.5, 1.0\}$ for 1 and 2 training epochs keeping a similar budget to compare with the *SFT-initialized* setup.

In the *SFT-initialized* setup, we experiment with both $\mathcal{L}_{\text{ORPO}_{\text{Align}}}$ and $\mathcal{L}_{\text{ASFT}_{\text{Align}}}$ alone, as well as in combination with $\mathcal{L}_{\text{SFT}}$, following the original methods. We tune the learning rates $\{5 \times 10^{-7}, 7 \times 10^{-7}, 1 \times 10^{-6}\}$ for one epoch, starting from an SFT model trained for 1 epoch at $6 \times 10^{-6}$.

### 4.2 $\beta$ SENSITIVITY.

Following the adaptation of ASFT and ORPO to include a $\beta$ parameter (Section 3.1.2), all DAAs under consideration can now be compared on a more consistent basis. We conduct a comprehensive $\beta$-sensitivity analysis to (i) evaluate the impact of the $\beta$ parameter on the performance of ORPO and ASFT, and (ii) determine the peak alignment capabilities and relative performance of each method. We consider three scenarios:

**Llama 3.2 3B TL;DR.** A relatively simpler Reddit TL;DR summarization task, evaluated via GPT side-by-side comparison on 500 samples from the "golden" validation split Rafailov et al. (2023; 2024).

**Llama 3.2 3B UF.** The UltraChat and UF datasets serve as more challenging alignment settings due to their coverage of diverse and complex tasks, including instruction following, code generation, creative writing, common sense reasoning, mathematical problem-solving, and general knowledge.

**Llama 3.1 8B UF.** A larger, more capable model on the same UltraChat and UF datasets, allowing us to assess how increased model capacity influences $\beta$-sensitivity in these diverse tasks.

For the UF-based experiments, we measure quality using AlpacaEval 2 Length-Controlled (LC) Win-Rate and ArenaHard (AH) WR; for TL;DR, we rely on GPT-4o (2024-08-06) preference judgments. In each scenario, we sweep at least six $\beta$ values and four learning rates $\{1 \times 10^{-6}, 7 \times 10^{-7}, 5 \times 10^{-7}, 3 \times 10^{-7}\}$ to determine peak alignment capabilities and relative performance. As an auxiliary diagnostic, we report KL divergence to a reference model and plot quality–KL Pareto fronts. For *RQ3*, we also test DAA peak-performance generalization on math reasoning with Qwen 2.5 (7B/14B) Yang et al. (2024) (Appendix B.1). Further implementation details, including training procedures and generation hyperparameters, are provided in Appendix A.

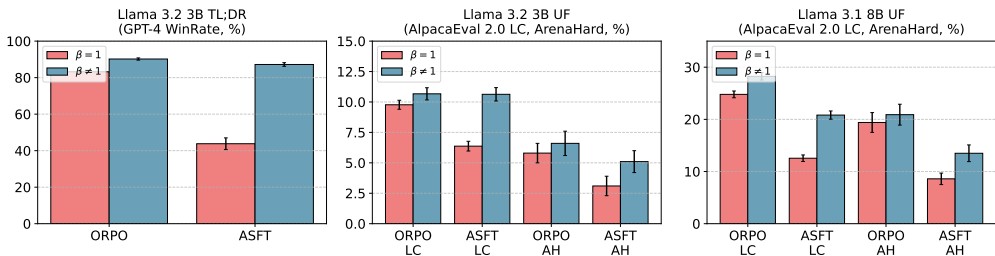

Figure 2: **Impact of the $\beta$ Parameter on ASFT and ORPO Alignment Quality.** The plot shows how tuning $\beta$ (Section 3.1.2) affects both ASFT and ORPO performance. Results are reported for GPT-4 Win Rate in the Llama 3.2 3B TL;DR setup and for AlpacaEval 2 LC Win Rate in the Llama 3.1 8B UF scenario. All other hyperparameters (e.g., learning rates) are selected via grid search, using each method's best configuration at $\beta = 1$ as the baseline. See Section 5.2 for more details.

### 4.3 SFT Data Quantity.

Our findings in Section 5.1 show that introducing an explicit SFT phase improves alignment quality - even for originally one-stage methods such as ORPO and ASFT. This enables a unified two-stage setup across all DAAs, where alignment begins from an SFT-initialized model. Given prior work on instruction tuning data efficiency Zhou et al. (2024) and distribution shift problem Xu et al. (2024), we prepare ablation study on sensitive different DAAs to SFT data volume.

We prepared seven SFT checkpoints by training Llama 3.1 8B Base on 1%, 3%, 5%, 10%, 25%, 50%, and 100% of the UltraChat dataset (ranging from 2,079 to 207,865 records) using our *SFT-initialized* setup. We then applied each alignment method – using *optimal hyperparameters* from our $\beta$-sensitivity experiments (Appendix Table 8) – to these seven SFT checkpoints and the original base model. Finally, we used AlpacaEval 2 LC to assess how model performance varies with the amount of SFT data used.

## 5 RESULTS

### 5.1 RQ1: DOES CONVERTING ORPO AND ASFT TO A TWO-STAGE PIPELINE IMPROVE ALIGNMENT QUALITY?

As shown in Table 1, the performance of ORPO and ASFT methods improves significantly when the alignment loss $\mathcal{L}_{\text{Align}}$ is applied after a preceding SFT stage. In particular, ORPO achieves results comparable to classical DPO in both LC Win Rate and AH WR metrics. In contrast, ASFT shows notable gains in AH WR after the SFT stage, although it still underperforms compared to ORPO or DPO. This performance difference aligns with our theoretical insights (Corollary D.2), as optimizing the ASFT objective, an upper bound on ORPO, appears less effective.

For one-stage methods, the use of $\lambda = 1$ provides the best results within the explored grid of $\lambda \in \{0.1, 0.2, 0.5, 1.0\}$, especially after two epochs of training. However, combining $\mathcal{L}_{\text{SFT}}$ and $\mathcal{L}_{\text{Align}}$ in a one-stage setup leads to suboptimal results compared to explicitly separating these phases, even when starting from an SFT-

| Init | Method | LC% (std) | WR% (std) | AH% (CI) |
|------|--------|-----------|-----------|----------|
| Base | SFT | 6.7 (0.43) | 4.5 (0.63) | 3.5 (-0.7, 0.8) |
| SFT | ORPO | **24.1** (0.84) | 17.8 (1.17) | 15.3 (-1.6, 1.8) |
| SFT | ASFT | 16.4 (0.72) | 11.9 (0.99) | 10.6 (-1.2, 1.3) |
| Base | ORPO† | 14.8 (0.71) | 10.3 (0.95) | 8.4 (-1.3, 1.3) |
| Base | ASFT† | 14.5 (0.73) | 10.2 (0.94) | 7.5 (-1.1, 1.2) |
| SFT | ORPO† | 13.4 (0.69) | 9.3 (0.91) | 7.7 (-0.9, 1.1) |
| SFT | ASFT† | 11.4 (0.63) | 7.5 (0.83) | 7.5 (-1.1, 1.1) |
| SFT | DPO | 23.4 (0.85) | **20.0** (1.18) | **17.5** (-1.8, 1.8) |

Table 1: **Base and SFT-initialized alignment methods on the Llama 3.1 8B model with the UF dataset.** SFT-initialized methods demonstrate better performance compared to their traditional formulations without $\mathcal{L}_{\text{SFT}}$. Results marked with † correspond to training with $\mathcal{L}_{\text{SFT}}$, using the best hyperparameters: lr $= 1 \times 10^{-6}$ for ORPO and lr $= 7 \times 10^{-7}$ for ASFT. For other setups, the best hyperparameters are: lr $= 5 \times 10^{-7}$ for standard SFT ORPO/ASFT, and lr $= 1 \times 10^{-5}/6 \times 10^{-6}$ for Base ORPO/ASFT.

trained model. Incorporating an explicit SFT stage improves overall performance for ORPO and ASFT methods. Therefore, all further experiments focus on applying the $\mathcal{L}_{\text{Align}}$ components of ORPO and ASFT on top of an SFT-trained model.

## 5.2 RQ2: Does the tempering factor enhance the alignment quality of ASFT and ORPO?

Figure 2 illustrates that introducing the $\beta$ parameter (as described in Section 3.1.2) improves the performance of both ASFT and ORPO $\mathcal{L}_{\text{Align}}$ in our tested scenarios. For a fair comparison, we used the best-performing learning rate for each baseline ($\mathcal{L}_{\text{ASFT}_{\text{Align}}}$ and $\mathcal{L}_{\text{ORPO}_{\text{Align}}}$) while fixing $\beta = 1$. In the Llama 3.2 3B TL;DR experiment, these adjustments led to an improvement of +7.0 for ORPO and +43.4 for ASFT in GPT-4 WR. In the Llama 3.1 8B UF setup, tuning $\beta$ provided additional gains of +3.46 for ORPO and +8.27 for ASFT on the AlpacaEval 2 LC WR.

## 5.3 RQ3: What factors of DAAs affect alignment quality?

Following the setup and evaluation scenarios described in Section 4.2, we assess the peak performance and KL divergence of each DAA under consideration, including the unified $\mathcal{L}_{\text{ASFT}_{\text{Align}}}^{\beta}$ and $\mathcal{L}_{\text{ORPO}_{\text{Align}}}^{\beta}$, under a common hyperparameter search space and two-stage training setup. Our analysis emphasizes how differences in scalar score ($r_{\theta}^{\text{ref}}$ vs. $r_{\theta}^{\text{odds}}$) and objective formulation (pairwise vs. pointwise) affect alignment quality.

|  | Win % | Tie % | Lose % |
|---|---|---|---|
| SFT | 35.6 | 4.8 | **59.6** |
| DPO | **91.2** | 1.0 | 7.8 |
| IPO | **91.4** | 0.4 | 8.2 |
| SimPO | **91.6** | 0.2 | 8.2 |
| ORPO | **90.2** | 0.6 | 9.2 |
| APO Zero | **92.6** | 0.6 | 6.8 |
| NCA | **91.8** | 1.0 | 7.2 |
| Cal-DPO | **91.4** | 0.4 | 8.2 |
| ASFT | **87.2** | 1.0 | 11.8 |

Table 2: **GPT-4 Evaluation of Llama 3.2 3B TL;DR setup.** The comparison shows multiple alignment methods (rows) using their best hyperparameters. Most methods exceed 90% Win Rate; ASFT achieves 87.2%, maintaining robust summarization performance. See Section 5.3 for more details.

**Llama 3.2 3B TL;DR:** Table 2 presents a comparison of all methods on the Reddit TL;DR validation subset, using their best hyperparameters. Most methods achieve a GPT-4 Win Rate exceeding 90%, indicating robust summarization performance on this relatively straightforward task. ASFT is slightly lower at 87.2% Win Rate, but still demonstrates strong overall results.

**Llama 3.2 3B UF and Llama 3.1 8B UF:** Table 3 summarizes the results for both Llama 3.2 3B UF and Llama 3.1 8B UF setups. For the smaller 3B model, the methods perform similarly on LC WR, with slight differences emerging on AH. Although these differences align with the pairwise vs. pointwise distinction (e.g., DPO, IPO, ORPO, SimPO vs. APO-Zero, NCA, Cal-DPO, ASFT), no single approach consistently dominates across metrics. The overlap in confidence intervals further indicates that the results for these methods are statistically similar in this setup, with no clear separation.

In contrast, the 8B model more clearly differentiates performance by ranking type: pairwise methods generally achieve higher peak scores on AlpacaEval 2 and ArenaHard, with ORPO best overall. On Qwen 2.5 7B/14B Math CoT, pairwise DAAs similarly match or exceed pointwise ones, while scalar score type ($r_{\theta}^{\text{ref}}$ vs. $r_{\theta}^{\text{odds}}$) yields no consistent performance differences (Appendix B.3). Note that $r_{\theta}^{\text{odds}}$-based methods do not start from KL $\approx 0$ at high $\beta$ since there is no explicit constraint toward $\pi_{\text{ref}}$; gradient scaling via $\beta$ still implicitly limits update magnitude (see Appendix E). Pareto fronts for the remaining setups are provided in Appendix H. For completeness, see Appendix I for results with varying lr/$\beta$ ratios.

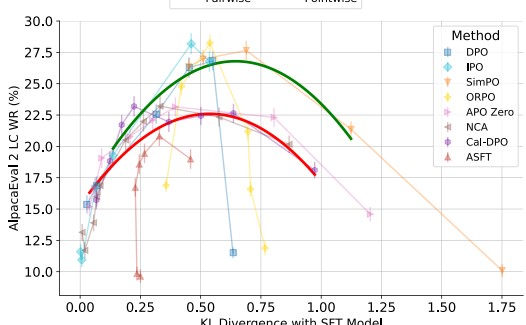

Figure 3: **Pareto front for alignment quality and KL divergence.** Results for Llama 3.1 8B UF on AlpacaEval 2 LC. Methods are grouped into pairwise and pointwise categories, with pairwise achieving higher LC values while remaining within overlapping confidence intervals.

| Method | Llama 3.2 3B UF | | | Llama 3.1 8B UF | | |
| --- | --- | --- | --- | --- | --- | --- |
| | AlpacaEval 2 | | ArenaHard | AlpacaEval 2 | | ArenaHard |
| | LC% (std) | WR% (std) | WR% (CI) | LC% (std) | WR% (std) | WR% (CI) |
| SFT | 5.02 (0.34) | 3.21 (0.55) | 1.4 (-0.4, 0.4) | 10.27 (0.54) | 5.44 (0.70) | 2.6 (-0.5, 0.6) |
| DPO | **11.43** (0.58) | 11.79 (0.99) | 6.8 (-1.0, 0.9) | 26.82 (0.77) | 23.69 (1.25) | 19.0 (-1.9, 1.8) |
| IPO | 11.24 (0.60) | 11.67 (1.01) | **6.8** (-1.0, 1.1) | 28.18 (0.83) | 24.43 (1.26) | 19.1 (-1.6, 1.5) |
| SimPO | 10.56 (0.44) | 11.94 (0.95) | 6.4 (-1.0, 1.1) | 27.65 (0.77) | 25.62 (1.29) | **21.5** (-1.9, 1.9) |
| ORPO | 10.67 (0.50) | **12.23** (0.97) | 6.6 (-1.0, 1.1) | **28.25** (0.71) | **28.59** (1.33) | 20.9 (-2.0, 2.0) |
| APO Zero | 10.36 (0.53) | 11.22 (0.98) | 6.0 (-1.0, 0.9) | 23.15 (0.76) | 19.03 (1.18) | 17.3 (-1.8, 1.8) |
| NCA | 10.33 (0.53) | 11.02 (0.97) | 5.1 (-0.7, 0.8) | 23.21 (0.80) | 18.67 (1.17) | 15.1 (-1.5, 1.6) |
| Cal-DPO | 10.62 (0.57) | 10.15 (0.94) | 4.8 (-0.9, 0.9) | 23.19 (0.82) | 18.85 (1.18) | 15.2 (-1.5, 1.6) |
| ASFT | 10.63 (0.55) | 9.21 (0.88) | 5.1 (-0.9, 0.9) | 20.82 (0.79) | 16.34 (1.13) | 13.5 (-1.6, 1.5) |

Table 3: **AlpacaEval 2 and ArenaHard Results for Llama 3.2 3B and Llama 3.1 8B UF.** The SFT model was trained on the UltraChat dataset. The best hyperparameters for each method were selected according to Section 4.2. Bold values indicate the best performance for each benchmark, while underlined values represent the second-best performance. See Section 5.3 for more details.

### 5.4 ABLATION STUDY ON SFT DATA VOLUME SENSITIVITY

Transforming ORPO and ASFT into two-stage methods enables a direct ablation on SFT data volume. Figures 4a and 4b show that all methods tend to saturate around 10% of UltraChat, reaching $\geq 95\%$ of their full-data performance. Pairwise methods generally achieve higher alignment quality than pointwise ones once the data exceeds 5%.

In the low-data regime (1-5%), DPO and IPO - both using a reference policy perform better. Interestingly, at 3% SFT, ASFT surpasses all other pointwise methods and some pairwise ones (e.g., ORPO, SimPO), while remaining behind DPO and IPO. These trends suggest nuanced dynamics worth further investigation and research. Nonetheless, the overall conclusion is clear - all DAAs benefit from SFT and require only 5-10% of the data to realize most of their alignment potential, regardless of their pairwise or pointwise formulation.

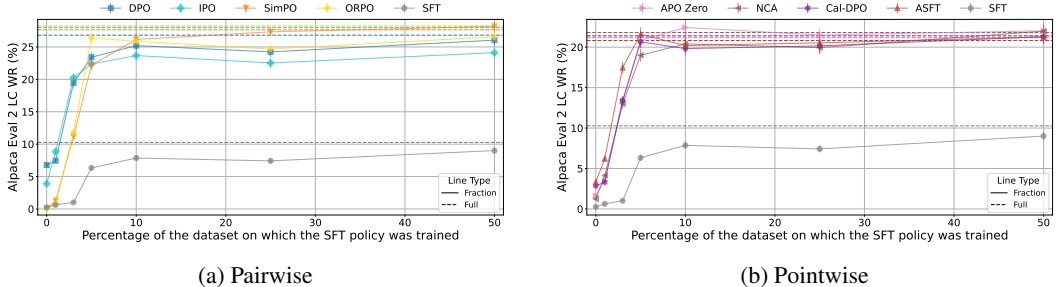

(a) Pairwise                                      (b) Pointwise

Figure 4: **Impact of SFT Dataset Size on Alignment Quality.** Performance of the pairwise (a) and pointwise (b) alignment methods on AlpacaEval 2 (LC WR metric) when the SFT policy is trained on different fractions of the UltraChat dataset. Even a small fraction of SFT data (e.g., 5-10%) yields substantial gains over starting from the raw base model. See Section 5.4 for more details.

## 6 DISCUSSION

Having combined all the results, one key question remains: *Why do pairwise objectives outperform pointwise ones?* First, assume that tasks may vary in difficulty depending on both the dataset and the model size. At the two extremes – very easy (simple datasets and large models) or very hard (difficult datasets and small models) – we observe (Llama 3.2 3B TL;DR/UF setups), little difference in quality between pointwise and pairwise approaches. For tasks of intermediate difficulty, however, pairwise methods consistently outperform pointwise ones (Llama 3.1 8B UF).

To understand why, observe that both $r_\theta^{\text{ref}}$ and $r_\theta^{\text{odds}}$ can be written using a single scoring function $r_\theta(x, y)$ defined over prompt-completion pairs. This allows us to define the marginalized score $\mathbb{E}_y[r_\theta(x, y)]$, which reflects how high or low will the average score be across all $y$ for a fixed $x$. Any dataset (and therefore any model trained on it) inherits a bias that mirrors $b_\theta(x) := \mathbb{E}_y[r_\theta(x, y)]$.

We hypothesise that observed performance gap stems from how each objective interacts with this bias, as formalized in Appendix G. This analysis assumes offline, static preference data, consistent with the scope of our experiments, and our conclusions do not automatically carry over to online or iterative preference optimization (e.g., Online DPO) settings. Once a model has learned part of the ranking among continuations for a prompt $x$, further optimization can move along two qualitatively different directions: (i) improving the ranking on harder or mis-ranked examples by changing the gaps $r_\theta(x, y_w) - r_\theta(x, y_l)$, and (ii) shifting all scores for that prompt in roughly the same direction, thereby modifying the marginalized score $b_\theta(x)$ while largely preserving the order. In the notation of Appendix G, pointwise objectives generically induce a non-zero total score gradient $G_\theta(x)$ and thus explicitly incentivize updates of type (ii), whereas pairwise objectives satisfy $G_\theta(x) = 0$ and are structurally indifferent to such uniform shifts.

Intuitively, pointwise methods keep pushing $r_\theta(x, y_w)$ upward and $r_\theta(x, y_l)$ downward even on already-easy pairs, implementing a form of bias *unlearning* on $b_\theta(x)$ that consumes capacity which could otherwise be spent on harder examples.

Pairwise training, by contrast, only requires that $y_w$ score higher than $y_l$ with strength controlled by $\beta$. Compared to pointwise losses, such pairwise updates structurally offer fewer direct avenues for reshaping the marginalized score $\mathbb{E}_y[r_\theta(x, y)]$ itself. Refining previously learned examples therefore consumes little extra capacity, allowing the model to focus on harder cases. Thus, for *hard* tasks there is insufficient capacity for the unlearning step, so both objectives perform similarly. For *easy* tasks, unlearning does not exhaust capacity, enabling pointwise methods to catch up. In the *intermediate* regime, capacity is sufficient to unlearn bias in pointwise methods, *but not* address harder examples, leading to a misalignment that makes pointwise objectives less efficient.

We ran additional experiments to test this hypothesis; the results appear in Appendix F. Previously, distinctions between DAA objectives were unclear, but our findings show that **they differ in how they handle dataset-induced biases**; whether bias removal is beneficial remains an open question. Beyond the $r_\theta^{\text{ref}}$ and $r_\theta^{\text{odds}}$ parameterizations, we also test (Appendix B.4) two scalar score families from recent work, AlphaPO (Gupta et al.) ($r_\alpha$) and the Forward-KL variant of f-DPO (Wang et al.), in our Qwen2.5–7B and 14B Math-CoT setup. In both cases, pairwise objectives again outperform their pointwise counterparts, further supporting our main findings in the static binary-preference regime of this paper.

# 7 RELATED WORK

Our work connects to several active research directions in preference optimization.

**Unifying frameworks for DAAs.** Recent works have developed increasingly general formulations of direct alignment objectives—spanning convex formulations (Tang et al., 2024b), $f$-divergences (Han et al., 2024; Wang et al.), mutual information views (Tutnov et al., 2025), and modular analyses of DPO variants via reward shaping or margins (Sun et al., 2025; Zhao et al., 2024; Gupta et al.; Wu et al.; Zhou et al., 2025). These advances focus primarily on the *pairwise* DPO-style family, exploring alternative score parameterizations and divergence measures while keeping the ranking objective itself fixed. In contrast, we establish a common ground for comparing *across* different algorithmic families (pairwise vs. pointwise; odds-ratio vs. policy–reference–ratio scores) that were previously incomparable due to disparate training protocols. Our unified perspective reveals that, across a broad range of score parameterizations (including AlphaPO-style rewards and the fKL variant of f-DPO), the choice of ranking objective is the main structural factor underlying DAA performance, with score parameterization playing only a secondary role.

**Beyond binary pairwise preferences.** Other studies investigate specific directions beyond the standard pairwise setup. Liu et al. (2025) frame alignment as listwise ranking, while methods like TriplePO and TreePO (Saeidi et al., 2025; Liao et al., 2024) leverage richer supervision signals (e.g., gold trajectories or multi-branch preference trees). These approaches, though promising, require

data formats beyond standard binary preferences and thus fall outside the scope of our controlled comparison of offline DAAs on static preference pairs.

**Online vs offline optimization.** Another direction compares offline DPO and RLHF, exposing DAAs' limits (Xu et al., 2024; Chu et al., 2025; Tang et al., 2024a), while Calandriello et al. (2024) study online preference optimization across contrastive vs. non-contrastive objectives Our work complements these by systematically isolating the impact of the ranking objective in the offline setting, a factor previously underexplored despite its fundamental role.

## 8 CONCLUSION

DAA research is fragmented, with many methods claiming superiority based on marginal differences. We provide the first unified framework that places all DAAs we study, including ORPO and ASFT, on equal footing by (i) reorganizing the explicit SFT term into a two-stage formulation and (ii) introducing a $\beta$ parameter. Within this setup, the previously under-explored ranking objective (pairwise vs. pointwise) emerges in our experiments as the primary driver of alignment quality, with differences in scalar score playing only a secondary role. Theoretical analysis, together with controlled experiments, links this effect to how objectives interact with prompt-specific bias, explaining why performance gaps appear mainly at intermediate task difficulty and model scale. Practically, we show that odds-ratio DAAs also benefit from SFT and $\beta$, and that most alignment gains can be achieved with only 5–10% of SFT data. This finding clarifies why previous claims of "best" DAA Meng et al. (2024); Xiao et al. (2024); Wang et al. (2024) often depend on underexplored details of setup and bias.

**Limitations & Future Work.** Our analysis is intentionally focused on the off-policy, SFT-based alignment setting, in order to disentangle conflicting claims among DAAs under controlled conditions. While our instruction-following results rely on GPT-based evaluation, we mitigate this by validating findings on specific task with verifiable metrics up to the 14B scale. Extending the unified framework to on-policy preference optimization remains an important direction, complementing prior online studies (Calandriello et al., 2024; Hanning Zhang, 2025). Our bias–capacity trade-off is supported by both toy experiments and ICC analysis on real data. Future work could formalize this mechanism and study its predictive power in broader alignment settings.

## ETHICS STATEMENT

This work complies with the ICLR Code of Ethics. All datasets used in our experiments are publicly available and widely adopted in the alignment and language modeling community. No human subjects, private, or sensitive data were used. We discuss fairness, bias, and the implications of prompt-specific biases in Section 6, and believe our findings do not pose immediate risks of misuse or harm. All code will be released for reproducibility and community benefit.

## REPRODUCIBILITY STATEMENT

To ensure reproducibility, we provide a detailed description of all methods and experimental protocols in Sections 2–4 and Appendix A. Hyperparameters, model checkpoints, and data splits are specified in the appendix and supplementary materials. Anonymized code for all experiments will be made available as part of the submission. Detailed proofs, derivations, and all additional results are included in the appendix.

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

## A  IMPLEMENTATION DETAILS

### A.1  PROBABILITY NORMALIZATION

| Method | Use normalization |
|---|:---:|
| DPO (Rafailov et al., 2023) | ✗ |
| IPO (Azar et al., 2024) | ✗ |
| SimPO (Meng et al., 2024) | ✓ |
| NCA (Chen et al., 2024) | ✗ |
| Cal-DPO (Xiao et al., 2024) | ✗ |
| APO-Zero (D'Oosterlinck et al., 2025) | ✗ |
| ORPO (Hong et al., 2024) | ✓ |
| ASFT (Wang et al., 2024) | ✓ |

Table 4: Methods that include (✓) or omit (✗) length-based probability normalization in their original formulation.

As discussed in Section 2.1, not all DAAs incorporate length-based probability normalization by default. In this paper, however, we apply such normalization only in cases where it was used in the original methods involving probabilities. This choice avoids introducing extra notation and reduces the cognitive load on the reader. Table 4 summarizes the methods that originally include length-based normalization.

## A.2 TRAINING DETAILS

Our experiments were conducted using the Llama 3.2 3B and Llama 3.1 8B Base models AI@Meta (2024). The training setup, datasets, and hyperparameters were designed to ensure reproducibility and consistency. Unless otherwise noted, the hyperparameters in Table 5 were used across all experiments.

Training was performed on 8 NVIDIA A100 GPUs with 80GB memory each. Depending on the number of epochs, training for each configuration took between 3 to 6 hours. The total compute used across all experiments amounted to approximately 651 GPU-days.

| Hyperparameter | Value |
|---|---|
| Max Tokens Length | 1024 (TL;DR setup), 4096 (UF setup) |
| Epochs | 1 *(or 2 when specified)* |
| Learning Rate (SFT) | $6.0 \times 10^{-6}$ |
| Learning Rate (Base Init.) | $\{6.0 \times 10^{-6}, 8.0 \times 10^{-6}, 1.0 \times 10^{-5}\}$ |
| Learning Rate (Alignment) | $\{3.0 \times 10^{-7}, 5.0 \times 10^{-7}, 7.0 \times 10^{-7}, 1.0 \times 10^{-6}\}$ |
| Optimizer | Adam (Kingma & Ba, 2014) |
| Adam $\beta_1$ | 0.9 |
| Adam $\beta_2$ | 0.95 |
| Batch Size | 128 |
| Learning Schedule | Linear Decay |
| Warm-up Ratio | 0.03 |
| Max Gradient Norm | 2 |
| Memory Optimization | DeepSpeed (Rasley et al., 2020) |
| Attention Mechanism | Flash Attention 2 (Dao, 2023) |

Table 5: Representative training hyperparameters for Llama 3.2 3B and Llama 3.1 8B models.

### A.2.1 DATASETS.

| Dataset | Training Examples | Validation Examples |
|---|---|---|
| UltraChat | 207,865 | 23,110 |
| UltraFeedback | 61,135 | 2,000 |
| Reddit TL;DR (SFT) | 41,947 | 11,941 |
| Reddit TL;DR (Preference) | 73,396 | 21,198 |

Table 6: Summary of dataset sizes used for training and validation.

We used two primary datasets:

- **Reddit TL;DR** (Bai et al., 2022): used to train the initial SFT model in $\beta$-sensitivity experiments with Llama 3.2 3B model.

- **UltraChat** (Ding et al., 2023): used to train the initial SFT model in $\beta$-sensitivity experiments with Llama 3.2 3B and Llama 3.1 8B models.

- **UltraFeedback** (Cui et al., 2023): used for both SFT (in the *Base vs. SFT-initialized* comparison, where we selected chosen subset from preference pairs) and for pairwise preference optimization in all DAA methods.

Dataset sizes are summarized in Table 6. For *Base* vs. *SFT-initialized* setups, only UltraFeedback was used. For the $\beta$-sensitivity experiments, the models were first trained on UltraChat for SFT and subsequently fine-tuned on UltraFeedback. The Reddit TL;DR dataset was deduplicated, retaining only uniquely preferred summaries for SFT.

### A.2.2 $\beta$-SENSITIVITY EXPERIMENTS.

We conducted a comprehensive analysis of the sensitivity of DAA methods to $\beta$, examining peak performance and the trade-offs between model quality and regularization strength (as reflected in KL divergence). Each method was trained with six or more distinct $\beta$ values to identify configurations that achieve stable and effective performance. The specific $\beta$ values tested for each method are shown in Table 7.

| Method | $\beta$ **Values Tested** |
|---|---|
| DPO | $\{0.001, 0.003, 0.005, 0.01, 0.05, 0.1\}$ |
| IPO | $\{0.0007, 0.001, 0.005, 0.01, 0.05, 0.1\}$ |
| SimPO | $\{0.05, 0.1, 0.2, 0.5, 1.0, 2.0, 5.0\}$ |
| ORPO | $\{0.05, 0.1, 0.2, 0.5, 1.0, 2.0\}$ |
| ASFT | $\{0.05, 0.1, 0.2, 0.5, 1.0, 2.0\}$ |
| APO-Zero | $\{0.001, 0.003, 0.005, 0.01, 0.05, 0.1, 0.2\}$ |
| Cal-DPO | $\{0.00005, 0.0001, 0.0003, 0.0005, 0.001, 0.003\}$ |
| NCA | $\{0.0001, 0.0003, 0.0005, 0.001, 0.005, 0.007, 0.01, 0.03, 0.05\}$ |

Table 7: Range of $\beta$ values tested for each DAA method on all Llama setups.

For each $\beta$, we tested four learning rates ($3.0 \times 10^{-7}$, $5.0 \times 10^{-7}$, $7.0 \times 10^{-7}$, $1.0 \times 10^{-6}$), training on the UltraFeedback dataset. All runs began from an SFT-initialized model trained on UltraChat (lr $= 6.0 \times 10^{-6}$, 1 epoch). The best-performing learning rate for each $\beta$ was selected to construct Pareto fronts, balancing quality (measured via AlpacaEval 2 LC Win-Rate) and KL divergence.

For SimPO in the Llama 3.1 8B UF setup, the ratio $\frac{\gamma}{\beta} = 0.5$ was kept fixed as recommended by Meng et al. (2024). Additionally, a single learning rate (lr $= 6.0 \times 10^{-7}$) was tested across all $\beta$ values for this method, as the same datasets and model scale were used. For Llama 3.2 TL;DR and UF setups, we tested four learning rates similar to other DAAs.

For DPO and IPO, the $3.0 \times 10^{-7}$ learning rate was not considered, as performance consistently deteriorated from $1.0 \times 10^{-6}$ to $5.0 \times 10^{-7}$, indicating that lower learning rates were unlikely to yield improvements.

Beyond the standard $\beta$ values described in Table 7, additional values were explored for specific configurations to reach the extreme points of the Pareto front. For example: - $\{0.00001, 0.00003\}$ for Cal-DPO in Llama 3.2 3B TL;DR and UF setups, - $\{0.00001, 0.00003, 0.00005\}$ for NCA in Llama 3.2 3B TL;DR, - $\{0.0003, 0.0005\}$ for APO-Zero in Llama 3.2 3B TL;DR, - $\{0.0003, 0.0005, 0.001, 0.003, 0.005\}$ for ASFT in Llama 3.2 3B TL;DR.

The hyperparameters resulting in the best performance are presented in Table 8.

### A.3 GENERATION DETAILS

We evaluated model performance on AlpacaEval 2 and ArenaHard for UltraFeedback setups, while for the Reddit TL;DR setup, we used side-by-side comparisons with GPT-4o on a curated golden validation subset of 500 samples. Additionally, KL divergence was measured on the validation subset for all setups using the generation hyperparameters listed in Table 9. For ArenaHard, the temperature was set to 0 to adhere to the original benchmark configuration.

| Method | Llama 3.2 3B TL;DR | | Llama 3.2 3B UF | | Llama 3.1 8B UF | |
|---|---|---|---|---|---|---|
| | **Learning Rate** | $\beta$ | **Learning Rate** | $\beta$ | **Learning Rate** | $\beta$ |
| DPO | $7.0 \times 10^{-7}$ | 0.05 | $1.0 \times 10^{-6}$ | 0.01 | $1.0 \times 10^{-6}$ | 0.003 |
| IPO | $1.0 \times 10^{-6}$ | 0.005 | $7.0 \times 10^{-7}$ | 0.001 | $1.0 \times 10^{-6}$ | 0.001 |
| SimPO | $3.0 \times 10^{-7}$ | 0.5 | $7.0 \times 10^{-7}$ | 1.0 | $6.0 \times 10^{-7}$ | 1.0 |
| ORPO | $3.0 \times 10^{-7}$ | 0.5 | $5.0 \times 10^{-7}$ | 0.2 | $5.0 \times 10^{-7}$ | 0.5 |
| ASFT | $3.0 \times 10^{-7}$ | 0.001 | $1.0 \times 10^{-6}$ | 0.2 | $7.0 \times 10^{-7}$ | 0.1 |
| APO Zero | $3.0 \times 10^{-7}$ | 0.001 | $3.0 \times 10^{-7}$ | 0.005 | $3.0 \times 10^{-7}$ | 0.003 |
| NCA | $3.0 \times 10^{-7}$ | 0.0001 | $3.0 \times 10^{-7}$ | 0.0005 | $3.0 \times 10^{-7}$ | 0.0003 |
| Cal-DPO | $3.0 \times 10^{-7}$ | 0.00003 | $5.0 \times 10^{-7}$ | 0.0003 | $3.0 \times 10^{-7}$ | 0.0003 |

Table 8: Best hyperparameters for each DAA method across Llama setups.

| Hyperparameter | Value |
|---|---|
| Temperature | 0.9 |
| Top-k | 40 |
| Top-p | 1.0 |
| Max New Tokens | 256 (TL;DR setup), 4096 (UF setup) |

Table 9: Generation hyperparameters for Llama 3.1 8B and Llama 3.2 3B models.

# B MATH REASONING EXPERIMENTS WITH QWEN2.5

To evaluate the generality of our findings for **RQ3**, we additionally consider mathematical reasoning tasks and a different model family (Qwen2.5), providing a judge-free evaluation environment that we assess at two scales, 7B and 14B.

## B.1 SETUP DETAILS

**Dataset.** We use the Math_CoT subset of the UltraInteract dataset Yuan et al. (2024), also employed in the NCA work Chen et al. (2024). The training split contains 78,080 examples with 266 validation samples for SFT, and 52,864 preference pairs with 279 validation pairs for alignment. Following standard practice, we prepend the following system prompt to all inputs: `"Please reason step by step, and put your final answer within \boxed{}."`

**Models.** Experiments are conducted with Qwen2.5-7B and Qwen2.5-14B Yang et al. (2024).

| Method | $\beta$ **Values Tested** |
|---|---|
| DPO | $\{0.001, 0.005, 0.01, 0.05, 0.1, 0.2\}$ |
| IPO | $\{0.0005, 0.001, 0.005, 0.01, 0.05, 0.1, 0.2\}$ |
| SimPO | $\{0.1, 0.2, 0.3, 0.5, 0.7, 1.0, 2.0, 5.0, 10.0, 15.0, 20.0\}$ |
| ORPO | $\{0.0005, 0.001, 0.005, 0.01, 0.05, 0.1, 0.2, 0.5, 1.0, 1.2, 2.0, 5.0, 10.0\}$ |
| ASFT | $\{0.01, 0.05, 0.1, 0.2, 0.5, 1.0, 2.0, 3.0, 5.0\}$ |
| APO-Zero | $\{0.0001, 0.0005, 0.001, 0.003, 0.01, 0.05, 0.1\}$ |
| Cal-DPO | $\{0.00001, 0.00003, 0.00005, 0.0001, 0.0003, 0.0005, 0.001, 0.003, 0.01, 0.05, 0.1\}$ |
| NCA | $\{0.0001, 0.0005, 0.001, 0.005, 0.01, 0.05, 0.1\}$ |

Table 10: Range of $\beta$ values tested for each DAA method on Qwen2.5 setups.

**Evaluation.** Performance is measured exclusively with verifiable metrics to avoid judge-model bias: GSM8K Cobbe et al. (2021), MATH500 Lightman et al. (2023), AMC23 Li et al. (2024a), MinervaMath Lewkowycz et al. (2022), and AIME24/25 Li et al. (2024a). We report average success rate

@k (avg@k) across 4 random seeds, with decoding hyperparameters `max_new_tokens=4096` and `temperature=1.0`.

**Training configuration.** The training protocol mirrors Section A, with four candidate learning rates $\{3.0\times10^{-7}, 5.0\times10^{-7}, 7.0\times10^{-7}, 1.0\times10^{-6}\}$. For Qwen2.5-7B, a full sweep over learning rates and $\beta$ values was conducted. For Qwen2.5-14B, the best learning rates from the 7B experiments were reused, while a full $\beta$ sweep was performed for each method. The ranges are reported in Table 10. For SimPO, we additionally swept the $\frac{\gamma}{\beta}$ ratio in $\{0.1, 0.2, 0.3, 0.5, 0.8, 1.0\}$. At 14B scale, an extended sweep including smaller values $\{0.001, 0.01, 0.03, 0.05\}$ was also performed.

## B.2 BEST HYPERPARAMETERS

Table 11 summarizes the best learning rates and $\beta$ values identified for each method.

| Method | Qwen2.5-7B | | Qwen2.5-14B | |
|---|---|---|---|---|
| | Learning Rate | $\beta$ | Learning Rate | $\beta$ |
| DPO | $5.0 \times 10^{-7}$ | 0.1 | $5.0 \times 10^{-7}$ | 0.2 |
| IPO | $1.0 \times 10^{-6}$ | 0.05 | $1.0 \times 10^{-6}$ | 0.01 |
| ORPO | $3.0 \times 10^{-7}$ | 0.001 | $3.0 \times 10^{-7}$ | 0.005 |
| SimPO | $3.0 \times 10^{-7}$ | 20.0 | $3.0 \times 10^{-7}$ | 20.0 |
| APO-Zero | $1.0 \times 10^{-6}$ | 0.01 | $1.0 \times 10^{-6}$ | 0.0005 |
| Cal-DPO | $1.0 \times 10^{-6}$ | 0.0003 | $1.0 \times 10^{-6}$ | 0.003 |
| NCA | $1.0 \times 10^{-6}$ | 0.001 | $1.0 \times 10^{-6}$ | 0.005 |
| ASFT | $3.0 \times 10^{-7}$ | 0.1 | $3.0 \times 10^{-7}$ | 1.0 |

Table 11: Best hyperparameters for Qwen2.5-7B and Qwen2.5-14B on Math_CoT (SimPO: best $\frac{\gamma}{\beta}$ was 0.8 for 7B and 0.05 for 14B).

## B.3 RESULTS

Tables 12 and 13 present the math benchmark results for Qwen2.5-7B and Qwen2.5-14B, respectively. For all datasets except AIME24 and AIME25, we report avg@8; for AIME24 and AIME25, we report avg@32 to better reflect performance due to their small question size. All results are averaged over 4 random seeds, with the corresponding standard deviations (computed across seeds) shown in parentheses.

At the 7B scale, pairwise and pointwise objectives perform comparably. At the 14B scale, however, a clear separation emerges: pairwise methods consistently outperform pointwise ones, whereas the scalar score axis ($r_\theta^{\mathrm{ref}}$ vs. $r_\theta^{\mathrm{odds}}$) yields no systematic differences. This replication of scale-dependent performance across instruction-following and mathematical reasoning strongly supports the generality of our main findings.

## B.4 ADDITIONAL SCALAR SCORE FAMILIES: ALPHAPO AND f-DPO (FORWARD-KL)

To test the robustness of our "ranking dominates" conclusion to changes in the scalar score family, we experimented with two alternative parameterizations drawn from recent work: the AlphaPO (Gupta et al.) reward $r_\alpha$ and the Forward-KL (fKL) score from f-DPO (Wang et al.), within the same static binary-preference setup.

**AlphaPO Scalar Score $r_\alpha$.** We adopt the scalar score

$$r_\alpha(y; x) = \frac{\beta}{\alpha}\Big(1 - \pi_\theta(y \mid x)^{-\alpha/|y|}\Big),$$

exactly as defined in AlphaPO, and plug it into two objectives: (i) a pairwise Bradley–Terry loss

$$L_{\mathrm{pair}} = -\log\sigma\big(r_\alpha(x, y_w) - r_\alpha(x, y_l)\big),$$

and (ii) a pointwise ASFT-style loss

$$L_{\text{point}} = -\log \sigma\big(r_\alpha(x, y_w)\big) - \log \sigma\big(-r_\alpha(x, y_l)\big).$$

In both cases, the scalar score $r_\alpha$ is identical; only the way it enters the loss (difference vs. separate terms) is changed.

| | Mean Avg@K | GSM8K | MATH500 | AMC23 | Minerva | AIME24 | AIME25 |
|---|---|---|---|---|---|---|---|
| Base | 0.1344 | 0.2233 | 0.2876 | 0.1805 | 0.0679 | 0.0292 | 0.0177 |
| | (0.0024) | (0.0012) | (0.0067) | (0.0248) | (0.0045) | (0.0035) | (0.0043) |
| SFT | 0.2216 | 0.6677 | 0.3671 | 0.1711 | 0.1135 | 0.0076 | 0.0029 |
| | (0.0019) | (0.0012) | (0.0026) | (0.0143) | (0.0055) | (0.0016) | (0.0010) |
| DPO | **0.3017** | 0.8295 | 0.5011 | 0.2641 | 0.1966 | 0.0107 | 0.0081 |
| | (0.0033) | (0.0015) | (0.0080) | (0.0174) | (0.0050) | (0.0033) | (0.0025) |
| IPO | *0.2996* | 0.8236 | 0.4866 | 0.2664 | 0.2007 | 0.0128 | 0.0078 |
| | (0.0035) | (0.0020) | (0.0055) | (0.0121) | (0.0027) | (0.0035) | (0.0013) |
| SimPO | 0.2892 | 0.7893 | 0.4837 | 0.2523 | 0.1898 | 0.0125 | 0.0073 |
| | (0.0039) | (0.0026) | (0.0037) | (0.0190) | (0.0014) | (0.0037) | (0.0026) |
| ORPO | 0.2966 | 0.8303 | 0.4882 | 0.2641 | 0.1759 | 0.0117 | 0.0094 |
| | (0.0010) | (0.0026) | (0.0046) | (0.0116) | (0.0047) | (0.0064) | (0.0039) |
| AlphaPO Pair | 0.2945 | 0.8147 | 0.4872 | 0.2578 | 0.1849 | 0.0143 | 0.0083 |
| | (0.0020) | (0.0014) | (0.0027) | (0.0141) | (0.0010) | (0.0021) | (0.0015) |
| APO-Zero | 0.2837 | 0.8071 | 0.4586 | 0.2352 | 0.1807 | 0.0115 | 0.0091 |
| | (0.0020) | (0.0030) | (0.0050) | (0.0069) | (0.0102) | (0.0031) | (0.0034) |
| NCA | 0.2861 | 0.7909 | 0.4715 | 0.2461 | 0.1922 | 0.0109 | 0.0047 |
| | (0.0027) | (0.0015) | (0.0026) | (0.0145) | (0.0017) | (0.0051) | (0.0010) |
| Cal-DPO | 0.2936 | 0.8272 | 0.4694 | 0.2531 | 0.1931 | 0.0109 | 0.0081 |
| | (0.0044) | (0.0014) | (0.0053) | (0.0209) | (0.0070) | (0.0028) | (0.0018) |
| ASFT | 0.2903 | 0.8132 | 0.4785 | 0.2437 | 0.1876 | 0.0130 | 0.0057 |
| | (0.0033) | (0.0020) | (0.0064) | (0.0149) | (0.0009) | (0.0028) | (0.0013) |
| AlphaPO Point | 0.2922 | 0.8233 | 0.4753 | 0.2539 | 0.1820 | 0.0122 | 0.0063 |
| | (0.0027) | (0.0012) | (0.0085) | (0.0074) | (0.0074) | (0.0021) | (0.0009) |
| fKL Pair | **0.2673** | 0.7825 | 0.4403 | 0.2141 | 0.1535 | 0.0073 | 0.0060 |
| | (0.0041) | (0.0031) | (0.0059) | (0.0195) | (0.0067) | (0.0017) | (0.0010) |
| fKL Point | 0.2643 | 0.7750 | 0.4363 | 0.2094 | 0.1521 | 0.0078 | 0.0052 |
| | (0.0020) | (0.0014) | (0.0032) | (0.0068) | (0.0033) | (0.0028) | (0.0019) |

Table 12: Math benchmark results for Qwen2.5-7B. Reported values are avg@8 (except AIME24/25: avg@32), averaged across 4 seeds; standard deviation across seeds is shown in parentheses.

We reuse the Math-CoT setup from Section B.1. For Qwen2.5–7B we perform a grid search over $\beta \in \{0.1, 0.5, 0.7, 1.0, 2.5, 5.0, 10.0, 20.0, 25.0\}$ and $\alpha \in \{-0.5, -0.1, 0.0, 0.1, 0.25, 0.5, 1.0, 2.0\}$, and then refine $\alpha \in \{2.5, 3.0, 5.0\}$ around the best $\beta \in \{0.5, 0.7, 1.0\}$. For Qwen2.5–14B we reuse the best learning rate from the 7B experiments and sweep $\alpha \in \{0.5, 1.0, 1.5, 2.0, 2.5\}$ for the same three $\beta$ values. In all runs we fix the learning rate at $3 \times 10^{-7}$, which is the best-performing value for SimPO in this math setup. The best configurations are $(\beta = 0.5, \alpha = 2.0)$ and $(\beta = 0.5, \alpha = 1.0)$ for pairwise/pointwise on Qwen2.5–7B, and $(\beta = 0.5, \alpha = 2.0)$ and $(\beta = 0.5, \alpha = 0.5)$ for pairwise/pointwise on Qwen2.5–14B.

The corresponding results are included in Tables 12 and 13. On both 7B and 14B, the AlphaPO-Pair variant lies in the same performance band as the other pairwise DAAs (DPO, IPO, SimPO, ORPO), while AlphaPO-Point tracks the pointwise cluster and is consistently slightly weaker than its pairwise counterpart. This mirrors the behavior observed for $r_\theta^{\text{ref}}$ and $r_\theta^{\text{odds}}$.

**f-DPO Forward-KL score.** For the f-DPO (Wang et al.) family we focus on the Forward-KL endpoint, which is known to underperform DPO overall, and study it under our unified protocol. We use the scalar score

$$r_\theta^{\text{fKL}}(x, y) = -\beta \frac{\pi_{\text{ref}}(y \mid x)}{\pi_\theta(y \mid x)},$$

corresponding to the $\alpha = 1$ case in the $\alpha$-divergence parameterization of f-DPO. As with $r_\alpha$, we keep $r_\theta^{\text{fKL}}$ fixed and compare a pairwise Bradley–Terry loss to a pointwise ASFT-style loss:

$$L_{\text{pair}} = -\log \sigma\big(r_\theta^{\text{fKL}}(x, y_w) - r_\theta^{\text{fKL}}(x, y_l)\big), \quad L_{\text{point}} = -\log \sigma\big(r_\theta^{\text{fKL}}(x, y_w)\big) - \log \sigma\big(-r_\theta^{\text{fKL}}(x, y_l)\big).$$

|  | Mean Avg@K | GSM8K | MATH500 | AMC23 | Minerva | AIME24 | AIME25 |
|---|---|---|---|---|---|---|---|
| Base | 0.1937 | 0.5343 | 0.3177 | 0.1773 | 0.1002 | 0.0188 | 0.0138 |
|  | (0.0065) | (0.0064) | (0.0048) | (0.0259) | (0.0069) | (0.0043) | (0.0054) |
| SFT | 0.2399 | 0.7162 | 0.4006 | 0.1719 | 0.1351 | 0.0115 | 0.0039 |
|  | (0.0013) | (0.0018) | (0.0105) | (0.0186) | (0.0052) | (0.0049) | (0.0013) |
| DPO | *0.3202* | 0.8509 | 0.5279 | 0.2773 | 0.2341 | 0.0221 | 0.0089 |
|  | (0.0030) | (0.0013) | (0.0080) | (0.0166) | (0.0085) | (0.0013) | (0.0045) |
| IPO | 0.3146 | 0.8584 | 0.5150 | 0.2594 | 0.2336 | 0.0164 | 0.0047 |
|  | (0.0014) | (0.0009) | (0.0031) | (0.0068) | (0.0067) | (0.0040) | (0.0018) |
| SimPO | 0.3148 | 0.8585 | 0.5446 | 0.2773 | 0.1823 | 0.0161 | 0.0099 |
|  | (0.0008) | (0.0035) | (0.0039) | (0.0053) | (0.0015) | (0.0047) | (0.0036) |
| ORPO | **0.3277** | 0.8690 | 0.5503 | 0.2883 | 0.2232 | 0.0219 | 0.0135 |
|  | (0.0066) | (0.0009) | (0.0049) | (0.0273) | (0.0062) | (0.0050) | (0.0054) |
| AlphaPO Pair | 0.3158 | 0.8693 | 0.5277 | 0.2695 | 0.1930 | 0.0234 | 0.0117 |
|  | (0.0019) | (0.0010) | (0.0128) | (0.0064) | (0.0034) | (0.0025) | (0.0037) |
| APO-Zero | 0.3081 | 0.8717 | 0.5052 | 0.2461 | 0.1965 | 0.0211 | 0.0078 |
|  | (0.0012) | (0.0018) | (0.0028) | (0.0053) | (0.0069) | (0.0032) | (0.0044) |
| NCA | 0.2979 | 0.8340 | 0.5058 | 0.2250 | 0.1983 | 0.0195 | 0.0049 |
|  | (0.0031) | (0.0034) | (0.0052) | (0.0238) | (0.0071) | (0.0021) | (0.0016) |
| Cal-DPO | 0.2943 | 0.8334 | 0.4946 | 0.2289 | 0.1916 | 0.0148 | 0.0026 |
|  | (0.0013) | (0.0008) | (0.0065) | (0.0097) | (0.0059) | (0.0027) | (0.0010) |
| ASFT | 0.3030 | 0.8330 | 0.5004 | 0.2492 | 0.2075 | 0.0216 | 0.0060 |
|  | (0.0042) | (0.0010) | (0.0040) | (0.0154) | (0.0051) | (0.0090) | (0.0031) |
| AlphaPO Point | 0.2923 | 0.8172 | 0.5032 | 0.2336 | 0.1766 | 0.0180 | 0.0049 |
|  | (0.0018) | (0.0017) | (0.0079) | (0.0208) | (0.0056) | (0.0035) | (0.0021) |
| fKL Pair | **0.2839** | 0.8215 | 0.4724 | 0.2125 | 0.1759 | 0.0169 | 0.0039 |
|  | (0.0029) | (0.0012) | (0.0027) | (0.0133) | (0.0064) | (0.0032) | (0.0029) |
| fKL Point | 0.2750 | 0.8125 | 0.4663 | 0.1852 | 0.1638 | 0.0164 | 0.0055 |
|  | (0.0036) | (0.0017) | (0.0017) | (0.0121) | (0.0067) | (0.0005) | (0.0010) |

Table 13: Math benchmark results for Qwen2.5-14B. Reported values are avg@8 (except AIME24/25: avg@32), averaged across 4 seeds; standard deviation across seeds is shown in parentheses.

On Qwen2.5–7B we sweep learning rates $\{3 \times 10^{-7}, 5 \times 10^{-7}\}$ and $\beta \in \{0.001, 0.005, 0.01, 0.05, 0.1, 0.2\}$ for both pairwise and pointwise formulations. On Qwen2.5–14B we reuse the best learning rate from 7B ($5 \times 10^{-7}$) and sweep the same $\beta$ values. The best $\beta$ values are 0.005 and 0.001 (pairwise/pointwise) for Qwen2.5–7B, and 0.005 for both pairwise and pointwise on Qwen2.5–14B. The resulting scores are reported alongside the other methods in Tables 12 and 13.

As expected from the original f-DPO results, the Forward-KL parameterization yields lower absolute performance than DPO / reverse-KL. However, the structural pattern is unchanged: in both 7B and 14B settings the fKL pairwise objective consistently outperforms its fKL pointwise counterpart, and the gap becomes more pronounced at 14B. This behavior matches what we observe for the other scalar score families considered in this paper and further supports our conclusion that, in the static binary-preference regime we study, the pairwise vs. pointwise ranking structure is the dominant driver of performance.

## C  EQUIVALENCE OF $\mathcal{L}_{\text{ASFT}_{\text{Align}}}$ AND BINARY CROSS-ENTROPY LOSS

**Lemma C.1.**
$$\log \sigma(r_\theta^{\text{odds}}(y, x)) = \log \pi_\theta(y|x)$$

*Proof.*

$$\log \sigma(r_\theta^{\text{odds}}(y, x)) = \log \sigma(\log \frac{\pi_\theta(y|x)}{1 - \pi_\theta(y|x)}) = \log \frac{1}{1 + e^{\log(1 - \pi_\theta(y|x)) - \log(\pi_\theta(y|x))}}$$

$$= \log \frac{1}{1 + \frac{1 - \pi_\theta(y|x)}{\pi_\theta(y|x)}} = -\log \left(1 + \frac{1 - \pi_\theta(y|x)}{\pi_\theta(y|x)}\right) = -\log \frac{\pi_\theta(y|x) + 1 - \pi_\theta(y|x)}{\pi_\theta(y|x)} = \log \pi_\theta(y|x).$$

$\square$

**Lemma C.2.**

$$\log \sigma(-r_\theta^{\text{odds}}(y, x)) = \log \left(1 - \pi_\theta(y|x)\right)$$

*Proof.*

$$\log \sigma(-r_\theta^{\text{odds}}(y, x)) = \log \sigma(-\log \frac{\pi_\theta(y|x)}{1 - \pi_\theta(y|x)}) = \log \frac{1}{1 + e^{\log(\pi_\theta(y|x)) - \log(1 - \pi_\theta(y|x))}} =$$

$$\log \frac{1}{1 + \frac{\pi_\theta(y|x)}{1 - \pi_\theta(y|x)}} = -\log \left(1 + \frac{\pi_\theta(y|x)}{1 - \pi_\theta(y|x)}\right) = -\log \frac{1 - \pi_\theta(y|x) + \pi_\theta(y|x)}{1 - \pi_\theta(y|x)} = \log(1 - \pi_\theta(y|x)).$$

$\square$

**Theorem C.3.** $\mathcal{L}_{\text{ASFT}_{\text{Align}}}$ *decomposes into likelihood and unlikelihood terms, corresponding exactly to the sum of binary cross-entropy (BCE) losses evaluated independently on the positive and negative samples:*

$$\mathcal{L}_{\text{ASFT}_{\text{Align}}} = -\log \pi_\theta(y_w|x) - \log \left(1 - \pi_\theta(y_l|x)\right).$$

*Proof.* To explicitly demonstrate this decomposition, we start from the definition of the ASFT loss:

$$\mathcal{L}_{\text{ASFT}} = -\log \pi_\theta(y_w|x) - \lambda \log \sigma(r_\theta^{\text{odds}}(y_w, x)) - \lambda \log \sigma(-r_\theta^{\text{odds}}(y_l, x)),$$

where the odds ratio is defined as:

$$r_\theta^{\text{odds}}(y, x) = \frac{\pi_\theta(y|x)}{1 - \pi_\theta(y|x)}.$$

Applying Lemma C.1 and Lemma C.2, we rewrite this as:

$$\mathcal{L}_{\text{ASFT}_{\text{Align}}} = -\log \pi_\theta(y_w|x) - \log \left(1 - \pi_\theta(y_l|x)\right),$$
$$\mathcal{L}_{\text{ASFT}} = -(1 + \lambda) \log \pi_\theta(y_w|x) - \lambda \log \left(1 - \pi_\theta(y_l|x)\right).$$

To illustrate the connection with the binary cross-entropy (BCE) loss explicitly, consider the BCE defined for an example $(x, y)$ with binary label $z \in \{0, 1\}$:

$$\mathcal{L}_{\text{BCE}}(y, z|x) = -z \log \pi_\theta(y|x) - (1 - z) \log(1 - \pi_\theta(y|x)).$$

Evaluating BCE independently at the chosen example $y_w$ (positive, $z = 1$) and rejected example $y_l$ (negative, $z = 0$), we have:

$$\mathcal{L}_{\text{BCE}}(y_w, 1|x) = -\log \pi_\theta(y_w|x),$$

$$\mathcal{L}_{\text{BCE}}(y_l, 0|x) = -\log \left(1 - \pi_\theta(y_l|x)\right).$$

Summing these two BCE terms yields exactly:

$$\mathcal{L}_{\text{BCE}}(y_w, 1|x) + \mathcal{L}_{\text{BCE}}(y_l, 0|x) = -\log \pi_\theta(y_w|x) - \log \left(1 - \pi_\theta(y_l|x)\right),$$

which matches precisely the alignment loss $\mathcal{L}_{\text{ASFT}_{\text{Align}}}$.

Thus $\mathcal{L}_{\text{ASFT}_{\text{Align}}}$ decomposes into two independent BCE terms, each representing likelihood and unlikelihood modeling separately. $\square$

## D  RELATIONSHIP BETWEEN ORPO AND ASFT LOSS FUNCTIONS

**Theorem D.1.** $\mathcal{L}_{\mathrm{ORPO}}$ *can be expressed as:*

$$\mathcal{L}_{\mathrm{ORPO}} = \mathcal{L}_{\mathrm{ASFT}} + \lambda \log \big( \pi_\theta(y_w|x)(1 - \pi_\theta(y_l|x)) + \pi_\theta(y_l|x)(1 - \pi_\theta(y_w|x)) \big).$$

*Proof.* We start by defining the ORPO loss:

$$\mathcal{L}_{\mathrm{ORPO}} = -\log \pi_\theta(y_w|x) - \lambda \log \sigma \left( \log \frac{\pi(y_w|x)}{1 - \pi(y_w|x)} - \log \frac{\pi(y_l|x)}{1 - \pi(y_l|x)} \right).$$

Expanding the second term using the identity $\log \sigma(x) = x - \log(e^x + 1)$, we get:

$$-\log \sigma \left( \log \frac{\pi_\theta(y_w|x)}{1 - \pi_\theta(y_w|x)} - \log \frac{\pi_\theta(y_l|x)}{1 - \pi_\theta(y_l|x)} \right)$$

$$= \log \frac{1 - \pi_\theta(y_w|x)}{\pi_\theta(y_w|x)} + \log \frac{\pi_\theta(y_l|x)}{1 - \pi_\theta(y_l|x)} + \log \left( \frac{\pi_\theta(y_w|x)(1 - \pi_\theta(y_l|x))}{\pi_\theta(y_l|x)(1 - \pi_\theta(y_w|x))} + 1 \right)$$

$$= \log \frac{1 - \pi_\theta(y_w|x)}{\pi_\theta(y_w|x)} + \log \frac{\pi_\theta(y_l|x)}{1 - \pi_\theta(y_l|x)} + \log \left( \frac{\pi_\theta(y_w|x) - 2\pi_\theta(y_w|x)\pi_\theta(y_l|x) + \pi_\theta(y_l|x)}{\pi_\theta(y_l|x)(1 - \pi_\theta(y_w|x))} \right)$$

$$= \underbrace{-\log \pi_\theta(y_w|x) - \log(1 - \pi_\theta(y_l|x)) + \log \big( \pi_\theta(y_w|x) - 2\pi_\theta(y_w|x)\pi_\theta(y_l|x) + \pi_\theta(y_l|x) \big)}_{\mathrm{ORPO_{Align}}}.$$

Combining all terms, we obtain:

$$\mathcal{L}_{\mathrm{ORPO}} = -(1 + \lambda) \log \pi_\theta(y_w|x) - \lambda \log(1 - \pi_\theta(y_l|x)) +$$

$$\lambda \log \big( \pi_\theta(y_w|x)(1 - \pi_\theta(y_l|x)) + \pi_\theta(y_l|x)(1 - \pi_\theta(y_w|x)) \big)$$

$$= \mathcal{L}_{\mathrm{ASFT}} + \lambda \log \big( \pi_\theta(y_w|x)(1 - \pi_\theta(y_l|x)) + \pi_\theta(y_l|x)(1 - \pi_\theta(y_w|x)) \big)$$

**Corollary D.2.** $\mathcal{L}_{\mathrm{ORPO}} \leq \mathcal{L}_{\mathrm{ASFT}}$ *and* $\mathcal{L}_{\mathrm{ORPO_{Align}}} \leq \mathcal{L}_{\mathrm{ASFT_{Align}}}$.

This follows from the fact that the additional term in $\mathcal{L}_{\mathrm{ORPO}}$ is non-positive when $\pi_\theta(y_w|x)$ and $\pi_\theta(y_l|x)$ lie in $[0, 1]$, and $\pi_\theta(y_w|x) + \pi_\theta(y_l|x) \leq 1$.

$\square$

## E  UNDERSTANDING TEMPERED ASFT AND ORPO

Consider gradients of $\nabla_\theta \mathcal{L}_{\mathrm{ASFT_{Align}}}^\beta$ and $\nabla_\theta \mathcal{L}_{\mathrm{ORPO_{Align}}}^\beta$:

$$\nabla_\theta \mathcal{L}_{\mathrm{ASFT_{Align}}}^\beta = -\beta \Big[ \big(1 - \sigma(\beta r_\theta^{\mathrm{odds}}(y_w, x))\big) \nabla_\theta r_\theta^{\mathrm{odds}}(y_w, x) + \sigma(\beta r_\theta^{\mathrm{odds}}(y_l, x)) \nabla_\theta r_\theta^{\mathrm{odds}}(y_l, x) \Big],$$

$$\nabla_\theta \mathcal{L}_{\mathrm{ORPO_{Align}}}^\beta = -\beta \Big[ \big(\nabla_\theta r_\theta^{\mathrm{odds}}(y_w, x) - \nabla_\theta r_\theta^{\mathrm{odds}}(y_l, x)\big) \times \Big(1 - \sigma\big(\beta r_\theta^{\mathrm{odds}}(y_w, x) - \beta r_\theta^{\mathrm{odds}}(y_l, x)\big)\Big) \Big],$$

where $\nabla_\theta r_\theta^{\mathrm{odds}}(y, x) = \frac{\nabla_\theta \log \pi_\theta(y|x)}{1 - \pi_\theta(y|x)}$.

When $\beta \to 0$, $\sigma(\beta \cdots) \approx \frac{1}{2}$, both methods aggressively improve the odds ratio (increasing for $y_w$ and decreasing for $y_l$). As $\beta$ increases, the updates become bounded by the factor $\sigma(\beta \cdots)$ (similar to a reward threshold in DPO). Hence, once the model improves, further updates are limited, either individually for $\mathcal{L}_{\mathrm{ASFT_{Align}}}^\beta$ or by pairwise ranking in $\mathcal{L}_{\mathrm{ORPO_{Align}}}^\beta$.

## F  EXPERIMENT ON PROMPT BIAS

To further investigate our hypothesis from Section 6 regarding how pairwise and pointwise objectives interact with prompt-specific biases, we designed a controlled toy experiment. The goal is to simulate the essential mechanics of DAA training and observe the behavior of different objectives under conditions with and without an artificially introduced prompt-specific bias.

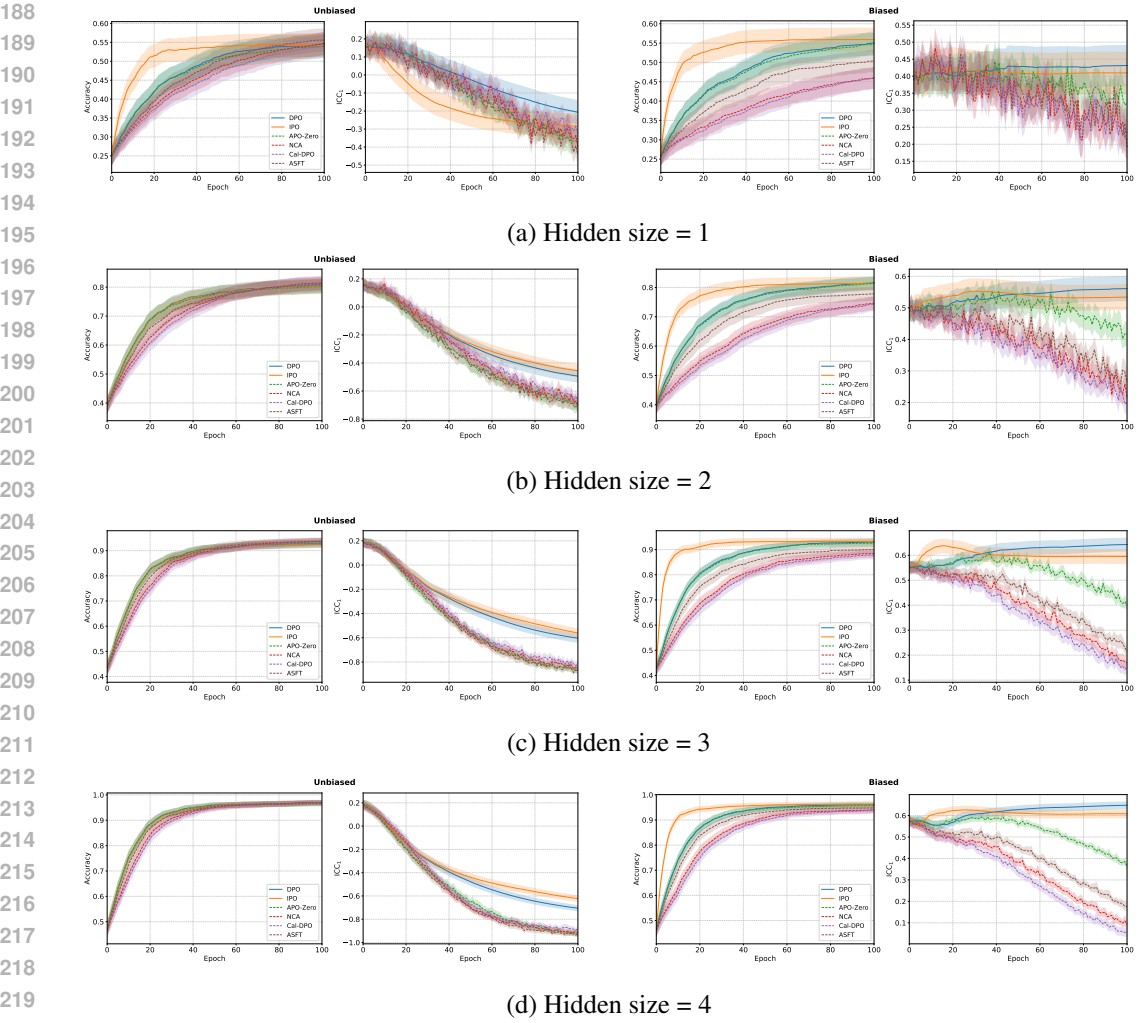

(a) Hidden size = 1

(b) Hidden size = 2

(c) Hidden size = 3

(d) Hidden size = 4

Figure 5: **Toy experiment: effect of model capacity ($h = 1, 2, 3, 4$) on accuracy and prompt bias** ($\text{ICC}_1$). Pairwise (solid) and pointwise (dashed) objectives compared under unbiased ($\text{bias\_strength} = 0.0$, left) and biased ($\text{bias\_strength} = 0.9$, right) conditions. Results averaged over 1000 seeds; 95% CI shown. See Section 6 for details.

**Experimental Setup.** For each run, we generate a dataset of $N = 2000$ samples. Each sample consists of a scalar prompt $x \sim \text{U}(0,1)$ and two scalar responses $s_{1,\text{base}}, s_{2,\text{base}} \sim \text{U}(0,1)$, representing the underlying "base quality" of the responses for that prompt.

Before introducing any bias, we center the base scores for each prompt:

$$\tilde{s}_{1,\text{base}} = s_{1,\text{base}} - \frac{1}{2}(s_{1,\text{base}} + s_{2,\text{base}}), \quad \tilde{s}_{2,\text{base}} = s_{2,\text{base}} - \frac{1}{2}(s_{1,\text{base}} + s_{2,\text{base}})$$

so that $\tilde{s}_{1,\text{base}} + \tilde{s}_{2,\text{base}} = 0$ for every prompt. This ensures that, in the absence of further modifications, there is no prompt-specific baseline in the response scores.

Next, we introduce prompt-specific bias by adding $b_x = \text{bias\_strength} \times \mathbb{I}(x < \text{bias\_threshold})$ to both centered scores, with $\text{bias\_threshold} = 0.5$ and $\text{bias\_strength}$ set to $0.0$ (unbiased) or $0.9$ (biased). The observed scores are therefore:

$$y_1 = \tilde{s}_{1,\text{base}} + b_x, \quad y_2 = \tilde{s}_{2,\text{base}} + b_x$$

For each prompt, the preferred ($y_w$) and dispreferred ($y_l$) observed scores are determined by applying the Bradley-Terry model (Bradley & Terry, 1952) to $(y_1, y_2)$ with a low temperature ($10^{-6}$),

making the assignment nearly deterministic: the higher of $y_1$ or $y_2$ is almost always selected as $y_w$, and the lower as $y_l$.

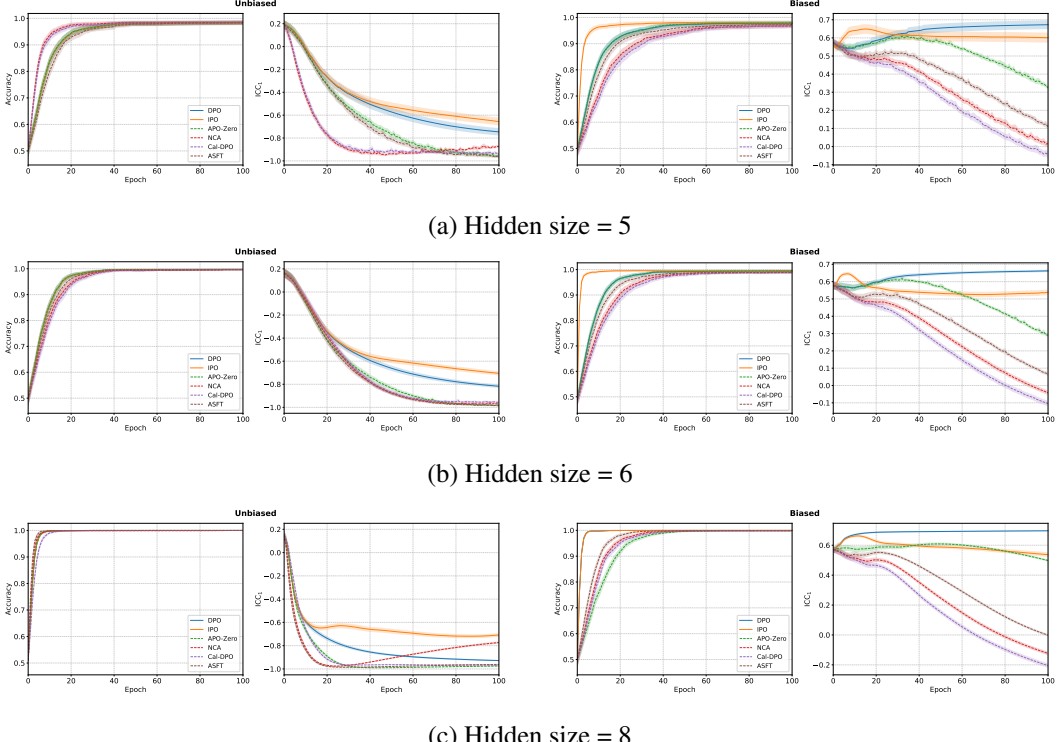

(a) Hidden size = 5

(b) Hidden size = 6

(c) Hidden size = 8

Figure 6: **Toy experiment: effect of model capacity ($h = 5, 6, 8$) on accuracy and prompt bias** ($ICC_1$). Pairwise (solid) and pointwise (dashed) objectives compared under unbiased (bias_strength = 0.0, left) and biased (bias_strength = 0.9, right) conditions. Results averaged over 1000 seeds; 95% CI shown. See Section 6 for details.

**Model and Training.** The model is a simple Multi-Layer Perceptron (MLP) with a single hidden layer and ReLU activation. It takes a 2-dimensional input (concatenation of the scalar prompt $x$ and scalar candidate response score $y$) and outputs a scalar score $r_\theta(x, y)$. We experiment with varying hidden layer sizes $h \in \{1, 2, 3, 4, 5, 6, 8\}$ to test different model capacities.

Since we focus solely on the bias-specific dependencies of each DAA objective, we do not investigate the differences between $r_\theta^{\mathrm{ref}}$ and $r_\theta^{\mathrm{odds}}$, operating exclusively with the scalar form $r_\theta(x, y)$. As a result, some of the loss functions discussed in Section 2 become equivalent in this context (for instance, DPO, SimPO, and ORPO) which we collectively refer to in this section as "DPO" for convenience. Other losses, such as APO-Zero, NCA, Cal-DPO, and ASFT, retain their distinct formulations involving $r_\theta(x, y)$, and are therefore referred to by their original names.

We fix $\beta = 1$ throughout, so that the scale of the loss does not confound the comparison of objectives; tuning $\beta$ merely regularizes the strength of preference optimization. This allows any differences in alignment to be attributed to the structural properties of the objectives.

Each configuration (objective, hidden size $h$, bias regime) is trained for 100 epochs, using 80% of the data for training and 20% for testing. For each configuration, the learning rate is selected by hyperparameter search over $\{0.3, 0.1, 0.05, 0.03, 0.01, 0.005, 0.003\}$ to maximize test alignment accuracy. All reported results are averaged over 1000 independent runs (with distinct random seeds for both data generation and model initialization). Confidence intervals are reported as $\pm 1.96\,\mathrm{SE}$, where SE is the standard error across runs.

We report two metrics on the test set: (i) accuracy, defined as the fraction of test pairs for which $r_\theta(x, y_w) > r_\theta(x, y_l)$; and (ii) the Intraclass Correlation Coefficient ($ICC_1$) Bartko (1966), which quantifies prompt-specific bias in the model's learned scores (see Appendix J for details).

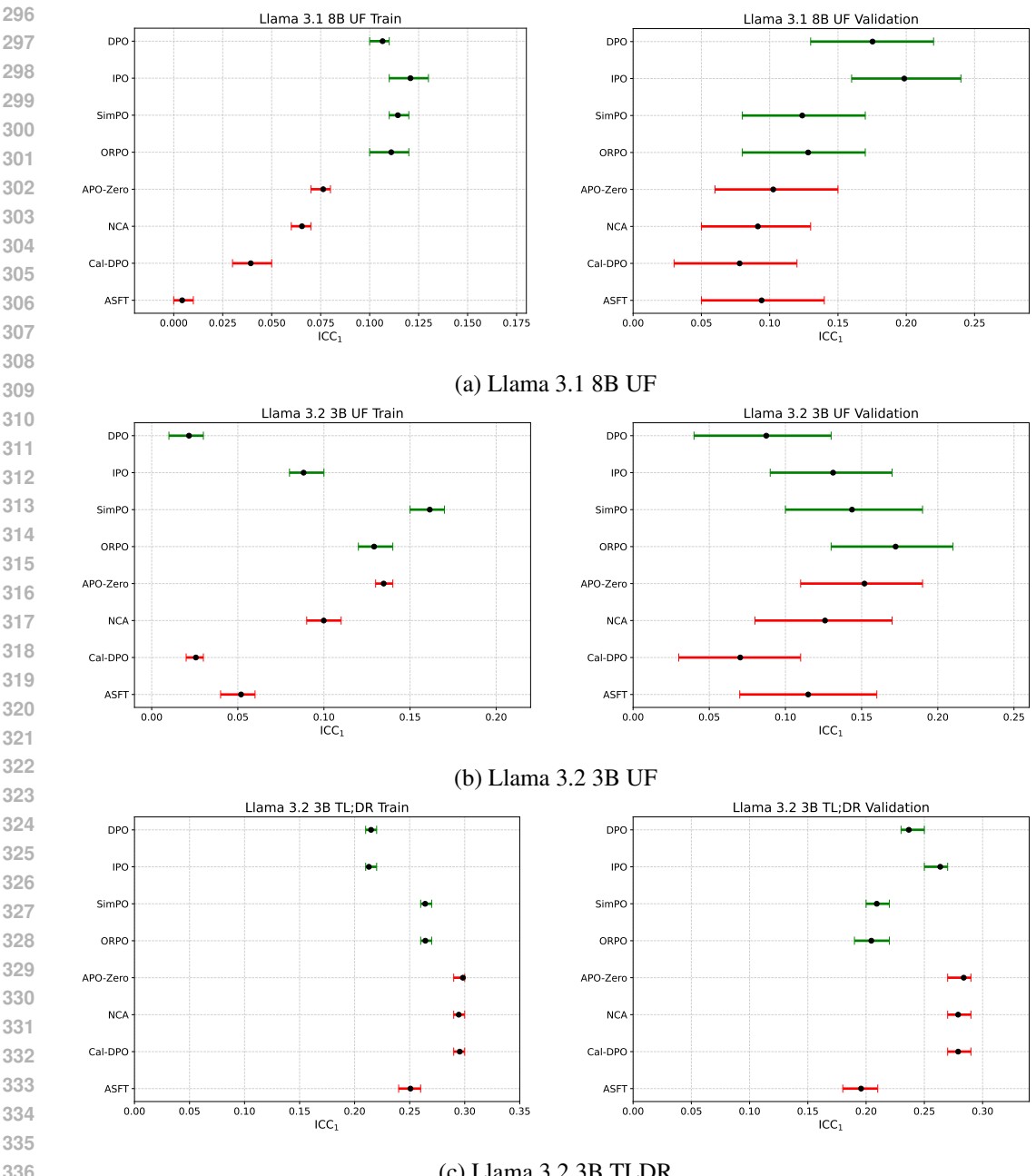

(a) Llama 3.1 8B UF

(b) Llama 3.2 3B UF

(c) Llama 3.2 3B TLDR

Figure 7: $ICC_1$ **on real data.** $ICC_1$ computed on the training and validation splits for the best model from each method, across **Llama 3.1 8B UF**, **Llama 3.2 3B UF**, and **Llama 3.2 3B TL;DR** setups. Error bars show $95\%$ confidence intervals. See Section 6 for details.

**Results.** Figures 5 and 6 present the results of the toy experiment, reporting test accuracy and $ICC_1$ across a range of model capacities (hidden dimension $h$), both for the unbiased (bias_strength $= 0.0$) and biased (bias_strength $= 0.9$) regimes.

In the unbiased condition (left panels), where the data contain no prompt-specific bias, all objectives – pairwise (DPO, IPO) and pointwise (ASFT, NCA, Cal-DPO, APO-Zero) – achieve identical accuracy for all $h$, and $ICC_1$ converges toward $-1$ as capacity increases. This confirms that when the underlying data are unbiased, neither class of objectives induces spurious prompt bias, and both are able to learn the quality structure of responses equally well.

In the biased condition (right panels), where prompt-specific bias is present in the data, the results partially mirror what we observe on real data. Examining $\mathrm{ICC}_1$, we see that our hypothesis is confirmed: pointwise methods reduce prompt bias, as indicated by lower $\mathrm{ICC}_1$, while for pairwise methods, $\mathrm{ICC}_1$ plateaus at a higher value. When comparing pointwise objective with $h = 1$ and $h = 3$, for $h = 3$ the reduction in $\mathrm{ICC}_1$ is more pronounced than for $h = 1$, indicating that a model with greater capacity is better able to reduce prompt bias.

If we examine the standard errors of accuracy, for $h = 1$ (which is most analogous to the Llama 3.2 3B UF setup), there is substantial overlap in the SE intervals across all methods. This closely resembles the trends observed in the ArenaHard column of Table 3, where IPO, DPO, SimPO, ORPO, and APO-Zero tend to achieve higher mean performance, while ASFT, NCA, and Cal-DPO are lower on average; however, the confidence intervals for many methods overlap, indicating that the differences are not always statistically significant in this lower-capacity regime. For $h = 3$, where in the pointwise case the model has more capacity to "spend" on removing bias, the gap between pairwise and pointwise objectives becomes more evident, mirroring the situation seen in Llama 3.1 8B UF. When $h > 4$, the task becomes trivial for the model, and the available capacity suffices both to minimize prompt bias and to achieve high ranking accuracy for all objectives; as a result, the performance of all methods converges. This parallels what we observe in the Llama 3.2 3B TL;DR setup.

These results are consistent with our hypothesis and provide strong evidence for why pairwise methods work better in certain regimes often encountered in real data - specifically, when the task is challenging enough that the model's capacity is insufficient to completely remove prompt bias. In such cases, differences between objectives are pronounced; for both very high and very low capacity, these differences vanish.

Additionally, Figure 7 reports $\mathrm{ICC}_1$ with $95\%$ confidence intervals, computed for the best-trained model of each method (hyperparameters in Table 8) on the training and validation splits (the large CI on the UF validation split is due to the small data size; see Table 6). Here, $r$ refers to $r_\theta^{\mathrm{ref}}$ and $r_\theta^{\mathrm{odds}}$ as appropriate for each method. These results also support our findings from the toy example and the hypothesis stated in Section 6: in Llama 3.1 8B UF, $\mathrm{ICC}_1$ is higher for pairwise methods, while for Llama 3.2 UF and TL;DR the results are mixed.

# G  THEORETICAL ANALYSIS OF PROMPT-SPECIFIC BIAS AND RANKING OBJECTIVES

In Section 6, we attribute the performance gap between pairwise and pointwise methods to how they interact with prompt-specific biases, specifically, that pointwise methods expend model capacity "unlearning" biases that pairwise methods naturally ignore. In this appendix, we formalize this mechanism. We define the *marginalized score* (prompt bias) for a general class of scalar score functions and analyze the gradient dynamics of pairwise and pointwise objectives with respect to this quantity.

## G.1  GENERAL SETUP AND DEFINITIONS

Consider a conditional language model parameterized by $\theta$, denoting the probability of a response $y$ given prompt $x$ as $\pi_\theta(y|x)$. Let $\mathcal{D}$ be a dataset of preference pairs $(x, y_w, y_l)$.

**Definition G.1** (Scalar Score Family). We assume the scalar score used for alignment takes the general form:

$$r_\theta(x, y) = F\big(\pi_\theta(y \mid x), \mathcal{C}(x, y)\big)$$

where $F : [0, 1] \times \mathcal{Z} \to \mathbb{R}$ is differentiable with respect to its first argument (the probability $\pi_\theta(y \mid x)$). The term $\mathcal{C}(x, y)$ represents fixed context-dependent quantities (e.g., reference model probabilities $\pi_{\mathrm{ref}}(y \mid x)$, sequence lengths $|y|$) that are independent of $\theta$. The dependence on $\theta$ occurs only through $\pi_\theta(y \mid x)$. For notational convenience, when $\mathcal{C}(x, y)$ is fixed we will sometimes write $F(\pi_\theta(y \mid x))$ and leave the second argument implicit. This formulation covers $r_{\mathrm{ref}}$, $r_{\mathrm{odds}}$ and $r_\alpha$ by Gupta et al. DAA families used in our experiments.

**Definition G.2** (Marginalized Score / Prompt Bias). For a fixed prompt $x$, let $q_x(y)$ be the empirical marginal distribution of candidate responses appearing in the dataset for that prompt (i.e., the set of all $y_w$ and $y_l$ associated with $x$). We define the **marginalized score** (or prompt-specific bias) as:

$$b_\theta(x) := \mathbb{E}_{y \sim q_x}[r_\theta(x, y)] = \sum_{y \in \mathcal{Y}_x} q_x(y) F(\pi_\theta(y \mid x)).$$

By construction, $q_x$ is a probability distribution over $\mathcal{Y}_x$, so $\sum_{y \in \mathcal{Y}_x} q_x(y) = 1$, and it is entirely determined by the dataset (hence independent of $\theta$).

### G.2 GRADIENT SENSITIVITY OF THE MARGINALIZED SCORE

First, we must establish that $b_\theta(x)$ is indeed sensitive to model parameters. If $\nabla_\theta b_\theta(x) = 0$ everywhere, the distinction between objectives would be moot.

Let $z(x)$ denote the vector of unnormalized logits for the candidate set $\mathcal{Y}_x = \{y_1, \ldots, y_K\}$ such that $\pi_\theta(y_k|x) = \text{softmax}(z(x))_k = \frac{e^{z_k}}{\sum_j e^{z_j}}$.

**Lemma G.3** (Gradient of Marginalized Score w.r.t logits). *The gradient of the prompt bias $b_\theta(x)$ with respect to the logit $z_m$ of a specific candidate $y_m$ is:*

$$\frac{\partial b_\theta(x)}{\partial z_m} = \pi_\theta(y_m|x) \left[ q_x(y_m) F'(\pi_m) - \sum_{y \in \mathcal{Y}_x} q_x(y) F'(\pi_y) \pi_\theta(y|x) \right]$$

*where $\pi_k \equiv \pi_\theta(y_k|x)$ and $F'(p) = \frac{\partial F}{\partial p}$.*

*Proof.* Using the chain rule: $\frac{\partial b_\theta}{\partial z_m} = \sum_k q_x(y_k) F'(\pi_k) \frac{\partial \pi_k}{\partial z_m}$. Recall the softmax derivative $\frac{\partial \pi_k}{\partial z_m} = \pi_k(\delta_{km} - \pi_m)$. Substituting this:

$$\frac{\partial b_\theta}{\partial z_m} = \sum_k q_x(y_k) F'(\pi_k) [\pi_k(\delta_{km} - \pi_m)]$$

$$= q_x(y_m) F'(\pi_m) \pi_m - \pi_m \sum_k q_x(y_k) F'(\pi_k) \pi_k.$$

This completes the proof. □

**Assumption G.4** (Non-Degeneracy). We assume the score function $F$ and the data distribution $q_x$ are such that $\frac{\partial b_\theta(x)}{\partial z} \neq \mathbf{0}$.

*Remark:* This holds generically. From Lemma G.3, for the gradient to vanish for a fixed prompt $x$ and all logits $z_m$ we would need

$$q_x(y_m) F'(\pi_\theta(y_m \mid x)) = \text{const} \quad \text{for all } y_m \in \mathcal{Y}_x.$$

In other words, $q_x(y_m)$ would have to be exactly proportional to $1/F'(\pi_m)$, which imposes a highly specific compatibility between the fixed data distribution $q_x$ and the evolving model distribution $\pi_\theta(\cdot \mid x)$. Since $q_x$ is $\theta$-independent while $\pi_\theta$ changes throughout training, this condition defines at most a measure-zero set of parameter values and is not satisfied in generic training dynamics.

**Corollary G.5.** *The prompt bias $b_\theta(x)$ is a learnable functional of the model parameters. The model can* adjust it. *The question is:* do the objectives ask the model to adjust it?

## G.3    GRADIENT SIGNAL FOR BIAS UPDATE

We define a general preference loss over a dataset $\mathcal{D}$ as $L(\theta) = \mathbb{E}_{(x,y_w,y_l)\sim\mathcal{D}}[\ell(r_w, r_l)]$, where $r_w, r_l$ are the scalar scores for $y_w, y_l$.

We can analyze how the loss generates gradient signals to update the prompt bias. Let $g_\theta(x, y) = \frac{\partial L}{\partial r_\theta(x,y)}$ be the backpropagated gradient signal w.r.t the score of a specific response $y$.

We define the **total score gradient** for prompt $x$ as:

$$G_\theta(x) := \sum_{y \in \mathcal{Y}_x} g_\theta(x, y) = \sum_{y \in \mathcal{Y}_x} \frac{\partial L}{\partial r_\theta(x, y)}.$$

**Theorem G.6** (Gradient Signal for Bias). *The total score gradient $G_\theta(x) = \sum_{y \in \mathcal{Y}_x} g_\theta(x, y)$ quantifies the sensitivity of the loss $L$ to a hypothetical uniform shift in the scores for prompt $x$. Specifically, if it were possible to independently add a small constant $\varepsilon$ to $r_\theta(x, y)$ for all $y \in \mathcal{Y}_x$, the first-order change in the loss would be $\delta L = \varepsilon \cdot G_\theta(x)$.*

*This makes $G_\theta(x)$ a direct measure of the objective's incentive to alter the prompt bias $b_\theta(x)$. A non-zero $G_\theta(x)$ implies that the loss function, in isolation, generates a gradient signal that would drive such a uniform shift, and thus a change in the bias. Whether the model can perfectly satisfy this incentive depends on its parameterization and capacity.*

*Proof.* Consider a uniform shift $\delta r = \varepsilon$ applied to the scores of all responses $y \in \mathcal{Y}_x$. The induced change in the loss $L$ is, to first order:

$$\delta L = \sum_{y \in \mathcal{Y}_x} \frac{\partial L}{\partial r_\theta(x, y)} \cdot \delta r = \varepsilon \cdot \sum_{y \in \mathcal{Y}_x} g_\theta(x, y) = \varepsilon \cdot G_\theta(x).$$

This shows that $G_\theta(x)$ is the derivative of $L$ with respect to a uniform shift in the scores for prompt $x$.

Now, note that a uniform shift $\delta r = \varepsilon$ for all $y \in \mathcal{Y}_x$ will change the prompt bias $b_\theta(x)$ by $\varepsilon$, because:

$$\delta b_\theta(x) = \sum_{y \in \mathcal{Y}_x} q_x(y) \cdot \delta r = \varepsilon \cdot \sum_{y \in \mathcal{Y}_x} q_x(y) = \varepsilon.$$

Therefore, the gradient of $L$ with respect to a change in the bias $b_\theta(x)$ (via a uniform shift) is exactly $G_\theta(x)$. Hence, $G_\theta(x) \neq 0$ implies that the objective function, in isolation, generates a gradient signal that would drive such a uniform shift, and thus a change in the bias. We emphasize that this argument considers a *hypothetical* perturbation in the space of scalar scores $r_\theta(x, y)$: it characterizes the structural sensitivity of the loss to per-prompt shifts, independently of whether the model parameterization allows implementing an exact uniform shift in score space. $\qquad\square$

## G.4    INVARIANCE OF PAIRWISE OBJECTIVES

We now prove that pairwise objectives are structurally invariant to prompt bias.

**Definition G.7** (Pairwise Objective). A pairwise objective is defined as $\ell_{\text{pair}}(r_w, r_l) = \phi(r_w - r_l)$, for some differentiable $\phi : \mathbb{R} \to \mathbb{R}$. Examples include DPO ($\phi(u) = -\log \sigma(\beta u)$), IPO, SimPO, and ORPO (in the two-stage formulation).

**Theorem G.8** (Pairwise Invariance). *For any pairwise objective $\ell_{pair}$, the total score gradient $G_\theta(x)$ is zero for every prompt $x$:*

$$G_\theta(x) = \sum_{y \in \mathcal{Y}_x} g_\theta(x, y) = 0 \quad \forall x.$$

*Consequently, pairwise objectives do not generate a gradient signal for uniform shifts in scores (i.e., they are invariant to prompt bias).*

*Proof.* Consider the set of pairs $\mathcal{P}_x = \{(y_w^{(i)}, y_l^{(i)})\}$ for prompt $x$. The total loss is $L_x = \sum_i \phi(r_{w,i} - r_{l,i})$. The derivative w.r.t. the score of a generic candidate $y$ is:

$$g_\theta(x, y) = \sum_{i:y_w^{(i)}=y} \phi'(\Delta_i) + \sum_{i:y_l^{(i)}=y} (-\phi'(\Delta_i))$$

where $\Delta_i = r_{w,i} - r_{l,i}$. The total score gradient is:

$$G_\theta(x) = \sum_{y \in \mathcal{Y}_x} g_\theta(x, y)$$

$$= \sum_{y \in \mathcal{Y}_x} \left( \sum_{i:y_w^{(i)}=y} \phi'(\Delta_i) - \sum_{i:y_l^{(i)}=y} \phi'(\Delta_i) \right)$$

$$= \sum_i \phi'(\Delta_i) - \sum_i \phi'(\Delta_i) = 0.$$

This completes the proof. $\square$

### G.5 COUPLING OF POINTWISE OBJECTIVES

**Definition G.9** (Pointwise Objective). A pointwise objective decomposes into separate terms for winners and losers:
$$\ell_{\text{point}}(r_w, r_l) = \psi_+(r_w) + \psi_-(r_l).$$
Examples include NCA, Cal-DPO, and ASFT.

**Theorem G.10** (Pointwise Coupling). *For pointwise objectives of the form $\ell_{point}(r_w, r_l) = \psi_+(r_w) + \psi_-(r_l)$, the total score gradient $G_\theta(x)$ is:*

$$G_\theta(x) = \sum_i \left[ \psi'_+(r_{w,i}) + \psi'_-(r_{l,i}) \right],$$

*where the sum is over all preference pairs $(y_w^{(i)}, y_l^{(i)})$ for prompt $x$.*

*This sum is not constrained to be zero by the structure of the loss. For any individual pair, the condition for zero contribution, $\psi'_+(r_w) + \psi'_-(r_l) = 0$, imposes a specific relationship between $r_w$ and $r_l$; requiring this to hold simultaneously for all pairs defines a non-generic equilibrium. Hence, for generic score configurations $(r_{w,i}, r_{l,i})$, pointwise objectives generate a non-zero gradient signal $G_\theta(x)$, explicitly incentivizing the model to change the prompt bias $b_\theta(x)$.*

*Proof.* The total score gradient is derived as follows. For a pointwise objective, the loss for a single pair is $\ell = \psi_+(r_w) + \psi_-(r_l)$. The gradients with respect to the scores are:

$$\frac{\partial \ell}{\partial r_w} = \psi'_+(r_w),$$

$$\frac{\partial \ell}{\partial r_l} = \psi'_-(r_l).$$

Summing over all pairs for prompt $x$, we obtain:

$$G_\theta(x) = \sum_i \left[ \psi'_+(r_{w,i}) + \psi'_-(r_{l,i}) \right].$$

This sum is not structurally constrained to be zero. For example, ASFT with $\psi_+(r) = -\log \sigma(r)$ and $\psi_-(r) = -\log \sigma(-r)$, we have $\psi'_+(r) = \sigma(r) - 1$ and $\psi'_-(r) = \sigma(r)$, giving a sum of $\sigma(r_w) - 1 + \sigma(r_l)$ which is zero only when $\sigma(r_w) + \sigma(r_l) = 1$. $\square$

### G.6 SUMMARY

This analysis shows that **pairwise objectives are invariant, at the loss level, to uniform shifts in the scalar scores** for a given prompt: their total score gradient satisfies $G_\theta(x) = 0$, so they do not create a first-order incentive to change $b_\theta(x)$ as such. Throughout this appendix, we work in the standard offline alignment setting with static binary preference data, matching the scope of our experiments in the main paper; our conclusions are not claimed to automatically extend to online or iterative preference optimization regimes. Any evolution of the marginalized score under pairwise training arises only as a side effect of updates that modify score *differences*, i.e., the ranking structure.

In contrast, **pointwise objectives are structurally sensitive to such shifts**: for generic score configurations they yield $G_\theta(x) \neq 0$ and therefore explicitly encourage the model to adjust not only relative scores but also their absolute level for each prompt. In capacity-limited regimes, this additional requirement to manipulate the prompt bias $b_\theta(x)$ competes with the primary ranking task, providing a mechanistic explanation for the performance gap we observe between pointwise and pairwise methods in our experiments. Finally, we note that this argument is formulated in terms of first-order gradient signals in score space: a sufficiently expressive model could still realize bias removal under pairwise training as a side effect, but the loss itself provides no direct incentive to do so, which is the key distinction we draw here.

## H PARETO FRONTS FOR LLAMA 3.2 SETUPS

The results presented in this section correspond to the best hyperparameter configurations identified during the hyperparameter search described in Section 4.2, including the optimal learning rate for each method. This ensures that the Pareto fronts reflect the upper performance limits for alignment quality.

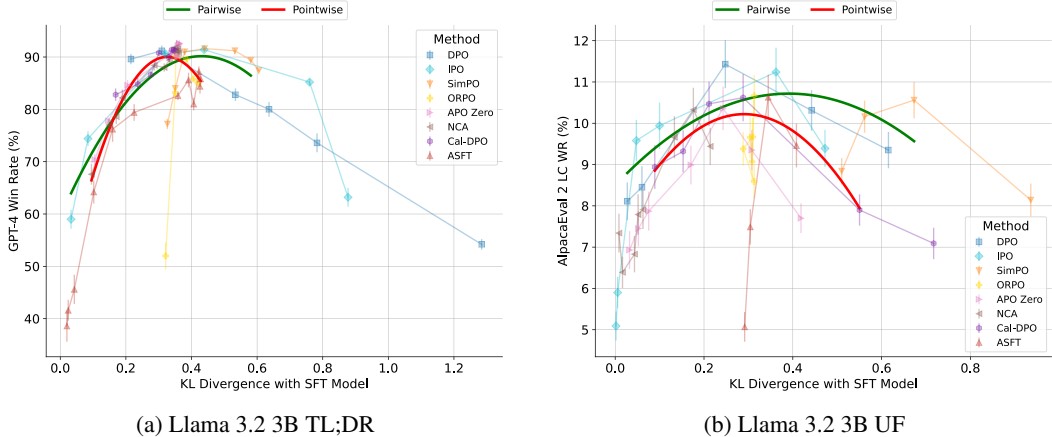

(a) Llama 3.2 3B TL;DR                          (b) Llama 3.2 3B UF

Figure 8: **Pareto front for alignment quality and KL divergence.** Results for Llama 3.2 3B TL;DR and UF setups on GPT-4 Win Rate vs. "golden" validation subset and AlpacaEval 2 LC respectively with different $\beta$ values. Methods are grouped into pairwise and pointwise categories. For the summarization task (Llama 3.2 3B TL;DR), both pointwise and pairwise methods achieve strong overall results. For the UF setup, methods also perform similarly within overlapping confidence intervals, indicating no clear separation.

## I LEARNING-RATE−TO−$\beta$ RATIO FOR DIFFERENT MODEL SIZES

Figure 9 shows that the relationship between alignment quality and the $lr/\beta$ ratio remains consistent across Llama 3B and 8B model scales for each method, despite minor shifts along the $x$-axis. The only noticeable trend is that $r_\theta^{\mathrm{ref}}$-based methods are generally more stable and less prone to qual-

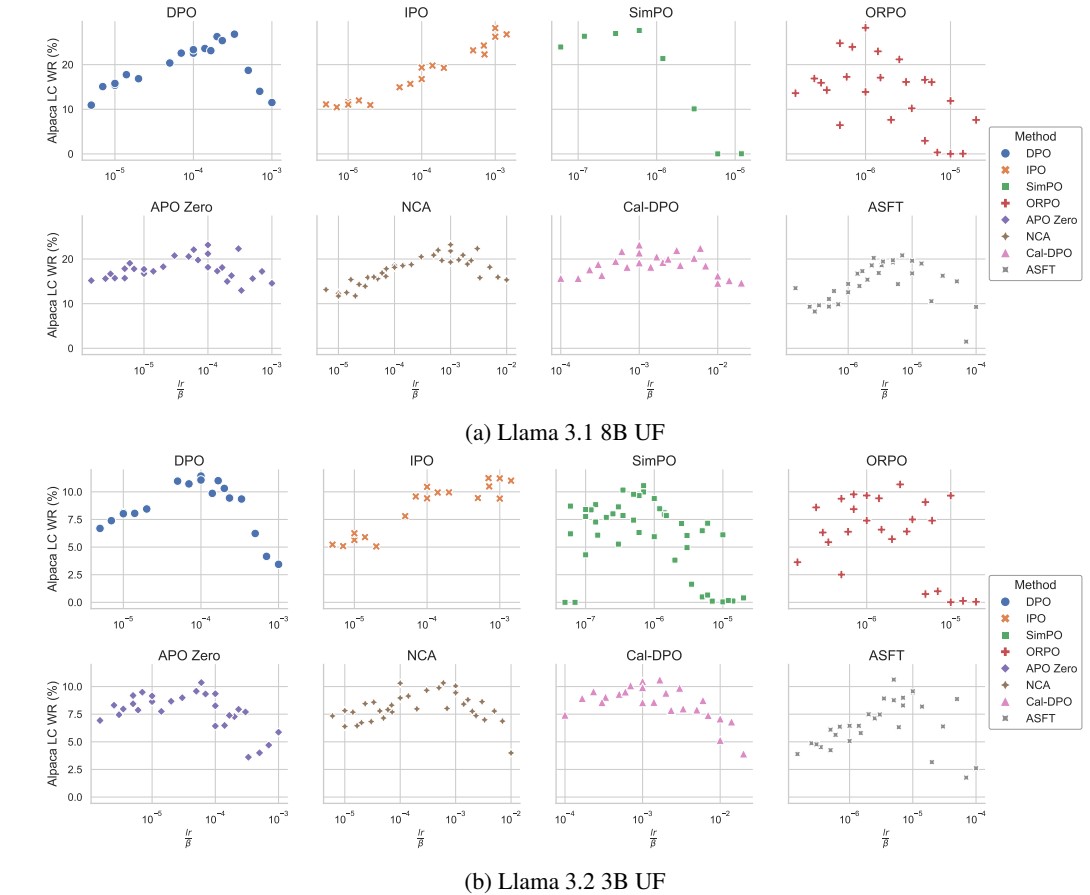

(a) Llama 3.1 8B UF

(b) Llama 3.2 3B UF

Figure 9: **Alignment quality versus $\frac{\text{lr}}{\beta}$.** Each point is a single run from the grid described in Section 4.2. The $x$-axis shows the ratio $\text{lr}/\beta$; the $y$-axis is AlpacaEval 2 LC WR.

ity degradation due to the presence of a reference policy, but this does not affect the peak quality achievable by each method (see Table 3).

## J   INTRACLASS CORRELATION COEFFICIENT (ICC$_1$) IN THE TOY EXPERIMENT

The Intraclass Correlation Coefficient (ICC$_1$) Bartko (1966); Shrout & Fleiss (1979) is a statistical measure used to quantify how much of the total variance in a set of observations is attributable to differences between groups (here, values of the context variable $x$), as opposed to random variation within each group (here, pairs of candidate scores for the same $x$).

**Purpose in Our Setting.**   In our toy experiment, the goal is to assess the extent to which the model's learned scoring function $r_\theta(x, r)$ exhibits prompt-specific bias: that is, systematic differences in the average score assigned to different contexts $x$, independent of differences between candidate completions for the same $x$.

**Mathematical Formulation.**   Given that for each value of $x$ we have two completions with model scores $r_\theta(x, r_w)$ and $r_\theta(x, r_l)$, we define the prompt-specific baseline as the average score for $x$:

$$\hat{b}(x) = \frac{r_\theta(x, r_w) + r_\theta(x, r_l)}{2}.$$

We are interested in the variance of $\hat{b}(x)$ across contexts, $\mathrm{Var}_x[\hat{b}(x)]$, which captures how much the model's scores "shift" between different values of $x$. The total variance in the model's scores is $\mathrm{Var}_{x,y}[r_\theta(x,r)]$, computed over all context-candidate pairs.

For the case of $k = 2$ candidates per context, the $\mathrm{ICC}_1$ is given by:

$$\mathrm{ICC}_1 = 2 \cdot \frac{\mathrm{Var}_x\left[\hat{b}(x)\right]}{\mathrm{Var}_{x,y}\left[r_\theta(x,r)\right]} - 1$$

This is a standard algebraic form of the one-way random effects ICC estimator for the case of $k = 2$ repeated measurements per group, as detailed in Shrout & Fleiss (1979); McGraw & Wong (1996); Searle et al. (2009).

**Interpretation.** $\mathrm{ICC}_1 \approx -1$: Virtually all variance is within each context (i.e., between the two candidate scores for the same $x$), and the model assigns no systematic bias per context. In our unbiased data condition, where the true input baseline is zero, a well-trained model should yield $\mathrm{ICC}_1$ close to $-1$.

$\mathrm{ICC}_1 \approx 0$: About half the variance is due to differences between contexts, and half is within contexts.

$\mathrm{ICC}_1 \to 1$: Most of the variance is between contexts, i.e., the model's output scores strongly reflect context-specific bias.

**Connection to Data Generation and Model Behavior.** In our experiment, the input scores to the model are centered so that (in the absence of injected bias) the true baseline for each context $x$ is zero. When a context bias is present in the data (nonzero $b_x$), a model that captures this bias will have $\mathrm{Var}_x[\hat{b}(x)] > 0$, yielding a higher $\mathrm{ICC}_1$. If the learning objective (e.g., pointwise) suppresses or removes this context bias, $\mathrm{Var}_x[\hat{b}(x)]$ will decrease, and $\mathrm{ICC}_1$ will approach $-1$.

Conversely, pairwise objectives, which focus only on differences between candidates for the same context, do not penalize nor remove such baseline shifts, and thus tend to preserve the bias structure of the data.

Thus, $\mathrm{ICC}_1$ is a direct measure of whether the model's learned scores have inherited context-specific bias (structure) from the training data, or have been actively normalized to remove such bias. This distinction is crucial for demonstrating how pairwise and pointwise objectives interact differently with data-induced biases in our toy experiment.

# K GPT-4 SIDE-BY-SIDE EVALUATION PROMPT

For our Side-By-Side evaluations with `GPT-4o`, we designed a prompt tailored to the Reddit TL;DR dataset to assess *accuracy*, *completeness*, *relevance*, and *conciseness*. The full prompt used in our experiments is detailed below.

---

```
Act as an impartial judge and evaluate the quality of the summaries provided
by two AI assistants for the text displayed below. Your evaluation should
consider accuracy, completeness, relevance, and conciseness.

You will be given a text, Assistant A's summary, and Assistant B's summary.
Your job is to evaluate which assistant's summary is better based on the
text provided.

Begin your evaluation by comparing both assistants' summaries with the
original text. Identify and correct any inaccuracies.
Ensure the summaries are complete, capturing all essential information
from the text without introducing fabricated details.
Assess the relevance of the information each assistant chose to include
```

in their summary, ensuring it reflects the core message of the text.
Evaluate the conciseness of the summaries, favoring those that efficiently
convey the necessary information without unnecessary verbosity.
Avoid any position biases and ensure the order in which the summaries
were presented does not influence your decision.
Do not allow the length of the summaries to influence your evaluation,
except in the context of conciseness and efficiency.
Do not favor certain names of the assistants.
Be as objective as possible.
You should only evaluate the summaries provided by both assistants
and NOT the original text itself.
If both summaries are irrelevant, contain hallucinations, or are
inconsistent with the original text, mark the comparison as inconclusive
and choose option "C".

After providing your explanation, output your final verdict by strictly
following this format:

"""
Comparison: <One-sentence comparison>
Winner: <A if assistant A is better, B if assistant B is better, and C for a tie.>
"""

