# OpenReview forum: "The Differences Between Direct Alignment Algorithms are a Blur"
_ICLR.cc/2026/Conference — Submitted to ICLR 2026_

### Official Review · Reviewer_hgKm · 2025-10-25

**Soundness:** 3
**Presentation:** 2
**Contribution:** 2
**Rating:** 2
**Confidence:** 3

**Summary:**

- The paper proposes a **unified training protocol** for Direct Alignment Algorithms (DAAs): (i) convert one-stage odds-ratio methods (**ORPO**, **ASFT**) into a **two-stage** pipeline with an explicit SFT phase, and (ii) introduce a **β tempering parameter** so all methods share a comparable hyperparameter space. In this unified setting, the **ranking objective (pairwise vs. pointwise)**—rather than the scalar score (policy/reference ratio vs. odds ratio)—is argued to be the primary driver of alignment quality.
- Experiments span **Llama‑3.2‑3B (Reddit TL;DR)**, **Llama‑3.1‑8B (UltraChat/UltraFeedback → AlpacaEval‑2 LC & Arena‑Hard)**, and **Qwen‑2.5 (7B/14B) for math‑reasoning**. TL;DR is judged with **GPT‑4o**; UltraFeedback uses **AlpacaEval‑2 LC** and **Arena‑Hard**.
- **β‑tuning** improves odds‑ratio methods.

**Strengths:**

- **Clean unification & fair comparisons.** Converting ORPO/ASFT to two‑stage and adding β places all DAAs in a shared search space, enabling apples‑to‑apples comparisons; the paper details the tempered formulation and common tuning.
- **Transparent, controlled evaluation** across model families/scales and benchmarks.
- **Actionable takeaway.** After unification and tuning, **pairwise vs. pointwise** explains most observed differences; scalar score choice (odds vs. reference‑ratio) is secondary.

**Weaknesses:**

1) **Missing / under‑engaged related work:**
   - **AlphaPO: Reward Shape Matters for LLM Alignment** — introduces an **α‑controlled reward‑shape** family directly relevant to your tempering/unification axis.
   - **Adaptive Reward Margin for Direct Preference Optimization** — proposes **instance‑adaptive reward margins**, closely related to temperature/margin control.
   - **Beyond Reverse KL: Generalizing Direct Preference Optimization with Diverse Divergence Constraints** — demonstrates how **data/reward‑quality shaping** highlights the importance of reward shaping.
   - **RainbowPO: A Unified Framework for Combining Improvements in Preference Optimization**.
   - **Principled Foundations for Preference Optimization** — provides a **general theoretical framing**.

2) **Scope of claims vs. analysis focus.** The paper claims to compare DAAs broadly, yet much of the **conceptual/derivational focus centers on ORPO/ASFT**—e.g., **“3.1 Generalizing ASFT and ORPO”** and **“Tempering ASFT and ORPO”**—within an odds‑based view. There is **limited treatment of explicit reward‑shaping/margin frameworks** and other variants of DAAs including the references mentioned above. Either broaden the discussion to cover these axes, or qualify the generality of the claim.

**Questions:**

1) How robust is the conclusion of “**ranking dominates**”? For example, if you incorporate **reward‑shape (AlphaPO)** or **adaptive margins (AlphaDPO)** or components from **RainbowPO** under your unified protocol, is the conclusion generally hold?

---

> ### Author Response · Authors · 2025-11-14
>
> Thank you for the detailed and thoughtful review. We appreciate that you highlight as strengths exactly the aspects we aimed for: a clean unification, fair comparisons, and the actionable takeaway that pairwise vs. pointwise explains most of the empirical differences once methods are put on equal footing. Below we address your concerns and clarify how the cited works fit relative to our scope and claims.
>
> **W1: Engagement with Related Work**
>
> **1. First, a factual clarification**
>
> The works f-DPO and RainbowPO you list are **already cited and briefly discussed** in our current Related Work section (lines 481-482).
>
> We agree that the discussion can be strengthened, and in the revision we will (i) make this connection more explicit and (ii) move the Related Work section earlier so that these links are easier to spot. But the criticism that we entirely "missed" this line of work is not accurate.
>
> **2. On timelines and ICLR policy**
>
> - **AlphaPO** and **AlphaDPO** are ICML 2025 papers whose proceedings were officially released after our ICLR submission deadline (PMLR volume dated in October, 6).
> - **Principled Foundations for Preference Optimization** is, at the time of submission, arXiv-only.
>
>     According to the ICLR FAQ (quoted by the program chairs), authors are **not required** to compare against (i) contemporaneous papers published within two months of the deadline, or (ii) works that only exist on arXiv; the lack of such comparisons “cannot be a basis for rejection”.
>
>     We want to stress this not to "hide behind rules", but to make clear for the ACs that our omission is a matter of timing, not a conceptual gap in the work. Now that you have pointed us to these papers, we are happy to incorporate them to more fully reflect the state of the research field.
>
>
> **3. How these works sit relative to our contributions**
> All of the methods you mention are **squarely within the pairwise DPO-style family** and mainly explore the **reward-shape / margin / divergence axis**, whereas our main contribution is about a more coarse-grained structural axis: **pairwise vs. pointwise ranking**, and about unifying **odds-ratio and reference-ratio** DAAs under a single two-stage, $\beta$-tempered protocol.
>
> Concretely:
>
> - **f-DPO**
>
>     Generalizes DPO by replacing the **reverse-KL regularizer** with a family of f-divergences, trading off alignment and diversity. The Bradley–Terry structure and the **pairwise logistic loss over a scalar score difference** are preserved; only the mapping from policy probabilities to that scalar reward is changed.
>
>     → In our terminology this is a move along the **“scalar score ($r$) / divergence” axis** *within* the pairwise family. Our experiments already show that once $\beta$ and the training protocol are unified, changes along this axis (DPO vs. SimPO vs. Cal-DPO vs. APO-Zero) lead to **second-order differences** compared to switching from pointwise to pairwise.
>
> - **RainbowPO**
>
>     Provides a very nice decomposition of *DPO-style pairwise methods* into orthogonal components (length normalization, margins, reference mixing, contextual scaling, etc.) and then recombines them. Again, everything happens **inside the DPO-like pairwise class.**
>
>     → Our work operates at a different layer: we fix a **common SFT→alignment pipeline and $\beta$-tempering** across DAAs, we investigate which structural axis has the greatest impact (RQ3). RainbowPO refines “what matters **inside** the DPO-style pairwise family”; our paper explains “what matters **between** families once pipelines and **$\beta$** are unified”. These are complementary.
>
> - **AlphaPO (reward-shape) and AlphaDPO (adaptive margins)**
>
>     Both methods **stay pairwise** and keep the Bradley–Terry structure:
>
>     - AlphaPO changes the **reward shape** $r_\alpha$ (via $\alpha$-divergence inspired scoring plus length-normalization) but still optimizes a pairwise logistic in $r_\alpha(x, y_w)-r_\alpha(x, y_l)$.
>     - AlphaDPO reparameterizes the **reference distribution** to induce instance-adaptive margins, again yielding a pairwise logistic loss in a shaped reward difference.
>
>     In our taxonomy, these are precisely examples of “moving along the **reward-shape / margin** axis while staying in the **pairwise column**”. This is conceptually very close to what we do when we introduce a true $\beta$-temperature for ORPO/ASFT: $\beta$ controls gradient scale and aggressiveness but does *not* change the fact that ORPO/ASFT are pairwise (once converted to two-stage).
>
>     We will explicitly add AlphaPO and AlphaDPO to Related Work and mention them in Section 6 as examples of **fine-grained reward shaping inside the pairwise DAA family**, orthogonal to our main pairwise vs. pointwise story.

---

> > ### Author Response · Authors · 2025-11-14
> >
> > - **Principled Foundations for Preference Optimization**
> >
> >     This paper gives a highly general theoretical “umbrella” that connects DPO to the theory of proper losses and stochastic choice, showing that many DPO variants (with margins, length corrections, etc.) live in a small region of a much larger design space.
> >
> >     Our work is intentionally **much narrower and more concrete**: we take a practically dominant regime (offline SFT→alignment on binary preference pairs) and:
> >
> >     1. put **odds-based one-stage methods (ORPO/ASFT)** into the **same two-stage,** $\beta$**-tempered protocol** as DPO-style methods;
> >     2. show empirically and mechanistically that **pairwise vs. pointwise** ranking is the main performance driver under that unified setup.
> >
> >     We will cite Principled Foundations explicitly and clarify that our contribution can be viewed as a *case study* within their general map: we pick one sub-regime and give a clean mechanistic explanation and controlled evidence for one structural axis (ranking type).
> >
> >     **Summarizing W1:** some of the cited works are already in the paper; others are either contemporaneous ICML proceedings or arXiv-only, so our omission is not grounds for rejection per ICLR policy. More importantly, all of them act **inside** the pairwise DPO-style family by reshaping rewards or margins, and are therefore **orthogonal** to our main axis of interest. In the revision we will expand Related Work to explicitly make this separation clear.
> >
> > **W2. “Scope of claims vs. analysis focus on ORPO/ASFT and odds-based view”**
> >
> > The motivation for the structure of Section 3 is primarily methodological rather than conceptual. Among all DAAs we evaluate, ORPO and ASFT are the ones that (i) were originally designed as one-stage SFT+alignment objectives and (ii) lacked $\beta$ precisely because they do not use a reference policy. Therefore, placing *all* DAAs into a common search space requires additional derivations **specifically for these two methods**: (i) showing how the explicit SFT term embedded in their joint loss can be separated into its own stage, and (ii) introducing a proper $\beta$-temperature on the odds and analyzing that it plays a role similar to that in reference-based pairwise methods (Appendix E). This derivation is what Section 3 focuses on and all this enabling them to reach their peak performance.
> >
> > After that, however, our **conceptual framework and empirical analysis are not odds-specific**:
> >
> > - The **pairwise/pointwise distinction** in Section 6 is defined for a scalar score $r\_\theta(x,y)$, independent of how this score is parameterized (odds, policy–reference ratio, shaped reward, etc.) and is not tied to odds or to any particular DAA.
> > - The **capacity–bias mechanism** (via the marginalized score $\mathbb{E}\_y[r\_\theta(x,y)]$) applies symmetrically to DPO, IPO, SimPO, Cal-DPO, APO-Zero, ORPO, ASFT and NCA.
> > - Our experiments in Tables 1–3 and Appendix F explicitly cover all of these methods; ORPO/ASFT are just one part of that picture.
> >
> > Regarding frameworks like RainbowPO, AlphaPO and AlphaDPO that emphasize **reward-shaping and margins:** we see them as highly complementary but deliberately out-of-scope for this paper. They explore which *components* inside the pairwise family improve performance, our goal is to isolate which *family* (pairwise vs. pointwise) dominates once pipelines and $\beta$ are controlled.

---

> > > ### Author Response · Authors · 2025-11-14
> > >
> > > **Q1. Robustness of “ranking dominates” under reward-shape and adaptive margins**
> > >
> > > Our mechanistic explanation in Section 6 is built at the level of a **single scalar score function $r\_\theta(x,y)$** and makes only two structural assumptions:
> > >
> > > **Pairwise objectives** depend on scores **only through differences**, e.g.
> > >
> > > $$
> > > L = -\log\sigma\big(m(x,y_w,y_l)[g(r_\theta(x,y_w)) - g(r_\theta(x,y_l))]\big).
> > > $$
> > >
> > > As long as:
> > >
> > > - the method remains **pairwise** (depends only on $g(r_\theta(x,y_w)) - g(r_\theta(x,y_l))$, and
> > > - $g$ is monotone (so it preserves the ordering of $y$ for each $x$),
> > >
> > > all of our invariance and capacity–bias arguments still apply:
> > >
> > > - Pairwise methods with sophisticated reward shapes/margins **still do not try to erase prompt-specific bias**; they only care about relative differences.
> > > - Pointwise methods with similar shaping **still have to adjust each score individually**, and thus still pay the bias-removal cost in the medium-difficulty, capacity-limited regime.
> > >
> > > This is exactly what we see empirically in our unified setup:
> > >
> > > - Within the **pairwise** family, quite different scalar choices and calibrations (DPO, SimPO, Cal-DPO, APO-Zero, $\beta$-tempered ORPO/ASFT) form a relatively tight performance band once tuned fairly.
> > > - Within the **pointwise** family, behavior is consistently and *qualitatively* different, matching the capacity–bias mechanism.
> > >
> > > Therefore, our prediction is:
> > >
> > > - Incorporating **AlphaPO, AlphaDPO, RainbowPO-style components** under our unified SFT→alignment protocol would very likely **keep them in the “pairwise cluster”**—potentially raising the whole cluster’s level, but not changing the fact that **pairwise methods outperform pointwise ones in the medium-difficulty, capacity-limited regime** we study.
> > > - Symmetric shaping for pointwise methods would not remove their structural tendency to expend capacity on bias removal.
> > >
> > > We will tighten the statement of our main claim to read:
> > >
> > > > In the offline SFT→alignment regime on binary preference data, once one-stage methods are converted to a two-stage, $\beta$-tempered framework, the pairwise vs. pointwise ranking type is the dominant structural factor behind performance differences, while the choice of scalar reward shape (odds vs. policy–reference ratio vs. shaped DPO-style variants) is secondary.
> > > >
> > >
> > > We view the systematic integration of RainbowPO/AlphaPO/AlphaDPO-style components into our framework as **valuable future work**, but our current theory and experiments strongly suggest that the “ranking dominates” conclusion will remain robust under such extensions.
> > >
> > > Again, thank you for the careful review and for pointing us to these additional works. With the clarified scope, stronger Related Work section, and explicit connection to reward-shaping frameworks, we believe the paper’s core contribution, identifying **ranking type** as the primary performance driver under a unified DAA protocol, remains solid and broadly useful for the community. If there are remaining concerns that would prevent a higher score, we’d be grateful to address them during the discussion.

---

> > ### Comment · Reviewer_hgKm · 2025-11-25
> >
> > Thanks for the clearification and the additional coverage of the related works.

---

> > > ### Author Response · Authors · 2025-11-25
> > >
> > > Thank you again for carefully reading our rebuttal and for your note about the improved related-work discussion; we really appreciate your time and input. In the revision, we have already updated the Related Work section and clarified the scope of our main claim along the lines you suggested. From your original review, our understanding was that the main issues were (i) incomplete or under-engaged coverage of concurrent work, and (ii) a perceived mismatch between the breadth of our claims and the fact that much of the conceptual/derivational focus in Section 3 is on ORPO/ASFT within an odds-based view.
> > >
> > > We are a bit frustrated by not knowing whether we have actually resolved the concerns you raised in the weaknesses. Since the discussion phase is still ongoing, could you please let us know whether our response has addressed these weaknesses, or whether there is something that prevents you from increasing your score?
> > >
> > > We sincerely appreciate your efforts and the time you have invested in helping us improve the paper.

---

> > > > ### Comment · Reviewer_hgKm · 2025-11-25
> > > >
> > > > The claim that “the conceptual framework and empirical analysis are not odds-specific” is my main concern, given that the experiments are restricted to $r_{\text{ref}}$ and $r_{\text{odds}}$ under a particular parametric form. It is therefore unclear to what extent the conclusions about pointwise vs.\ rankwise behavior continue to hold when $r_{\text{ref}}$ and $r_{\text{odds}}$ depart from this parametric family—for example, when using divergences arising in f-DPO or AlphaPO.

---

> > > > > ### Author Response · Authors · 2025-11-28
> > > > >
> > > > > Thank you very much for your precise follow-up and for insisting on this point. When we wrote in our previous response that “the conceptual framework and empirical analysis are not odds-specific”, what we had in mind was the part of the paper *after* Section 3, where we move from the odds-ratio derivations for ORPO/ASFT to a formulation in terms of an abstract scalar score $r_\theta(x,y)$. We agree that this is an important point to verify, however, that this needed to be demonstrated more directly by going beyond the original $r_{\text{ref}}$ and $r_{\text{odds}}$ families. To address this head-on, we have now added an explicit experiment with a different parametric reward family taken from AlphaPO.
> > > > >
> > > > > Concretely, we take the AlphaPO reward
> > > > >
> > > > > $$
> > > > > r_\alpha(y;x) = \frac{\beta}{\alpha}\Bigl( 1 - \pi_\theta(y|x)^{-\alpha/|y|} \Bigr)
> > > > > $$
> > > > >
> > > > > exactly as defined in the AlphaPO paper, and plug it into two different objectives:
> > > > >
> > > > > $$
> > > > >   L_{\text{pair}} = -\log \sigma\big(r_\alpha(y_w;x) - r_\alpha(y_l;x)\big);
> > > > > $$
> > > > >
> > > > > $$
> > > > >   L_{\text{point}} = -\log \sigma\big(r_\alpha(y_w;x)\big)-\log \sigma\big(-r_\alpha(y_l;x)\big).
> > > > > $$
> > > > >
> > > > > In both cases the scalar score $r_\alpha$ is identical; the only difference is how it enters the loss: via a difference (pairwise Bradley–Terry) or via separate terms for the preferred and dispreferred responses, analogous to ASFT. This is precisely the scenario you asked about: we move to a different reward family $r_\alpha$, not tied to $r_{\text{ref}}$ or $r_{\text{odds}}$, and ask whether the structural pairwise vs. pointwise behavior persists.
> > > > >
> > > > > In these experiments we do not use the $\gamma$ margin parameter from SimPO/AlphaPO. Our aim here is not to exhaustively re-tune the full $(\alpha,\beta,\gamma)$-cube, but to isolate the effect of changing the score family itself. The parameters $\alpha$ and $\beta$ define exactly this family: $\alpha$ controls the reward shape, and $\beta$ plays the role of a temperature on that shaped reward. By contrast, $\gamma$ is an additive margin applied *after* the scalar score has been defined; it is not part of the $r_\alpha$-family itself. For this additional experiment, we therefore deliberately vary $\alpha$ and $\beta$, which directly addresses your question about robustness to the score parameterization, while keeping the search space manageable. Even with two parameters the grid is already fairly dense; adding a third axis would turn this add-on into a separate, much larger study. Our goal here is to provide the minimal but meaningful extension that directly speaks to your concern.
> > > > >
> > > > > We run these AlphaPO-style pairwise and pointwise variants in the same Qwen2.5–7B and 14B Math-CoT setups we use in the paper, precisely to avoid any GPT-judge bias. We use a learning rate of 3e-07. This choice is motivated by the fact that AlphaPO generalizes SimPO, and in this math reasoning setup 3e-07 is the learning rate that yielded the best SimPO results; larger or lower learning rates led to clear degradation, so we keep this safe value for a fair and stable comparison.
> > > > >
> > > > > On Qwen2.5–7B, we perform a grid search over $\beta \in \{0.1, 0.5, 0.7, 1.0, 2.5, 5.0, 10.0, 20.0, 25.0\}$ and $\alpha \in \{-0.5, -0.1, 0.0, 0.1, 0.25, 0.5, 1.0, 2.0\}$. For the best $\beta$ values $(0.5, 0.7, 1.0)$, we further sweep $\alpha \in \{2.5, 3.0, 5.0\}$ to make sure we have a good picture of the optimum for both the pairwise and pointwise AlphaPO variants. On Qwen2.5–14B, we use the same learning rate and reuse the three best $\beta$ values from the 7B setup, $\beta \in \{0.5, 0.7, 1.0\}$, sweep $\alpha \in \{0.5, 1.0, 1.5, 2.0, 2.5\}$, again for both pairwise and pointwise formulations.
> > > > >
> > > > > The resulting tables show a consistent pattern on both model scales. On Qwen2.5–7B, the **AlphaPO-Pair** variant sits very close to the other pairwise methods (DPO, IPO, SimPO, ORPO), while **AlphaPO-Point** tracks the pointwise cluster and is slightly weaker than its pairwise counterpart. On Qwen2.5–14B, we see the same behavior: AlphaPO-Pair achieves performance in the same range as the other pairwise DAAs and, like them, tends to outperform its pointwise analogue across metrics. In other words, once we plug the AlphaPO reward $r_\alpha$ into our unified training setup, it behaves exactly like another member of the **pairwise family**, and the **pairwise vs. pointwise gap** remains visible inside this new score family as well.

---

> > > > > > ### Author Response · Authors · 2025-11-28
> > > > > >
> > > > > > **Qwen2.5-7B Math-CoT (Mean Avg@K with std, 4 seeds)**
> > > > > >
> > > > > > |  | **Mean Avg@K (std)** | **GSM8K avg@8 (std)** | **MATH500 avg@8 (std)** | **AMC23 avg@8 (std)** | **Minerva avg@8 (std)** | **AIME24 avg@32 (std)** | **AIME25 avg@32 (std)** |
> > > > > > | --- | --- | --- | --- | --- | --- | --- | --- |
> > > > > > | Base | 0.1344 (0.0024) | 0.2233 (0.0012) | 0.2876 (0.0067) | 0.1805 (0.0248) | 0.0679 (0.0045) | 0.0292 (0.0035) | 0.0177 (0.0043) |
> > > > > > | SFT | 0.2216 (0.0019) | 0.6677 (0.0012) | 0.3671 (0.0026) | 0.1711 (0.0143) | 0.1135 (0.0055) | 0.0076 (0.0016) | 0.0029 (0.0010) |
> > > > > > |  |  |  |  |  |  |  |  |
> > > > > > | DPO | 0.3017 (0.0033) | 0.8295 (0.0015) | 0.5011 (0.0080) | 0.2641 (0.0174) | 0.1966 (0.0050) | 0.0107 (0.0033) | 0.0081 (0.0025) |
> > > > > > | IPO | 0.2996 (0.0035) | 0.8236 (0.0020) | 0.4866 (0.0055) | 0.2664 (0.0121) | 0.2007 (0.0027) | 0.0128 (0.0035) | 0.0078 (0.0013) |
> > > > > > | SimPO | 0.2892 (0.0039) | 0.7893 (0.0026) | 0.4837 (0.0037) | 0.2523 (0.0190) | 0.1898 (0.0014) | 0.0125 (0.0037) | 0.0073 (0.0026) |
> > > > > > | ORPO | 0.2966 (0.0010) | 0.8303 (0.0026) | 0.4882 (0.0046) | 0.2641 (0.0116) | 0.1759 (0.0047) | 0.0117 (0.0064) | 0.0094 (0.0039) |
> > > > > > | **AlphaPO Pair** | 0.2945 (0.0020) | 0.8147 (0.0014) | 0.4872 (0.0027) | 0.2578 (0.0141) | 0.1849 (0.0010) | 0.0143 (0.0021) | 0.0083 (0.0015) |
> > > > > > |  |  |  |  |  |  |  |  |
> > > > > > | APO Zero | 0.2837 (0.0020) | 0.8071 (0.0030) | 0.4586 (0.0050) | 0.2352 (0.0069) | 0.1807 (0.0102) | 0.0115 (0.0031) | 0.0091 (0.0034) |
> > > > > > | NCA | 0.2861 (0.0027) | 0.7909 (0.0015) | 0.4715 (0.0026) | 0.2461 (0.0145) | 0.1922 (0.0017) | 0.0109 (0.0051) | 0.0047 (0.0010) |
> > > > > > | Cal-DPO | 0.2936 (0.0044) | 0.8272 (0.0014) | 0.4694 (0.0053) | 0.2531 (0.0209) | 0.1931 (0.0070) | 0.0109 (0.0028) | 0.0081 (0.0018) |
> > > > > > | ASFT | 0.2903 (0.0033) | 0.8132 (0.0020) | 0.4785 (0.0064) | 0.2437 (0.0149) | 0.1876 (0.0009) | 0.0130 (0.0028) | 0.0057 (0.0013) |
> > > > > > | **AlphaPO Point** | 0.2922 (0.0027) | 0.8233 (0.0012) | 0.4753 (0.0085) | 0.2539 (0.0074) | 0.1820 (0.0074) | 0.0122 (0.0021) | 0.0063 (0.0009) |
> > > > > >
> > > > > > **Qwen2.5-14B Math-CoT (Mean Avg@K with std, 4 seeds)**
> > > > > >
> > > > > > |  | **Mean Avg@K (std)** | **GSM8K avg@8 (std)** | **MATH500 avg@8 (std)** | **AMC23 avg@8 (std)** | **Minerva avg@8 (std)** | **AIME24 avg@32 (std)** | **AIME25 avg@32 (std)** |
> > > > > > | --- | --- | --- | --- | --- | --- | --- | --- |
> > > > > > | Base | 0.1937 (0.0065) | 0.5343 (0.0064) | 0.3177 (0.0048) | 0.1773 (0.0259) | 0.1002 (0.0069) | 0.0188 (0.0043) | 0.0138 (0.0054) |
> > > > > > | SFT | 0.2399 (0.0013) | 0.7162 (0.0018) | 0.4006 (0.0105) | 0.1719 (0.0186) | 0.1351 (0.0052) | 0.0115 (0.0049) | 0.0039 (0.0013) |
> > > > > > | DPO | 0.3202 (0.0030) | 0.8509 (0.0013) | 0.5279 (0.0080) | 0.2773 (0.0166) | 0.2341 (0.0085) | 0.0221 (0.0013) | 0.0089 (0.0045) |
> > > > > > | IPO | 0.3146 (0.0014) | 0.8584 (0.0009) | 0.5150 (0.0031) | 0.2594 (0.0068) | 0.2336 (0.0067) | 0.0164 (0.0040) | 0.0047 (0.0018) |
> > > > > > | SimPO | 0.3148 (0.0008) | 0.8585 (0.0035) | 0.5446 (0.0039) | 0.2773 (0.0053) | 0.1823 (0.0015) | 0.0161 (0.0047) | 0.0099 (0.0036) |
> > > > > > | ORPO | 0.3277 (0.0066) | 0.8690 (0.0009) | 0.5503 (0.0049) | 0.2883 (0.0273) | 0.2232 (0.0062) | 0.0219 (0.0050) | 0.0135 (0.0054) |
> > > > > > | **AlphaPO Pair** | 0.3158 (0.0019) | 0.8693 (0.0010) | 0.5277 (0.0128) | 0.2695 (0.0064) | 0.1930 (0.0034) | 0.0234 (0.0025) | 0.0117 (0.0037) |
> > > > > > |  |  |  |  |  |  |  |  |
> > > > > > | APO Zero | 0.3081 (0.0012) | 0.8717 (0.0018) | 0.5052 (0.0028) | 0.2461 (0.0053) | 0.1965 (0.0069) | 0.0211 (0.0032) | 0.0078 (0.0044) |
> > > > > > | NCA | 0.2979 (0.0031) | 0.8340 (0.0034) | 0.5058 (0.0052) | 0.2250 (0.0238) | 0.1983 (0.0071) | 0.0195 (0.0021) | 0.0049 (0.0016) |
> > > > > > | Cal-DPO | 0.2943 (0.0013) | 0.8334 (0.0008) | 0.4946 (0.0065) | 0.2289 (0.0097) | 0.1916 (0.0059) | 0.0148 (0.0027) | 0.0026 (0.0010) |
> > > > > > | ASFT | 0.3030 (0.0042) | 0.8330 (0.0010) | 0.5004 (0.0040) | 0.2492 (0.0154) | 0.2075 (0.0051) | 0.0216 (0.0090) | 0.0060 (0.0031) |
> > > > > > | **AlphaPO Point** | 0.2923 (0.0018) | 0.8172 (0.0017) | 0.5032 (0.0079) | 0.2336 (0.0208) | 0.1766 (0.0056) | 0.0180 (0.0035) | 0.0049 (0.0021) |
> > > > > >
> > > > > > We will add these AlphaPO-Pair vs. AlphaPO-Point results to Appendix B in the revision version, as they provide a direct and, we believe, quite strong confirmation of our main structural conclusion under a nontrivial change of score parameterization.

---

> > > > > > > ### Author Response · Authors · 2025-11-28
> > > > > > >
> > > > > > > Finally, we would like to note that our controlled prompt-bias experiment (Appendix F) is explicitly formulated at the level of an abstract scalar score $r_\theta(x,y)$, without committing to any particular parametric form. In that synthetic setting we construct data with prompt-dependent bias in the marginal $\mathbb{E}\_y[r\_\theta(x,y)]$ and compare pointwise and pairwise objectives under identical conditions and limited capacity. Because the pairwise loss depends only on differences $r\_\theta(x,y_w)-r\_\theta(x,y_l)$, it is invariant to adding an arbitrary per-prompt shift $b_x$ and therefore does not “see” the prompt bias; the pointwise loss is not invariant and must spend part of its finite capacity on cancelling this bias. Together with the new AlphaPO-based experiment, this suggests that in the regime we actually study, offline SFT->alignment on binary preference data, the structural difference between pairwise and pointwise objectives remains the main driver, while the specific choice of scalar score family plays a secondary role. We will make this intended scope more explicit in the revised version. We are also currently evaluating the Forward-KL variant from f-DPO under the same unified protocol and will include these results in the revision as soon as they are ready. We are grateful for your question, which prompted us to perform this additional AlphaPO-based study and to refine and better ground our claims.

---

> > > > > > > > ### Author Response · Authors · 2025-11-30
> > > > > > > >
> > > > > > > > As a follow-up to our previous reply, we have now completed the experiments with the Forward-KL (fKL) variant from f-DPO under the same unified protocol.
> > > > > > > >
> > > > > > > > To probe your concern within the f-DPO family, we instantiate the scalar score as
> > > > > > > >
> > > > > > > > $$
> > > > > > > > r^{\text{fKL}}\_\theta(x,y) = -\beta\frac{\pi\_{\text{ref}}(y\mid x)}{\pi\_\theta(y\mid x)},
> > > > > > > > $$
> > > > > > > >
> > > > > > > > which corresponds to the Forward-KL case in f-DPO (the $\alpha=1$ case of the $\alpha$-divergence family). In f-DPO, the limit $\alpha\to 0$ recovers the standard DPO / reverse-KL objective, while $\alpha\to 1$ yields the Forward-KL objective. Since the f-DPO paper already shows (Table 2, Fig. 3) that the reverse-KL/DPO setting achieves the strongest task performance within this family, an $\alpha$-sweep would mostly rediscover behavior very close to DPO, which is already part of our study. Instead, we focus on the Forward-KL setting, which lies on the opposite side of the reward–KL trade-off and yields weaker task performance, and ask whether the pairwise vs. pointwise gap persists even in this less favorable regime.
> > > > > > > >
> > > > > > > > As before, we keep the scalar score fixed and only vary how it enters the loss:
> > > > > > > >
> > > > > > > > $$
> > > > > > > > L\_{\text{pair}} = -\log \sigma\big(r^{\text{fKL}}\_\theta(x,y_w) - r^{\text{fKL}}\_\theta(x,y_l)\big),
> > > > > > > > $$
> > > > > > > >
> > > > > > > > $$
> > > > > > > > L\_{\text{point}} = -\log\sigma\big(r^{\text{fKL}}\_\theta(x,y_w)\big) - \log\sigma\big(-r^{\text{fKL}}\_\theta(x,y_l)\big).
> > > > > > > > $$
> > > > > > > >
> > > > > > > > We run these fKL-pair and fKL-point variants in the same Qwen2.5–7B/14B Math-CoT setup. On Qwen2.5–7B, we tune learning rate (3e-07, 5e-07) and $\beta \in \{0.001, 0.005, 0.01, 0.05, 0.1, 0.2\}$ for each formulation. On Qwen2.5–14B, we reuse the best learning rate from 7B (5e-07) and sweep the same $\beta$ values.
> > > > > > > >
> > > > > > > > **Qwen2.5-7B Math-CoT (Mean Avg@K with std, 4 seeds)**
> > > > > > > >
> > > > > > > > |  | **Mean Avg@K (std)** | **GSM8K avg@8 (std)** | **MATH500 avg@8 (std)** | **AMC23 avg@8 (std)** | **Minerva avg@8 (std)** | **AIME24 avg@32 (std)** | **AIME25 avg@32 (std)** |
> > > > > > > > | --- | --- | --- | --- | --- | --- | --- | --- |
> > > > > > > > | fKL pair | **0.2673** (0.0041) | 0.7825 (0.0031) | 0.4403 (0.0059) | 0.2141 (0.0195) | 0.1535 (0.0067) | 0.0073 (0.0017) | 0.0060 (0.0010) |
> > > > > > > > | fKL point | 0.2643 (0.0020) | 0.7750 (0.0014) | 0.4363 (0.0032) | 0.2094 (0.0068) | 0.1521 (0.0033) | 0.0078 (0.0028) | 0.0052 (0.0019) |
> > > > > > > >
> > > > > > > > **Qwen2.5-14B Math-CoT (Mean Avg@K with std, 4 seeds)**
> > > > > > > >
> > > > > > > > |  | **Mean Avg@K (std)** | **GSM8K avg@8 (std)** | **MATH500 avg@8 (std)** | **AMC23 avg@8 (std)** | **Minerva avg@8 (std)** | **AIME24 avg@32 (std)** | **AIME25 avg@32 (std)** |
> > > > > > > > | --- | --- | --- | --- | --- | --- | --- | --- |
> > > > > > > > | fKL pair | **0.2839** (0.0029) | 0.8215 (0.0012) | 0.4724 (0.0027) | 0.2125 (0.0133) | 0.1759 (0.0064) | 0.0169 (0.0032) | 0.0039 (0.0029) |
> > > > > > > > | fKL point | 0.2750 (0.0036) | 0.8125 (0.0017) | 0.4663 (0.0017) | 0.1852 (0.0121) | 0.1638 (0.0067) | 0.0164 (0.0005) | 0.0055 (0.0010) |
> > > > > > > >
> > > > > > > > The full tables will be added to the appendix. As expected from f-DPO, the overall performance of the fKL variants is noticeably lower than that of DPO/SimPO/ORPO. However, the **structural pattern is the same as in all other score families we tested**: the pairwise fKL objective consistently outperforms its pointwise analogue, and this gap becomes more pronounced at 14B scale.

---

> > > > > > > > > ### Author Response · Authors · 2025-11-30
> > > > > > > > >
> > > > > > > > > In parallel with these additional experiments, we have also added a short theoretical appendix, *“Theoretical Analysis of Prompt-Specific Bias and Ranking Objectives”*. There we formalize the “prompt bias” mechanism at the level of a general scalar score family
> > > > > > > > >
> > > > > > > > > $$
> > > > > > > > > r_\theta(x,y) = F\big(\pi_\theta(y\mid x), \mathcal{C}(x,y)\big),
> > > > > > > > > $$
> > > > > > > > >
> > > > > > > > > where $\mathcal{C}(x,y)$ may include reference probabilities, sequence length, etc. We define the marginalized score (prompt bias)
> > > > > > > > >
> > > > > > > > > $$
> > > > > > > > > b\_\theta(x) = \mathbb{E}\_{y\sim q_x}[r\_\theta(x,y)]
> > > > > > > > > $$
> > > > > > > > >
> > > > > > > > > for the empirical static offline dataset distribution $q_x$ at prompt $x$, and show that, under mild non-degeneracy conditions, $b_\theta(x)$ is a learnable quantity with non-zero gradient in generic settings.
> > > > > > > > >
> > > > > > > > > We then analyze how different objectives couple to this bias via the **total score gradient**
> > > > > > > > >
> > > > > > > > > $$
> > > > > > > > > G\_\theta(x) = \sum\_{y\in\mathcal{Y}\_x} \frac{\partial L}{\partial r\_\theta(x,y)}.
> > > > > > > > > $$
> > > > > > > > >
> > > > > > > > > This quantity measures the sensitivity of the loss to a uniform shift in all scores for prompt (x), and thus whether the objective creates a first-order incentive to adjust $b_\theta(x)$. For any **pairwise** loss of the form $\ell_{\text{pair}}(r_w,r_l)=\phi(r_w-r_l)$ (DPO, SimPO, ORPO, f-DPO variants, AlphaPO-Pair, etc.), we prove that
> > > > > > > > >
> > > > > > > > > $$
> > > > > > > > >   G_\theta(x)=0 \quad \text{for all } x,
> > > > > > > > > $$
> > > > > > > > >
> > > > > > > > > i.e., the loss is structurally invariant to per-prompt shifts of the scalar scores and does *not* directly push on the prompt bias.
> > > > > > > > > For **pointwise** losses of the form $\ell_{\text{point}}(r_w,r_l)=\psi_+(r_w)+\psi_-(r_l)$ (NCA, Cal-DPO, ASFT, AlphaPO-Point, our fKL-point), we show that, for generic score configurations, $G_\theta(x)\neq 0$, so these objectives do create a gradient signal that encourages changing the prompt bias, competing with the primary ranking task.
> > > > > > > > >
> > > > > > > > > These results are stated for a general $F$ and directly cover the concrete scalar scores we use in our experiments, including $r_{\text{ref}}$, $r_{\text{odds}}$, the AlphaPO family $r_\alpha$, and the fKL score $r^{\text{fKL}}_\theta$. Combined with the new AlphaPO and fKL experiments, this supports our central empirical claim in the regime we actually study (offline SFT->alignment on binary preference data with BT/logistic losses): once pipelines and temperatures are unified, the **pairwise vs. pointwise ranking structure** is the dominant factor, while the specific choice of scalar score family (including reasonable reward-shaping choices such as AlphaPO or f-DPO/fKL) plays a secondary role. We will incorporate these fKL results and the theoretical appendix in the revised version.

---

### Official Review · Reviewer_b2jc · 2025-10-27

**Soundness:** 4
**Presentation:** 4
**Contribution:** 4
**Rating:** 10
**Confidence:** 4

**Summary:**

This work presents comparison and analysis of Direct Alignment Algorithms - a class of methods for aligning LLMs without and explicit reward model. The authors argue that current state of DAA literature is fragmented, which makes it difficult to isolate which design choices drive improvements in alignment quality. To address this, they propose a unified training protocol for DAAs that makes it possible to compare different methods fairly. This protocol introduces two important changes: 1) it converts one-stage methods (ORPO, ASFT) into a two-stage pipeline with an explicit SFT stage, 2) it uses parameter $\beta$ for all tested methods to ensure a common hyperparameter space.

The authors conduct experiments across various models (like Llama or Qwen), sizes (3-14B), and tasks. Their core finding is that the ranking objective (pairwise vs. pointwise) rather than scalar-score choice or heuristic loss design, is the primary determinant of alignment quality.  They provide evidence that observed performance gaps arise from the interactions between each objective and prompt-specific data biases. They also show that most methods are highly data-efficient.

**Strengths:**

- The paper tackles an important area of LLM alignment. Its main claim - that the ranking objective (pairwise vs. pointwise) is the most critical design choice - provides a new dimension for analyzing DAA; it may help clarify conflicting reports of algorithm superiority.
- This work keeps excellent methodological rigor; the authors establish a fair and controlled basis for comparison of DAA methods that was previously missing.
- Empirical evidence is very strong, with experiments spanning multiple model families (Llama, Qwen) , scales (3B to 14B) , and diverse tasks (summarization, instruction-following, and verifiable math reasoning).
- The paper is extremely well-written.
- The authors not only show theoretical results and mathematical proofs, but also practical takeaways for the community.

Overall, I was very impressed by this paper. It definitely deserves the highest score!

**Weaknesses:**

- The wording in the initial motivation in section 3.1.1 could be changed. It suggests that the SFT loss term in one-stage methods may be "redundant because it's encapsulated by the alignment term". However, results in RQ1 (Table 1) and the SFT data ablation (Section 5.4) show that an explicit SFT training stage is important for all methods. Better distingushment between $L_{SFT}$ and SFT training stage may be useful.
- The placement of the related work section is rather unusual. It may be better if this section is before the conclusions section.

**Questions:**

I don't have any specific questions. I would be very happy to see the fixes for the Weaknesses section.

---

> ### Author Response · Authors · 2025-11-14
>
> Thank you for your incredibly positive and encouraging review. We are thrilled that you recognized the importance of our central claim, the methodological rigor of our unified framework, and the strength of our empirical evidence. We deeply appreciate your high praise for the paper's writing and its practical takeaways for the community.
>
> We also thank you for your constructive suggestions for improvement. We completely agree with both points and will implement them in the revised version of our paper.
>
> 1. **Clarifying the motivation in Section 3.1.1:** Thank you for pointing out the potential for confusion between the $L_{SFT}$ loss term used *during alignment* and the dedicated SFT training stage. We will revise the wording in Section 3.1.1 to make this distinction crystal clear. Specifically, we will rephrase our motivation to state that while a separate SFT stage is crucial, including an additional $L_{SFT}$ term during the subsequent alignment phase is what we hypothesize is redundant, as this functionality is already encapsulated by the alignment objective itself. This will better align our initial hypothesis with the findings presented in RQ1 and Section 5.4.
> 2. **Repositioning the Related Work section:** We agree with this point regarding the standard structure of academic papers. We will move the Related Work section to appear before the Conclusion, as you suggested.
>
> Thank you again for your strong support and for helping us further improve the clarity and presentation of our work.

---

> > ### Comment · Reviewer_b2jc · 2025-11-18
> > **Thanks**
> >
> > Thank you for making the suggested changes.

---

### Official Review · Reviewer_u7nZ · 2025-10-28

**Soundness:** 3
**Presentation:** 3
**Contribution:** 2
**Rating:** 2
**Confidence:** 4

**Summary:**

This paper provides a systematic comparison of Direct Alignment Algorithms (DAAs), methods that align large language models (LLMs) without explicit reward modeling or reinforcement learning. The authors unify one-stage and two-stage approaches by introducing a common framework with an explicit Supervised Fine-Tuning (SFT) phase and a temperature parameter (beta), allowing consistent evaluation across methods such as DPO, ORPO, and ASFT. Through extensive experiments on instruction-following and math reasoning benchmarks, they find that the ranking objective (pairwise vs. pointwise), rather than the scalar score type (likelihood vs. odds ratio), is the main determinant of alignment quality. Their results show that once unified and tuned, most DAAs perform similarly, with performance differences largely explained by prompt-specific biases rather than algorithmic superiority.

**Strengths:**

They provide a comprehensive analysis of various state-of-the-art alignment algorithms. They also offer some theory alongside their work to support their hypothesis.

**Weaknesses:**

I believe addressing the following concern might help to improve the quality:

1. The main concern lies in the novelty of the work. The SimPO [1] paper has already demonstrated that ORPO with an SFT stage outperforms the single-stage variant where the backbone is a pre-trained model. This suggests that the improvement attributed to introducing an SFT stage has been previously recognized in existing literature. More clarification about the contribution would be helpful.

2. The recently proposed Triple Preference Optimization (TPO) method has been developed to unify the SFT and preference optimization stages, directly addressing the concern discussed in this paper. Extending the analysis to include TPO-based approaches [2][3] would provide a more comprehensive and up-to-date comparison.

3. Furthermore, incorporating other preference optimization algorithms such as pair-wise KTO [4] and CPO [5] into the analysis could offer deeper insights and a more complete understanding of the differences among preference optimization methods.

---
**References**

[1] SimPO https://arxiv.org/abs/2405.14734

[2] Triple Preference Optimization https://arxiv.org/pdf/2405.16681v2

[3] Tree Preference Optimization https://arxiv.org/abs/2410.12854

[4] KTO https://arxiv.org/abs/2402.01306

[5] CPO https://arxiv.org/abs/2401.08417

**Questions:**

Please read the weaknesses section.

---

> ### Author Response · Authors · 2025-11-14
>
> Thank you for this detailed feedback and the thoughtful references. Below we clarify the relationship between our work and SimPO, TPO/Tree PO, and KTO.
>
> **W1: About ORPO in SimPO paper**
>
> We would like to clarify that SimPO does **not** study the phenomenon we analyze, nor does it perform anything close to our loss-level ablation of ORPO/ASFT.
>
> SimPO treats ORPO purely as a **baseline**, always in its original one-stage form,
>
> $L_{\text{ORPO}} = L_{\text{SFT}} + \lambda L_{\text{ORPO}_{Align}}$,
>
> and trains it from SFT checkpoints “for fairness”. In the appendix, the authors write:
>
> > “ORPO can directly train on preference data without the SFT stage. For fair comparisons, we start ORPO from the same SFT checkpoints as other baselines, which yields better results than starting from base checkpoints.”
> >
>
> Crucially:
>
> 1. **SimPO never compares ORPO with vs. without the SFT term in the loss**.
> They only change the *initial checkpoint* (base vs. SFT), while **keeping the loss unchanged** in both cases.
> There is no analysis of whether the one-stage objective itself is optimal or redundant.
> 2. **SimPO does not quantify their claim**, and no numeric comparison is provided for
>     - ORPO(base init) with $L_{\text{SFT}} + L_{\text{Align}}$,
>     - ORPO(SFT init) with $L_{\text{SFT}} + L_{\text{Align}}$.
>     Their statement is qualitative and is not treated as a research finding, only as a setup choice for baselines.
> 3. **Our work investigates a fundamentally different question**.
> In Section 3.1.1 (and Appendix B–C), we prove that ORPO and ASFT already contain an **implicit SFT-like term inside their alignment component**. This yields a clear testable hypothesis:
>
>     > Once an SFT term is already contained within the alignment component, is it better to keep optimizing a single mixed objective (SFT + alignment) in one stage, or to separate SFT into its own stage and then optimize an alignment term in a second stage, which, as we show, already contains an SFT-like contribution and therefore does not need an additional SFT regularization term?
>     >
>
>     To answer this, we evaluate three regimes for ORPO/ASFT in Table 1, and use pure SFT (row 1) and two-stage DPO (row 8) as baselines under the same overall protocol.
>
>     - base-initialized single-stage ORPO/ASFT (Table 1, rows 4–5);
>     - SFT-initialized single-stage ORPO/ASFT (Table 1, rows 6-7), i.e. what SimPO uses;
>     - standalone SFT -> ORPO/ASFT alignment (our two-stage reorganization) (Table 1, rows 2-3);
>
>     These controlled comparisons show that the one-stage formulation with an explicit $L_\text{SFT}$ term during alignment is suboptimal compared to the two-stage SFT -> alignment-only formulation. In other words, what matters is not simply “starting from an SFT checkpoint” (as SimPO notes qualitatively), but reorganizing the objective so that SFT is done in its own stage and the alignment phase is not burdened with an additional SFT term.
>
> 4. **Our results therefore go beyond SimPO’s remark.**
>
>     In fact, in Table 1 the SFT-initialized one-stage ORPO$^\dagger$ performs worse than the base-initialized one-stage ORPO$^\dagger$. What improves performance in our study is **not** the choice of initialization alone (base vs. SFT), but the **reorganization of the training scheme**: we move SFT into a separate first stage and then run ORPO/ASFT using the alignment loss only, without an extra $L_\text{SFT}$ term during the alignment phase.
>
> 5. **The core novelty is the unified, loss-level analysis.**
> Our contribution is not “SFT sometimes helps”, SimPO already assumes this without analysis.
> Our novelty is:
>     - deriving the implicit SFT structure inside ORPO/ASFT;
>     - identifying why one-stage training is inconsistent;
>     - introducing a unified two-stage, $\beta$-tempered framework that puts ORPO/ASFT into the same hyperparameter space as other DAAs;
>     - and using this unified setup to isolate the **pairwise vs. pointwise** distinction as the main driver of alignment performance.
>
> In short, SimPO never inspects the structure of the ORPO loss; it only reports a qualitative setup choice. Our work provides the first **theoretical explanation and controlled experimental analysis** of how ORPO/ASFT behave when the objective is decomposed and reorganized, something that SimPO neither attempts nor claims to do.

---

> > ### Author Response · Authors · 2025-11-14
> >
> > **W2: Triple PO / Tree PO**
> >
> > Thank you for pointing us to these recent works; we will add them to the Related Work section. But we respectfully argue, that they fall outside the scope of our study.
> >
> > Our primary goal in this paper is to provide a *systematic and controlled* comparison of DAA objectives in the standard **offline alignment setting with binary preference pairs on static datasets**. To answer the question *“what actually matters when choosing an offline alignment method?”* we deliberately put all methods into the **same regime**: identical data format (chosen vs. rejected), the same SFT → alignment pipeline, and a comparable hyperparameter space. This tight control is what allows us to make clean causal statements about the effect of the objective itself (in particular, the pairwise vs. pointwise distinction).
> >
> > - Triple PO assumes a **richer supervision signal** than the methods we study: for each prompt it requires a *gold* trajectory in addition to preferred and rejected responses. In many public preference datasets and real-world deployments, such gold trajectories are either unavailable or only sparsely present. As a result, Triple PO exploits an extra source of information that other DAAs do not have access to. Evaluating it in our setting would either require changing the supervision format (collecting gold trajectories) or synthesizing “gold” from the chosen responses, both options would introduce additional confounders and make it impossible to attribute performance differences purely to the loss design.
> > - Tree PO goes even further and is explicitly designed for **multi-branch, multi-step preference trees** and listwise rankings over many trajectories per prompt. In their work, this structure is obtained by synthetic Tree-of-Thought generation with Qwen followed by GPT-based grading of each trajectory. This supervision format (tree-structured, listwise, model-generated and model-scored) is very different from the human binary preference datasets that our paper and the DAAs we analyze are built around, and is much harder to obtain in practical pipelines. Bringing Tree PO into our experimental suite would require us to adopt their entire data-generation and ranking pipeline, which would introduce substantial additional nuances coming from data design rather than from the alignment objective itself and would blur the scope of our analysis.
> >
> > In summary, Triple PO and Tree PO explore **distinct supervision regimes** (gold trajectories; preference trees and listwise rankings), whereas our paper is intentionally restricted to **binary preference data with a separate SFT corpus**. We view these methods as complementary directions that are best studied in dedicated work on enriched feedback structures, and not as baselines for the specific, tightly controlled comparison of offline DAAs that we perform here.

---

> > > ### Author Response · Authors · 2025-11-14
> > >
> > > **W3: KTO and CPO inclusion**
> > >
> > > Thank you for the suggestion to include KTO and CPO. We agree that these methods are interesting, and we carefully evaluated whether adding them would improve the methodological value of our study. Below we explain why they would **not** change our conclusions, and why they were not included in the final experimental suite.
> > >
> > > **About CPO**
> > >
> > > CPO is explicitly defined as a combination of a contrastive preference loss and an additional NLL term on preferred samples. Concretely, its preference component is a DPO/SimPO-style objective:
> > >
> > > $$
> > > \mathcal{L}\_\text{prefer}= -\mathbb{E}\_{(x,y_w,y_l)}\Big[\log \sigma\big(\beta\log \pi_\theta(y_w|x)-\beta\log \pi_\theta(y_l|x)\big)\Big].
> > > $$
> > >
> > > This term can be rewritten as
> > >
> > > $$
> > > \mathcal{L}\_\text{prefer}= -\beta\mathbb{E}[\log \pi\_\theta(y_w|x)]+ \mathbb{E}\Big[\log(\pi\_\theta(y_w|x)^{\beta}+\pi\_\theta(y_l|x)^{\beta})\Big].
> > > $$
> > >
> > > The first part is exactly an SFT/NLL-like term on winners (scaled by $\beta$), while the second part is a contrastive coupling between winner and loser. The full CPO loss then adds yet another NLL term on winners:
> > >
> > > $$
> > > \mathcal{L}\_\text{CPO}= \mathcal{L}\_\text{prefer}- \mathbb{E}[\log \pi\_\theta(y_w|x)].
> > > $$
> > >
> > > Thus, the full CPO objective belongs to the same family as the original ORPO/ASFT: it combines an alignment loss that already contains an implicit SFT component with an additional explicit one. Our work demonstrates (Section 5.1, Table 1, lines 4-7) that this "redundant SFT during alignment" is consistently suboptimal once methods are placed in a proper two-stage framework.
> > >
> > > Crucially, the CPO paper itself validates its method using a **single-stage training process**, starting from an already fine-tuned model. They compare their combined loss against its individual components ($\mathcal{L}\_\text{prefer}$, $\mathcal{L}\_\text{NLL}$), but never test the two-stage protocol our work investigates: a dedicated SFT phase followed by an alignment stage that only uses the pure preference objective.
> > >
> > > Therefore, CPO is not a missing baseline that challenges our findings, but rather another instance of the one-stage design whose limitations we analyze and overcome. We expect CPO would similarly benefit from our proposed two-stage framework, making its inclusion a redundant validation of our central claim at a significant computational cost.

---

> > > > ### Author Response · Authors · 2025-11-14
> > > >
> > > > **About KTO**
> > > >
> > > > Although the KTO paper uses pairwise preference data as input, the **training objective itself is fundamentally pointwise**. KTO explicitly decomposes each preference pair $(x, y_w, y_l)$ into two independent examples with labels "desirable" and "undesirable". The loss for each $(x, y)$ is of the form:
> > > >
> > > > \\[
> > > > v(x,y)=
> > > > \\left\\{
> > > >   \\begin{array}{ll}
> > > >     \\lambda_D\\,\\sigma\\left(\\beta \\left(r_\\theta(x,y)-z_0(x)\\right)\\right), & \\text{if $y$ is desirable}, \\\\
> > > >     \\lambda_U\\,\\sigma\\left(\\beta \\left(z_0(x)-r_\\theta(x,y)\\right)\\right), & \\text{if $y$ is undesirable}
> > > >   \\end{array}
> > > > \\right.
> > > > \\]
> > > >
> > > >
> > > >
> > > > where $r_\theta(x,y) = \log \frac{\pi_\theta(y\mid x)}{\pi_{\text{ref}}(y\mid x)}$ and $z_0(x) = \mathrm{KL}(\pi_\theta(\cdot\mid x)||\pi_{\text{ref}}(\cdot\mid x))$.
> > > >
> > > > Thus, the full KTO objective is
> > > >
> > > > $$
> > > > L_{\text{KTO}} =
> > > > \mathbb{E}_{(x,y)}\big[ \lambda_y - v(x,y) \big],
> > > > $$
> > > >
> > > > which is a **sum over individual (x, y)** terms.
> > > > Critically:
> > > >
> > > > - The loss does **not** depend on the pairwise difference $r_\theta(x,y_w) - r_\theta(x,y_l)$.
> > > > - The desirable/undesirable examples interact only through the prompt-level baseline $z_0(x)$, not through a direct comparison of scores of matched winners and losers.
> > > > - Permuting which loser belongs to which winner in the dataset *does not change* KTO’s optimization - a defining property of pointwise methods in our taxonomy.
> > > >
> > > > Therefore, KTO belongs squarely to the **pointwise** family in our framework, behaving analogously to APO-Zero, NCA, Cal-DPO, and ASFT.
> > > >
> > > > In early experiments, we did run KTO on Llama3 3B and 8B under the UF setup, using the same style of learning-rate (1e-6, 7e-7, 5e-7, 3e-7) and $\beta$ sweeps (1e-4, 3e-4,5e-4, 1e-3, 3e-3,  5e-3, 7e-3, 0.01, 0.03, 0.05, 0.1) similar to other methods. The best configurations we found achieved AlpacaEval scores in the same range as the other pointwise objectives in Table 3 and did not exhibit qualitatively new behavior.
> > > >
> > > > The best settings we found (reported via AlpacaEval 2) were (DPO and NCA added for context):
> > > >
> > > > | Method | LR | β | LC (std) | WR (std) |
> > > > | --- | --- | --- | --- | --- |
> > > > | KTO (8B) | 3e-7 | 0.0005 | 24.2 (0.82) | 23.09 (1.3) |
> > > > | DPO (8B) | 1e-6 | 0.003 | 26.82 (0.77) | 23.69 (1.25) |
> > > > | NCA (8B) | 3e-7 | 0.0003 | 23.21 (0.80) | 18.67 (1.17) |
> > > > |  |  |  |  |  |
> > > > | KTO (3B) | 5e-7 | 0.005 | 10.13 (0.54) | 10.28 (0.93) |
> > > > | DPO (3B) | 1e-6 | 0.01 | 11.43 (0.58) | 11.79 (0.99) |
> > > > | NCA (3B) | 3e-7 | 0.0005 | 10.33 (0.53) | 11.02 (0.97) |
> > > >
> > > > These results are **fully consistent with our Table 3**: KTO behaves like the other pointwise objectives, without exhibiting qualitatively new trends. Further large-scale sweeps would be computationally expensive and, based on these experiments, are unlikely to affect any of our conclusions, so we chose to focus the main comparison on a smaller set of objectives. If you find it helpful, we can explicitly mention KTO as a **pointwise** method in our taxonomy and briefly summarize these preliminary results in the appendix to clarify that it behaves consistently with the other pointwise DAAs under our unified setup.
> > > >
> > > > In summary, both CPO and KTO fit naturally into categories already covered by our analysis:
> > > >
> > > > - CPO: an alignment objective with an SFT-like term embedded in the preference loss **plus** an explicit NLL term, conceptually aligned with our ORPO/ASFT analysis of “redundant SFT during alignment”.
> > > > - KTO: structurally **pointwise**, matching the behavior of the other pointwise methods we study.
> > > >
> > > > Including them would significantly increase the experimental cost while reinforcing, rather than challenging, our main claim that the primary driver of performance differences between DAAs is the **pairwise vs. pointwise** nature of the ranking objective.
> > > >
> > > > We appreciate your thoughtful review. If there are remaining concerns that would prevent a higher score, we’d be grateful to address them during the discussion.

---

> > > ### Comment · Reviewer_u7nZ · 2025-11-25
> > >
> > > I would also like to respond to the authors’ statement that “in many public preference datasets and real-world deployments, such gold trajectories are either unavailable or only sparsely present.” In the TPO paper, the authors explicitly show that even when y_gold has a judgment score similar to y_preferred, TPO still achieves significantly better performance than DPO. Refer to section 4.4. This contrasts with the explanation given in the current paper, which suggests that TPO fundamentally requires a strong supervision signal.
> > >
> > >
> > > Furthermore, I do not believe data availability or curation should be considered a primary limitation in alignment, as multiple recent datasets provide large quantities of multi-response preference data. Examples include:
> > >
> > > 1. UltraFeedback 60K – 4 ranked responses per prompt
> > > https://huggingface.co/datasets/openbmb/UltraFeedback
> > >
> > > 2. Nectar 183K – 7 ranked responses per prompt
> > > https://huggingface.co/datasets/berkeley-nest/Nectar
> > >
> > > 3. UltraFeedback-ArmoRM – 5 ranked responses per prompt
> > > https://huggingface.co/datasets/princeton-nlp/llama3-ultrafeedback-armorm
> > >
> > > 4. UltraFeedback-PairRM – 5 ranked responses per prompt
> > > https://huggingface.co/datasets/princeton-nlp/llama3-ultrafeedback
> > >
> > > Given these resources and recent advances, I do not see the proposed method as introducing a fundamentally new direction. The alignment landscape is no longer limited to pairwise optimization, and newer Direct Alignment Algorithms, such as TPO, have already demonstrated the ability to overcome the challenges outlined in the paper.

---

> > > > ### Author Response · Authors · 2025-11-26
> > > >
> > > > Thank you for your comments, but it seems we are still talking past each other.
> > > >
> > > > Before addressing specific points, let us clearly define the terminology:
> > > >
> > > > - By an **SFT stage** we mean a *separate training phase* on supervised (x, y_gold) data that precedes preference optimization.
> > > > - By an **SFT term** we mean an *explicit NLL / log-likelihood component* $L_{\text{SFT}}$ that appears **inside the alignment loss itself** and is optimized jointly with the preference $L_{\text{Align}}$ term in a one-stage objective.
> > > >
> > > > In Section 3.1.1 and Appendix D, we explicitly manipulate the ORPO/ASFT losses and show algebraically that, for **reference-free odds-ratio objectives**, the alignment components $L\_{\text{ORPO}\_\text{Align}}$ and $L\_{\text{ASFT}\_\text{Align}}$ already contain an SFT contribution. This is a property of the *loss formula* itself and does not depend on how the model is initialized.
> > > >
> > > > Let us once more clarify the situation regarding SFT.
> > > >
> > > > **SFT with a reference model.**
> > > >
> > > > That the **SFT stage** should be run before methods that *use a reference policy* is an obvious fact, and nobody disputes this. It is important so that the reference policy in DPO is close to the distribution of the chosen responses. This is stated explicitly in the DPO paper (https://arxiv.org/pdf/2305.18290)
> > > >
> > > > > “Since the preference datasets are sampled using $π^\text{SFT}$, we initialize $π^\text{ref}=π^\text{SFT}$ whenever available. … This procedure helps mitigate the distribution shift between the true reference distribution which is unavailable.”
> > > > >
> > > >
> > > > (page 5).
> > > >
> > > > **What SimPO does.**
> > > >
> > > > In our original response we quoted exactly the same sentence from SimPO that you now cite:
> > > >
> > > > > “*ORPO can directly train on preference data without the SFT stage. For fair comparisons, we start ORPO from the same SFT checkpoints as other baselines, which yields better results than starting from base checkpoints.*”
> > > > >
> > > >
> > > > This means that they apply the *original* ORPO loss with an **SFT term**.
> > > >
> > > > $L\_{\text{ORPO}} = L\_{\text{SFT}} + \lambda L\_{\text{ORPO}\_{Align}}$ after the **SFT stage**, where $L\_{\text{ORPO}\_{Align}}$ is $L\_\text{OR}$ from the original ORPO paper (Eq. 7 in https://arxiv.org/pdf/2403.07691).
> > > >
> > > > **What we propose.**
> > > >
> > > > We show (Section 3.1.1 and Appendix D) that the $L\_{\text{SFT}}$ **SFT term** (NLL component) is already embedded inside the reference-free odds–ratio alignment losses $L\_{\text{ORPO}\_\text{Align}}$ and $L\_{\text{ASFT}\_\text{Align}}$*,* which makes any additional explicit $L\_{\text{SFT}}$ **term** potentially redundant. Concretely, if we *remove* this explicit one-stage-style $L\_{\text{SFT}}$ term (the extra NLL component), then, according to Table 1, ORPO reaches 24.1 LC-WR and 15.3 AH-WR, and ASFT reaches 16.4 LC-WR and 10.6 AH-WR. By contrast, the SimPO-style setup, with which we also compare (rows 6–7 in the same Table 1), performs worse: ORPO gets 13.4 LC-WR and 7.7 AH-WR, and ASFT 11.4 LC-WR and 7.5 AH-WR. This makes the effect of removing the explicit SFT term in **reference-free $r_\text{odds}$ methods** a genuinely new observation, distinct from the well-known “SFT-before-DPO” recipe for reference-based objectives.
> > > >
> > > > Thus, as we already wrote, **SimPO never compares ORPO with vs. without the SFT term in the loss** - they never remove the **SFT term** from the original one-stage method. This distinction is important precisely for $r_\text{odds}$-based methods, which were originally designed **without** a separate **SFT stage** but **with** an explicit **SFT term** (the NLL component) in the loss.

---

> > ### Comment · Reviewer_u7nZ · 2025-11-25
> >
> > Thanks to the authors for their response. I would like to emphasize that the reorganization of the training schema they describe is not new. Similar approaches have been explored in prior alignment work, such as Zephyr (https://arxiv.org/abs/2310.16944), which was one of the earlier works demonstrating that SFT is a necessary stage for Direct Preference Optimization (DPO). This idea is further examined in Insight into Alignment (https://arxiv.org/abs/2404.14723v2), which analyzes several state-of-the-art direct preference methods.
> >
> > Regarding the SimPO paper, the authors clearly state on page 6 (https://arxiv.org/pdf/2405.14734) that: *ORPO can directly train on preference data without the SFT stage. For fair comparisons, they start ORPO from the same SFT checkpoints as other baselines, which yields better results than starting from base checkpoints.*
> >
> > Regarding the pairwise comparison, I still find it difficult to understand why the authors primarily focus on pairwise methods, especially given that the issue they highlight has already been addressed by other Direct Alignment Algorithms (DAA), such as TPO (https://arxiv.org/abs/2405.16681v2). In this context, it is also unclear why TPO is not considered a subset of DAA.
> >
> > If the authors could clarify this point, it would greatly help readers better understand the methodological distinctions.

---

> ### Author Response · Authors · 2025-11-26
>
> Regarding the arXiv-only TPO work and its setting with a separate gold trajectory: the authors themselves note in Section 4.4
>
> > “TPO is based on the assumption that the dataset D = {x, y_gold, y_w, y_l} satisfies y_gold ≻ y_w ≻ y_l. However, constructing such a dataset can be challenging, as in many cases, two of the preferences may have very similar scores.”
> >
>
> This directly confirms our concern about the difficulty of obtaining strictly better gold trajectories; the TPO authors explicitly acknowledge this. We also explicitly state, and this is already reflected in the revision, that our methodology is intended to isolate the factors that reliably influence DAA behavior within the binary preference regime.
>
> When a gold trajectory is *absent*, TPO itself writes:
>
> > “This finding suggests that even when y_gold and y_w share the same score, using distinct responses for y_gold and y_w yields significantly better performance compared to the case where y_gold = y_w, which is similar to the CPO loss function.”
> >
>
> This is exactly the CPO-like case and is, as we wrote, “another instance of the one-stage design whose limitations we analyze and overcome.”
>
> Finally, we note that TPO is currently an arXiv-only work. According to the ICLR 2026 reviewer guidelines (https://iclr.cc/Conferences/2026/ReviewerGuide), authors are not required to perform full empirical comparisons to concurrent or arXiv-only papers, and the absence of such experiments should not be treated as a reason for rejection. We work in an explicitly stated scope, and in the Related Work we already mention directions that use additional signals beyond chosen/rejected, which we believe adequately reflects the current state of the field.
>
> We hope that these clarifications help resolve the misunderstanding and allow you to better appreciate the specific contribution of our work.

---

> > ### Comment · Reviewer_u7nZ · 2025-11-26
> >
> > Thank you for the authors’ response. I appreciate the clarifications provided. However, my concerns remain primarily on the optimization side. The paper focuses on direct alignment algorithms, yet the discussion centers almost exclusively on pairwise methods. As I previously mentioned, approaches such as TPO demonstrate clearly that the landscape is broader and that meaningful advances are emerging beyond pairwise formulations.
> >
> > Given that this work is framed as an analysis paper titled “Differences Between Direct Alignment Algorithms”, it is difficult to justify the omission of triple-preference optimization methods, especially since several recent algorithms in this category have shown promising performance and have been accepted to ICLR 2025. Unfortunately, the authors did not address why these methods were excluded from the discussion.
> >
> > For these reasons, I will maintain my overall score and adjust the contribution rating to 1.
> >
> > Below is a list of triple-preference algorithms accepted to ICLR 2025:
> >
> > XPO: https://openreview.net/forum?id=QYigQ6gXNw
> > VPO: https://openreview.net/forum?id=SQnitDuow6
> > TPO: https://openreview.net/forum?id=O0sQ9CPzai

---

### Official Review · Reviewer_JxS9 · 2025-10-28

**Soundness:** 2
**Presentation:** 2
**Contribution:** 2
**Rating:** 2
**Confidence:** 4

**Summary:**

This paper aims to unify various Direct Alignment Algorithms (DAAs) such as DPO, ORPO, ASFT, and others under a common training framework. It does so by introducing two key adjustments: (1) converting one-stage methods (e.g., ORPO and ASFT) into a two-stage setup by isolating a supervised fine-tuning (SFT) phase, and (2) introducing a β scaling parameter to odds-ratio-based methods that originally lacked it. The authors argue that under this unified framework, the most significant factor affecting alignment quality is the choice between pairwise vs. pointwise ranking objectives, rather than the specific scalar score used (e.g., odds ratio or policy-reference ratio). The findings are backed by extensive empirical studies across instruction-following and math reasoning benchmarks.

**Strengths:**

1. The paper is well-written and clearly structured, making complex theoretical concepts accessible.
2. The experimental coverage is thorough, spanning multiple models (LLaMA 3 3B/8B, Qwen 2.5 7B/14B) and tasks (e.g., TL;DR, UltraFeedback, math reasoning).
3. The investigation into pairwise vs. pointwise objective effects is well-motivated and adds nuance to current discourse around preference optimization.

**Weaknesses:**

1. Lack of Novelty in β Introduction
The paper presents the introduction of a β parameter to odds-ratio-based methods (ORPO, ASFT) as a key contribution (Section 3.1.2). However, open-source implementations (e.g., Hugging Face’s trl/orpo_trainer.py) already incorporate β scaling in reward computation:
```
chosen_rewards = self.beta * logp_chosen
rejected_rewards = self.beta * logp_rejected
```
This suggests that the use of β is not novel and has already been explored in practice.

2. Low Impact of Two-Stage Conversion
The paper emphasizes the benefit of converting one-stage methods to two-stage ones with an explicit SFT phase (Section 3.1.1, Section 5.1). However, this setup mirrors conventional RLHF pipelines and is already widely accepted in practice. The experimental confirmation of this expected result lacks sufficient novelty to be positioned as a central contribution.

3. Insufficient Justification for Claims of “Unification”
The claim that the proposed framework “unifies” DAAs feels more like a reframing of existing training practices than a substantial theoretical advancement (Sections 3–4). Particularly, the conversion of ORPO/ASFT to two-stage with β introduces tuning knobs rather than fundamentally new methods.

4. Under-explored Comparison with Other Recent Techniques
While the paper compares a wide range of DAAs, it overlooks emerging approaches such as iterative label refinement, policy interpolation, or listwise ranking methods (e.g., LIPO, Cal-DPO). This weakens the generalizability and completeness of the claims (Section 5.3).

5. Limited Generalization Across Domains and Model Scales
Although the paper reports results across several tasks and models, it remains unclear how well the findings generalize to broader domains such as multi-turn dialogue, multilingual alignment, or real-world deployment scenarios. The benchmarks are still relatively narrow (Section 5, Appendix B).

6. Framing Over Substance
Many of the key observations (e.g., pairwise objectives are often more robust than pointwise ones) are already intuitively understood in the ranking literature. The paper validates them empirically but lacks deep theoretical insight or novel mechanisms to move the field forward (Section 6).

**Questions:**

1. Can the authors clarify whether they were aware that ORPO implementations already used β scaling? If so, why is it presented as a novel addition?
2. Given the well-established practice of SFT preceding alignment (e.g., in RLHF), how do the authors justify positioning this conversion to two-stage training as a key contribution?
3. Did the authors consider comparing against listwise or iterative label refinement-based methods? If not, how can they justify the completeness of the comparative study?

---

> ### Author Response · Authors · 2025-11-13
>
> Thank you for your thoughtful review and for highlighting the clarity of our presentation, the breadth of our experimental coverage across models and tasks, and the value of our nuanced analysis. We appreciate your recognition of these aspects and the opportunity to address your concerns:
>
> **W1/Q1: “$ \beta $  already exists in ORPO implementations, so the idea is not new.”**
>
> We respectfully but firmly note that this statement is factually incorrect. In existing ORPO implementations, such as Hugging Face’s *trl*, the hyperparameter called `beta` does not act as a temperature on the odds ratio. Instead, it coincides with the λ coefficient from the original ORPO paper and simply rescales the overall alignment loss component. It is never used to scale the log-odds difference itself.
>
> To illustrate, here is the relevant code snippet from trl's `odds_ratio_loss` function
>
> ```python
> def odds_ratio_loss(
>     self, policy_chosen_logps: torch.FloatTensor, policy_rejected_logps: torch.FloatTensor
> ) -> tuple[torch.FloatTensor, torch.FloatTensor, torch.FloatTensor, torch.FloatTensor, torch.FloatTensor]:
>
>     log_odds = (policy_chosen_logps - policy_rejected_logps) - (
>         torch.log1p(-torch.exp(policy_chosen_logps)) - torch.log1p(-torch.exp(policy_rejected_logps))
>     )
>     ratio = F.logsigmoid(log_odds)
>     losses = self.beta * ratio
>
>     chosen_rewards = self.beta * (policy_chosen_logps.to(self.accelerator.device)).detach()
>     rejected_rewards = self.beta * (policy_rejected_logps.to(self.accelerator.device)).detach()
>
>     return losses, chosen_rewards, rejected_rewards, torch.mean(ratio), torch.mean(log_odds)
> ```
>
> As the code shows, `self.beta` (i.e., $\lambda$) multiplies only the final `logsigmoid(log_odds)` term, while the `log_odds` themselves are never scaled. This is materially different from introducing a temperature **inside** the log-odds, and consequently the gradient dynamics are fundamentally different from our proposal.
>
> Our contribution is the introduction of a temperature parameter $ \beta $  that scales the reward difference, i.e., `logsigmoid(β * log_odds)`. This is a novel and critical modification. As we detail in Appendix E, this tempering directly controls the gradient magnitude and plays for reference-free odds-ratio objectives the same conceptual role that $ \beta $ plays in DPO-style objectives with an explicit reference policy: it becomes the knob that trades off the strength of preference enforcement against the size of parameter updates from the SFT initialization, even though no KL term or reference model appears in the loss. The original ORPO and ASFT papers lack any such mechanism, and existing implementations do not introduce it either. Our work is the first to motivate, introduce, and empirically validate this "DPO-style" temperature in ORPO/ASFT, showing that it unlocks significant performance gains (Figure 2, Section 5.2) and is essential for a fair and unified comparison with other DAAs.

---

> > ### Author Response · Authors · 2025-11-13
> >
> > **W2/Q2: "Two-stage training is the RLHF default; demonstrating its benefit is not novel."**
> >
> > It is obvious, that SFT pre-training is a standard practice in the broader RLHF pipeline. However, our contribution is specific and directly challenges the core premise of ORPO and ASFT, which were explicitly proposed as **one-stage** algorithms designed to unify SFT and alignment.
> >
> > Our work provides two key insights that are far from obvious:
> > 1. **Theoretical:** We show in Section 3.1.1 that the SFT objective is already implicitly contained within the alignment part of the loss of these methods (formally showed in Appendix C and E). This implies that adding an explicit L_SFT term *during alignment* is theoretically redundant and may even be counterproductive.
> > 2. **Empirical:** We demonstrate that original one-stage design is suboptimal. By converting to a two-stage process (an explicit SFT phase followed by alignment), we significantly improve performance. For example, our change boosts ORPO's win-rate from 14.8% to **24.1%** on Llama-3 8B with the UltraFeedback dataset, nearly closing the gap with DPO (Table 1).
> >
> > 3. **Conceptual:** Our work challenges the implicit assumption that a two-stage process is only beneficial for methods with a reference policy and a KL constraint (like DPO). The community often conflates "two-stage" with the full DPO/RLHF machinery. Our findings demonstrate that this is a false equivalence. We show that the benefits of separating SFT and alignment are more fundamental. Even for methods like ORPO and ASFT that lack a reference policy during training, a distinct SFT phase, followed by an alignment phase *without* the redundant SFT loss, is the optimal approach. This decouples the principle of staged training from the specific implementation details of DPO, which is a novel and important finding.
> >
> > Therefore, this result is not a routine confirmation of existing practices; it is a direct refutation of the one-stage training recipe proposed in the original ORPO/ASFT papers. Methodologically, this step is also what enables a truly unified and fair comparison across all DAAs, forming the bedrock for our main analysis, which you rightly noted as a strength. Viewed from this perspective, our contribution is novel in two complementary senses: **(i)** we provide, to the best of our knowledge, the first systematic empirical evidence that converting explicitly “one-stage” methods such as ORPO and ASFT to a two-stage SFT -> alignment pipeline yields substantial gains, and **(ii)** we make explicit and formalize a design principle that was, at best, implicit folk wisdom in the community rather than an articulated and tested hypothesis.
> >
> > **W3: "Unification’ is only reframing."**
> >
> > We wish to clarify that the goal of our "unification" is not to propose new standalone methods, but to resolve a critical methodological gap in the literature. Prior work has been fragmented, with methods like ORPO/ASFT being incomparable to DPO-style methods due to different training pipelines (one-stage vs. two-stage) and hyperparameter spaces (lacking $ \beta $).
> >
> > Our framework establishes a principled, common ground for comparison by:
> >
> > 1. Standardizing the training pipeline to SFT -> Alignment for all DAAs.
> > 2. Introducing a temperature parameter ($ \beta $), motivated by gradient analysis, to odds-ratio methods, placing them in the same hyperparameter space as other DAAs.
> >
> > This rigorous setup is precisely what enables our core analysis of the factors driving alignment performance. The valuable insight that the ranking objective (pairwise vs. pointwise) is the primary driver would have been impossible to uncover without this methodological unification.

---

> > > ### Author Response · Authors · 2025-11-13
> > >
> > > **W4/Q3: "Missing list-wise or iterative-refinement baselines."**
> > >
> > > Our primary goal in this work is to provide a *systematic and controlled* comparison of DAA objectives in the standard **offline alignment setting with binary preference pairs on static datasets**. To answer the question “what actually matters when choosing an offline alignment method?”, we deliberately put all methods into the *same* regime: the same data format, the same training pipeline (SFT → alignment), and a comparable hyperparameter space. This tight control is what allows us to make clean causal statements about the effect of the objective itself.
> > >
> > > With this in mind, we would first like to respectfully clarify that **Cal-DPO** — which is not a listwise method — **is indeed included in our comparison** across all models and tasks (Tables 2, 3, 12, and 13).
> > >
> > > By contrast, the techniques mentioned by the reviewer fall into paradigms that are *not* directly comparable under this controlled setup:
> > >
> > > - **Listwise methods** such as LiPO assume a different supervision format (ranking of N>2 candidates) and, in practice, often rely on additional components such as an external ranker. Applying them on our pairwise-only datasets would either require re-annotating the data (changing the underlying problem) or synthesizing listwise labels from pairwise ones (baking in method-specific biases). Similarly, comparing pipelines that depend on an auxiliary ranker would entangle the quality of that external model with the intrinsic properties of the loss, which is precisely what our study aims to *factor out*.
> > > - **Iterative or policy-interpolation methods** change the data distribution over the course of training (through iterative label refinement or online data collection), whereas our analysis is explicitly about the behavior of different objectives on a *fixed* preference dataset. Mixing offline-static and online-dynamic regimes would make it much harder to attribute performance differences to the objective, rather than to differences in data generation.
> > >
> > > For these reasons, we intentionally restricted the scope to the dominant and practically important regime of **pairwise, static offline alignment methods**, including strong and widely used representatives DAAs. Broadening the comparison to listwise or iterative methods within the same paper would require relaxing the very controls that make our conclusions sharp, and would likely *dilute* rather than strengthen the core contribution.
> > >
> > > We fully agree that understanding how our findings extend to listwise or iterative-refinement paradigms is an interesting next step, and we will clarify in the paper that our claims are explicitly targeted at offline DAAs trained on binary preference data, rather than at *all* possible alignment pipelines.
> > >
> > > **W5: "Limited generalisation across domains."**
> > >
> > > We respectfully argue that this critique misinterprets the central contribution of our paper. Our work is not intended as a broad survey of DAA performance across all possible domains, but as a targeted methodological intervention designed to resolve specific, documented inconsistencies in the literature.
> > >
> > > The choice of benchmarks was therefore not a limitation, but a **deliberate and necessary design choice**. We focused on instruction-following, summarization, and math reasoning precisely because these are the domains where DAAs made competing claims. To fairly evaluate and deconstruct these claims, we had to meet them on their own terms, using their own evaluation paradigms.
> > >
> > > Our key finding, that the pairwise vs. pointwise distinction is the primary performance driver, a point you highlighted as a strength, could only have been revealed through such a controlled, head-to-head comparison. A broader, less focused study across many disparate domains would have introduced too many confounding variables, making it impossible to isolate the core algorithmic factors at play.
> > >
> > > Thus, our work provides the foundational clarity that the field currently lacks. It establishes which design choices genuinely matter, creating a solid platform from which future research can, and should, explore broader generalization.

---

> > > > ### Author Response · Authors · 2025-11-13
> > > >
> > > > **W6: "Framing over substance: no deep theory."**
> > > >
> > > > We respectfully contend that our work provides precisely the novel mechanism the reviewer finds lacking, moving far beyond the well-known but poorly understood intuition that "pairwise is better than pointwise," especially in the context of LLM alignment. Our contribution is not merely validating an old idea, but proposing and verifying a concrete causal model for *why* and *when* performance gaps between DAA families emerge.
> > > >
> > > > Specifically, we first formalize the abstract concept of "prompt bias" as a concrete mathematical quantity: the **marginalized score**, $\mathbb{E}\_y[r\_\theta(x,y)]$ (Section 6). This allows us to demonstrate a fundamental structural difference between objective classes. Crucially, we show that **pointwise objectives** are compelled to expend finite model capacity on "unlearning" this bias by driving the marginalized score to zero. In contrast, **pairwise objectives** are structurally invariant to this bias, allowing them to dedicate the model's full capacity to the primary ranking task.
> > > >
> > > > This isn't just theory; it's a testable hypothesis about a capacity-for-bias-removal trade-off. We verify it directly:
> > > >
> > > > 1. Our **controlled experiment in Appendix F** isolates and confirms this exact mechanism, showing the performance gap emerges *only* when model capacity is limited and data bias is present.
> > > > 2. Our **ICC analysis on real data (Figure 7)** demonstrates this mechanism is active in practice, as pairwise methods retain higher prompt bias (higher ICC) precisely in the medium-difficulty regime where they show a clear performance advantage.
> > > >
> > > > Therefore, our work provides the first mechanistic explanation for these observed performance differences in LLM alignment. This moves the field from folk wisdom to a predictive framework, which we believe is a significant and novel contribution.
> > > >
> > > > We appreciate your critical review. If there are remaining concerns that would prevent a higher score, we’d be grateful to address them during the discussion.

---

> ### Comment · Reviewer_JxS9 · 2025-11-25
>
> Thank you for the clear clarifications. I appreciate your distinction between β-tempering and λ-scaling, and your framing of pairwise vs. pointwise as the structural axis of comparison.
>
> That said, I’d like to better understand how your β-tempering relates to recent work like AlphaPO, which also introduces a parameterized reward-shaping mechanism (α) within pairwise objectives. While you argue these operate along different axes, there appears to be conceptual overlap in how both approaches control preference signal strength through reward transformation.
>
> Given this, I’m wondering whether the novelty of introducing β in the context of ORPO/ASFT might be reduced in light of these similar ideas emerging concurrently. To what extent do you see your approach as conceptually distinct versus complementary to this line of work? I’m asking to better situate the contribution within the evolving landscape of reward shaping in alignment.

---

> > ### Author Response · Authors · 2025-11-25
> >
> > Thank you for raising the connection to AlphaPO, this is an important piece of concurrent work, and we appreciate the chance to clarify how it relates to our β–tempering of odds–ratio objectives.
> >
> > Our paper explicitly studies two scalar score families for direct alignment: the reference-ratio / log-likelihood family $r_{\text{ref}}(y,x) = \log \frac{\pi_\theta(y|x)}{\pi_{\text{ref}}(y|x)}$ and the odds-ratio family $r_{\text{odds}}(y,x) = \log \frac{\pi_\theta(y|x)}{1-\pi_\theta(y|x)}$. We place methods based on both scores into the same unified two-stage SFT → alignment protocol and show that, once unified, the primary axis that determines performance is the ranking objective (pairwise vs pointwise), while the choice between $r_{\text{ref}}$ and $r_{\text{odds}}$ is secondary.  In this framework, ORPO and ASFT are the representatives of the odds-ratio family; originally they are one-stage, reference-free methods without any temperature parameter β in the reward difference. A key technical contribution of our work is to introduce, motivate, and analyze a DPO-style temperature β for these odds-ratio objectives: for example, we use losses of the form $-\log \sigma(\beta (r_{\text{odds}}(y_w,x) - r_{\text{odds}}(y_l,x)))$ for ORPO-like alignment.  This β multiplies only the difference in odds-ratio rewards inside the logistic link; the odds-ratio score $r_{\text{odds}}$ itself is left unchanged. In other words, β is a pure temperature on the reward difference, controlling the magnitude of gradients and the trade-off between enforcing preferences and staying close to the SFT initialization, but it does not alter the functional form of the reward. Our gradient analysis in Appendix E makes this explicit: β scales the argument of $\sigma$ and thus the steepness of the pairwise objective, but the underlying odds-ratio geometry is fixed.  This is precisely analogous to β in DPO, but applied for the first time to reference-free odds-ratio objectives, where no such knob existed.
> >
> > AlphaPO, in contrast, does not operate on odds-ratio scores at all; it stays entirely within the log-likelihood / SimPO-like family. Its reward is defined as
> >
> > $$
> > r_\alpha(y;x)
> > = \frac{\beta}{\alpha}\Bigl(1 - \pi_\theta(y|x)^{-\alpha/|y|}\Bigr),
> > $$
> >
> > with the limit $\alpha \to 0$ recovering the SimPO reward $r_{\text{SimPO}}(y;x) = \frac{\beta}{|y|}\log \pi_\theta(y|x)$.  This is a standard α-divergence construction: $\frac{x^\alpha - 1}{\alpha} \to \log x$ as $\alpha \to 0$. In this sense, AlphaPO builds a one-parameter family of reward functions that interpolate between different “shapes” around the log-reward; the role of α is to change the curvature and asymmetry of the mapping from probabilities $\pi_\theta(y|x)$ to scalar rewards, not merely to scale an existing reward difference. Theorems 3.1 and 3.2 (with Corollary 3.3) in AlphaPO make this explicit: α enters inside the reward function itself, and the resulting gradients can change not only in magnitude but also in their relative weighting of preferred and dispreferred responses and even in the direction of change of $\pi_\theta(y_w|x)$.  This is reward shaping in the strict sense: different values of α correspond to genuinely different reward functions $r_\alpha$, with the log-reward only as a special limiting case.
> >
> > Formally, then, the mechanisms we and AlphaPO introduce are of different types. Our β for odds-ratio methods multiplies the fixed odds-ratio reward difference inside a sigmoid and leaves $r_{\text{odds}}$ unchanged; it is a temperature parameter. AlphaPO’s α sits inside the reward, transforming $\pi_\theta(y|x)$ into $\pi_\theta(y|x)^{-\alpha/|y|}$ and then into $r_\alpha$, so that changing α changes the reward landscape itself. Temperature scaling vs reward shaping are mathematically orthogonal: one rescales an argument of a link function, the other modifies the function being linked.

---

> ### Author Response · Authors · 2025-11-25
>
> It is instructive to ask what an “AlphaPO-style” construction would look like if one were to apply it to odds-ratio scores. Let us denote the odds as $\operatorname{odds}(p) = \frac{p}{1-p}$. A direct analog of AlphaPO’s reward family for odds-based scores would be something like
>
> $$
> r^{\text{odds}}_\alpha(p)
> = \frac{\beta}{\alpha}\bigl(1 - \operatorname{odds}(p)^{-\alpha}\bigr)
> = \frac{\beta}{\alpha}\left(1 - \left(\frac{1-p}{p}\right)^{\alpha}\right).
> $$
>
> *A straightforward limit calculation shows that*
>
> $$
> \lim_{\alpha\to 0} r^{\text{odds}}\_\alpha(p)
> = \beta\left(\log p - \log(1-p)\right)
> = \beta r_{\text{odds}}(p),
> $$
>
> so this family recovers the usual odds-ratio reward as a special case when $\alpha \to 0$. This illustrates two points. First, the odds-ratio and log-likelihood families are structurally different: AlphaPO’s $r_\alpha$ is always a function of $\pi_\theta(y|x)$ alone (with length normalization), not of the odds $\pi/(1-\pi)$. Second, extending the α-divergence idea to odds-ratio would require defining and analyzing a new family $r^{\text{odds}}_\alpha$ along these lines. This family does not appear in AlphaPO, which never considers $\log \frac{\pi}{1-\pi}$ as its base score, and it does not appear in ORPO, ASFT, or any other odds-ratio DAA either. AlphaPO therefore does not provide α-reward shaping for odds-ratio methods; it instead generalizes the SimPO/DPO-style log-reward family.
>
> From this perspective, our and AlphaPO’s contributions are complementary, not overlapping. AlphaPO addresses the question “within the log-likelihood / SimPO family, how does changing the reward shape via α affect likelihood displacement and alignment performance?” and remains confined to that score family.  Our work addresses a different but equally important question: “across scalar score families (reference-ratio vs odds-ratio) and ranking types (pairwise vs pointwise), what actually matters for alignment performance, once pipelines are unified and temperatures are made explicit?” and, in doing so, we are the first to introduce a DPO-style temperature β for reference-free odds-ratio objectives, analyze its gradient-level role, and empirically demonstrate that it is critical to make ORPO/ASFT competitive in a unified comparison.
>
> Finally, we would like to emphasize that AlphaPO is **concurrent work** that generalizes the log-likelihood / SimPO family and does not treat odds-ratio objectives at all. In the ICLR 2026 guidelines (https://iclr.cc/Conferences/2026/ReviewerGuide), such contemporaneous work is explicitly not required as a comparison point and cannot be used as a basis for rejection or score reduction. We therefore cite AlphaPO as related, complementary work because it extends the $r_{\text{ref}}$/log-likelihood axis that we also analyze, but it does **not** overlap with our core technical novelty: we are, to the best of our knowledge, the first to (i) introduce a DPO-style temperature β into reference-free odds-ratio DAAs (ORPO, ASFT) and (ii) place these odds-ratio methods on equal footing with reference-ratio methods in a unified two-stage, pairwise-vs-pointwise comparison. Any potential α-style reward shaping for $r_\text{odds}$ would itself be a new contribution beyond both AlphaPO and our work, and thus belongs to future research rather than to the novelty criteria of this submission.

---

> ### Comment · Reviewer_JxS9 · 2025-11-26
>
> Thank you for the clear and detailed clarification. The distinction between β-tempering over odds-ratio objectives and α-based reward shaping in AlphaPO is now much clearer, particularly in terms of their functional roles and underlying score families.
>
> While there may still be some conceptual parallels in the broader goal of modulating preference signal strength, I agree that the technical contributions are orthogonal. I appreciate the careful analysis you provided, and based on this, I’m updating my score to reflect that.

---

> > ### Author Response · Authors · 2025-11-26
> >
> > Thank you very much for engaging so carefully with our clarification and for updating your score, we really appreciate the time and attention you’ve put into understanding the $\beta$–tempering vs. $\alpha$–shaping distinction.
> >
> > If there is anything else we can clarify or strengthen in the revision, we would be very glad to do so. Your feedback has already helped us improve the paper, and since your overall assessment is still on the negative side, we’re happy to address any remaining concerns you may have to further strengthen the work.

---

### Official Review · Reviewer_MA69 · 2025-11-01

**Soundness:** 3
**Presentation:** 3
**Contribution:** 3
**Rating:** 6
**Confidence:** 3

**Summary:**

This paper focuses on Direct Alignment Algorithms (DAAs)—a class of methods that simplify LLM alignment by optimizing policies directly (bypassing reward modeling and RL). To validate this, the authors propose a unified training framework for DAAs: (1) converting one-stage methods (ORPO, ASFT) into a two-stage pipeline with an explicit SFT phase; (2) introducing a tempering parameter $\(\beta\)$ to unify all methods into the same hyperparameter space (improving odds-ratio DAAs like ORPO/ASFT).

**Strengths:**

1. The unified training protocol (converting one-stage DAAs to two-stage with \(\beta\)) is a novel solution to method incomparability.
2. The identification of the ranking objective as the core performance driver fills a gap in prior DAA analyses, which focused on SFT or scalar scores.

**Weaknesses:**

1. Experiments focus on medium-small models (3B, 7B, 8B, 14B) but omit larger scales (e.g., Llama 3 70B, Qwen 2.5 72B). Larger models often exhibit different alignment behaviors (e.g., reduced data sensitivity), so it remains unclear if the ranking objective’s dominance holds at scale.
2. The Section F validates the interaction between ranking objectives and prompt bias, but the paper lacks theoretical modeling of this mechanism.

**Questions:**

1. The paper validates results on models up to 14B parameters but not larger scales (e.g., Llama 3 70B, Qwen 2.5 72B). Do you anticipate the ranking objective’s dominance over scalar scores to hold for larger models, or might scale-induced changes (e.g., improved bias mitigation) alter this relationship?
2. You attribute performance gaps to the interaction between ranking objectives and prompt-specific biases. Do you plan to formalize this mechanism with a theoretical model or will you rely on experimental observations?
3. The paper mentions that "bias removal is beneficial remains an open question" - could you discuss potential scenarios where preserving prompt-specific biases might be advantageous?
4. Have you considered extending the unified framework to include online preference optimization methods?

---

> ### Author Response · Authors · 2025-11-12
>
> Thank you for your thoughtful review and for recognizing the novelty of our unified training framework and our work's contribution to analyzing DAA methods. We appreciate the opportunity to address your concerns:
>
> **W1/Q1: Scalability and the dominance of the ranking objective on larger models (e.g., 70B)**
>
> Our hypothesis, supported by our experiments, is that the performance gap between pairwise and pointwise methods is driven by the ratio of model capacity to task difficulty, not absolute model size. Our existing results already provide evidence for this principle across different regimes:
>
> **1. Evidence from the high-capacity regime:** We explore this dynamic by examining the other side of the ratio. Our results on the **Llama 3.2 3B TL;DR setup** (Table 2) demonstrate this: on a relatively simple task, the capable 3B model pushes all objective types to achieve similarly high performance. As our hypothesis predicts, the gap between methods vanishes when model capacity is ample for the task's complexity—the very scenario one would expect for a 70B model on moderately difficult tasks.
>
> **2. Direct validation via controlled experiments:** Our controlled experiment in Appendix F (Figures 5-6) provides a direct validation of this principle. By systematically varying model capacity (hidden size *h*), we show that all objectives converge once capacity greatly exceeds difficulty. Specifically, for *h*>4, the model can both minimize prompt bias *and* saturate ranking accuracy for all loss functions, causing the performance differences to disappear.
>
> **3. Regarding the rigor of 70B-scale experiments:** Our current study already spans ≈651 GPU-days to ensure fair, reproducible comparisons via extensive hyperparameter sweeps. Conducting this same rigorous process at the 70B scale would be not only computationally prohibitive but also methodologically challenging.
>
> To directly address your concern, we have begun a 70B-scale experiment (Qwen-2.5-72B) comparing one pairwise method (DPO) and one pointwise method (NCA) on the math task, following our rigorous tuning protocol (β sweep, shared LR grid). We plan to share the results during the discussion period.
>
> **W2/Q2: Formalizing the bias interaction mechanism**
>
> We provide a concrete formalization sufficient for the claims in this paper, built on three pillars:
>
> 1. **Definition of bias:** In Section 6, we formalize the prompt-specific bias as the marginalized score $\mathbb{E}\_{y}[r_\theta(x,y)]$ , derived directly from the scalar score function $r_\theta(x,y)$ used by DAAs.
> 2. **The objective's impact:** We then demonstrate how different objective classes structurally interact with this term.
>     - A **pointwise** objective, by pushing probabilities of chosen responses towards 1 and rejected responses towards 0, necessarily forces the marginalized score $\mathbb{E}_{y}[r\_\theta(x,y)]$ towards zero. This is a direct consequence of its functional form, which consumes model capacity to "unlearn" the bias.
>     - A **pairwise** objective, which depends only on the difference $r\_{\theta}(x,y_w) - r\_{\theta}(x,y_l)$, is structurally invariant to any constant shift in the scores for a given prompt $x$. Thus, it is not incentivized to alter the marginalized score and can dedicate model capacity purely to ranking.
> 3. **Empirical validation:** This formal distinction leads to a testable hypothesis regarding model capacity, which we validate.
>     - The **controlled experiment in Appendix F** serves as a direct empirical test of this mechanism, demonstrating precisely how the performance gap emerges and vanishes as a function of model capacity and data bias.
>     - **ICC analysis on real data** (Figure 7), where pairwise methods retain higher prompt-specific variance precisely in the medium-difficulty regime where they show an advantage.
>
> Together, these provide a clear, testable account of the mechanism. We believe that the present formalization and its empirical tests already support our conclusions.

---

> > ### Author Response · Authors · 2025-11-12
> >
> > **Q3: When might preserving prompt-specific biases be advantageous?**
> >
> > Our results suggest that attempting to *remove* prompt-specific bias (as pointwise methods do) can be detrimental when model capacity is limited relative to the task's difficulty. The process of "unlearning" the bias consumes finite model resources that could otherwise be used to learn the primary ranking task.
> >
> > Therefore, preserving the bias is advantageous in precisely these capacity-constrained scenarios. By being invariant to the bias, pairwise methods can dedicate the model's full capacity to what matters most: learning the relative preferences between outputs. This leads to better overall alignment quality.
> >
> > Our remark that "whether bias removal is beneficial remains an open question" is a statement of academic caution. While our experiments consistently show that needless bias removal is inefficient, we cannot rule out hypothetical scenarios (e.g., tasks with extreme biases) where its removal might be necessary. However, our work strongly suggests that for typical alignment tasks, preserving the bias is a feature of a more efficient learning objective.
> >
> > **Q4: Extending the framework to online optimization**
> >
> > Our work was intentionally focused on the **offline, SFT-based alignment setting**. The primary motivation was to create a controlled environment to disentangle the conflicting claims of superiority that have emerged from the fragmented literature on offline DAAs.
> >
> > We recognize the importance of online PO and agree it is a valuable direction for future work. However, extending our comprehensive framework to the online setting would require even greater GPU resources and time, constituting a separate line of research, especially as there are already works analyzing different aspects of the online setting, such as the one we cited [1]. We will also add a reference to the work on Online DPO [2] in our Future Work section. That paper also shows, in an online setting, that an added SFT term is detrimental compared to standard DPO. This reinforces that many aspects examined in our work are important to investigate for the online PO setup in the future.
> >
> > **References**
> >
> > [1] Calandriello D. et al. Human alignment of large language models through online preference optimisation
> >
> > [2] Zhang et. al. Online-DPO-R1: Unlocking Effective Reasoning Without the PPO Overhead. URL: [https://efficient-unicorn-451.notion.site/Online-DPO-R1-Unlocking-Effective-Reasoning-Without-the-PPO-Overhead-1908b9a70e7b80c3bc83f4cf04b2f175](https://www.notion.so/1908b9a70e7b80c3bc83f4cf04b2f175?pvs=21)
> >
> > We appreciate your careful reading. If there are remaining concerns that would prevent a higher score, we’d be grateful to address them during the discussion.

---

> > > ### Author Response · Authors · 2025-11-20
> > >
> > > **Additional follow-up for W1/Q1: 72B-scale validation**
> > >
> > > As promised, we completed a **Qwen-2.5-72B** experiment using *exactly* the same protocol as in the paper (Appendix B), mirroring the 14B setup.
> > >
> > > - **Learning rates:** we reused the best LR found for Qwen-7B (5e-7 for DPO, 1e-6 for NCA), following Table 11.
> > > - **β sweeps:**
> > >     - DPO: $\beta$ ∈ {0.001, 0.005, 0.01, 0.05, 0.1, 0.2}, best = 0.1
> > >     - NCA: $\beta$ ∈ {0.0005, 0.001, 0.005, 0.01, 0.05, 0.1}, best = 0.005
> > > - **Compute:** each run used **64 GPUs (ZeRO-2 + TP=8)** and ~8 hours (256 GPU-days in total).
> > >
> > >     Because of the cost, we evaluated one representative **pairwise** (DPO) and one **pointwise** (NCA) method, the exact contrast discussed in our response.
> > >
> > >     ### **72B results (Mean Avg@K with std, 4 seeds)**
> > >
> > >     | method | Mean Avg@K | GSM8K | MATH500 | AMC23 | Minerva | AIME24 | AIME25 |
> > >     | --- | --- | --- | --- | --- | --- | --- | --- |
> > >     | Base | 0.1813 (0.0044) | 0.3370 (0.0037) | 0.3164 (0.0049) | 0.2150 (0.0234) | 0.1532 (0.0068) | 0.0427 (0.0076) | 0.0233 (0.0058) |
> > >     | SFT | 0.3056 (0.0036) | 0.8130 (0.0025) | 0.5206 (0.0049) | 0.2794 (0.0187) | 0.1870 (0.0083) | 0.0250 (0.0049) | 0.0085 (0.0041) |
> > >     | DPO (pairwise) | **0.3896 (0.0047)** | 0.8847 (0.0020) | **0.6448 (0.0035)** | **0.4125 (0.0198)** | **0.3155 (0.0061)** | **0.0552 (0.0075)** | **0.0250 (0.0029)** |
> > >     | NCA (pointwise) | 0.3805 (0.0035) | **0.8964 (0.0030)** | 0.6320 (0.0066) | 0.3812 (0.0235) | 0.3119 (0.0031) | 0.0421 (0.0073) | 0.0196 (0.0036) |
> > >
> > >     These results confirm the same trend we report at smaller scales: DPO (pairwise) remains consistently ahead of NCA (pointwise) across most math benchmarks, although the absolute gap becomes narrower compared to the 14B scale. This matches exactly what your question anticipated: increased capacity reduces the magnitude of differences but does not eliminate them (at least within this setup). Importantly, we see no indication that scaling flips the relationship in favor of pointwise objectives, pairwise achieves higher peak performance even at 72B. These large-scale results therefore directly support our main claim. Thank you for the question, which allowed us to explore the even larger-scale setting; we hope this fully resolves your concern and look forward to a potential score increase.

---

> > > > ### Comment · Reviewer_MA69 · 2025-11-26
> > > >
> > > > Thank you for your response. It has resolved some of my concerns, and I will maintain my rating.

---

> > > > > ### Author Response · Authors · 2025-11-26
> > > > >
> > > > > Thank you again for your thoughtful review. We also really appreciate that you took the time to read our extended rebuttal and the additional 72B-scale experiment.
> > > > >
> > > > > Since the discussion period is still ongoing and we would like to further improve the paper, could you please share which of your original concerns you now consider resolved, and which points (if any) you still see as open or only partially addressed? This would help us focus our remaining efforts on the aspects that matter most for you.
> > > > >
> > > > > If our clarifications and the new 72B results have increased your confidence in the robustness or generality of our findings, we would also be grateful if you could reflect that in the confidence score.
> > > > >
> > > > > Thank you once again for your time and for engaging with our work.

---

> > > > > > ### Author Response · Authors · 2025-12-02
> > > > > >
> > > > > > To specifically address your desire for a formal mechanism, in the revised version **we have added a comprehensive new appendix:** **Appendix G: "Theoretical Analysis of Prompt-Specific Bias and Ranking Objectives”,** that makes the mechanism behind “ranking objective vs. prompt bias” explicit in the same offline, static binary-preference settings we study. The structure is as follows:
> > > > > >
> > > > > > **1. General scalar-score family and prompt bias.**
> > > > > >
> > > > > > We work with a broad class of scalar scores
> > > > > >
> > > > > > $$
> > > > > > r_\theta(x,y) = F(\pi_\theta(y\mid x), \mathcal C(x,y)),
> > > > > > $$
> > > > > >
> > > > > > where $\mathcal C(x,y)$ is fixed (e.g., reference model probabilities, length) and only $\pi\_\theta$ depends on $\theta$. This covers the scalar scores used in our main experiments ($r^{\text{ref}}\_\theta$*, $r^{\text{odds}}_\theta$*) and also the AlphaPO-style $r\_\alpha$ parameterization that we additionally evaluate in the Qwen–Math appendix. For each prompt $x$, we define the **marginalized score / prompt bias**
> > > > > >
> > > > > > $$
> > > > > > b\_\theta(x) := \mathbb{E}\_{y\sim q_x}[r\_\theta(x,y)],
> > > > > > $$
> > > > > >
> > > > > > where $q_x$ is the empirical distribution of candidates for that prompt.
> > > > > >
> > > > > > **2. Bias is generically learnable.**
> > > > > >
> > > > > > We compute $\partial b_\theta(x)/\partial z_m$, the gradient of the marginalized score w.r.t. a logit $z_m$ for candidate $y_m$, and show that it vanishes only under a very special compatibility condition between $q_x$ and $\pi_\theta(\cdot\mid x)$. Outside this measure-zero set, **the prompt bias $b_\theta(x)$ is a learnable functional of the parameters**: the model *can* change it.
> > > > > >
> > > > > > **3. Total score gradient as “bias signal”.**
> > > > > >
> > > > > > For a general preference loss
> > > > > >
> > > > > > $$
> > > > > > L(\theta) = \mathbb{E}{(x,y_w,y_l)}[\ell(r_w,r_l)],
> > > > > > $$
> > > > > >
> > > > > > *we define $g_\theta(x,y) := \frac{\partial L}{\partial r_\theta(x,y)}$* and the **total score gradient for prompt $x$**
> > > > > >
> > > > > > $$
> > > > > > G_\theta(x) := \sum_{y\in\mathcal Y_x} g_\theta(x,y).
> > > > > > $$
> > > > > >
> > > > > > We prove that $G_\theta(x)$ is exactly the derivative of $L$ w.r.t. a hypothetical uniform shift $r_\theta(x,y)\mapsto r_\theta(x,y)+\varepsilon$ for all responses to that prompt. Since such a shift changes $b_\theta(x)$ by $\varepsilon$, $G_\theta(x)$ **measures the objective’s first-order incentive to modify the prompt bias**.
> > > > > >
> > > > > > **4. Pairwise objectives: $G_\theta(x)=0$ (invariant to bias).**
> > > > > >
> > > > > > For any pairwise objective of the form
> > > > > >
> > > > > > $$
> > > > > > \ell_{\text{pair}}(r_w,r_l) = \phi(r_w - r_l),
> > > > > > $$
> > > > > >
> > > > > > we show that contributions from each preference pair cancel at the prompt level: winners contribute $+\phi'(\Delta)$, losers $-\phi'(\Delta)$, and summing over all candidates yields $G_\theta(x)=0$ **for every prompt**. Thus, pairwise losses are structurally invariant to uniform per-prompt shifts and **do not generate a direct gradient signal to change $b_\theta(x)$**.
> > > > > >
> > > > > > **5. Pointwise objectives: $G_\theta(x)\neq 0$ generically (coupled to bias).**
> > > > > >
> > > > > > For pointwise objectives
> > > > > >
> > > > > > $$
> > > > > > \ell_{\text{point}}(r_w,r_l) = \psi_+(r_w) + \psi_-(r_l),
> > > > > > $$
> > > > > >
> > > > > > we obtain
> > > > > >
> > > > > > $$
> > > > > > G_\theta(x) = \sum_i \big[\psi_+'(r_{w,i}) + \psi_-'(r_{l,i})\big],
> > > > > > $$
> > > > > >
> > > > > > where $i$ ranges over preference pairs for that prompt. This sum is **not constrained to be zero by the loss structure**; requiring it to vanish imposes non-generic relations between $r_w$ and $r_l$ across all pairs. For typical score configurations we therefore have $G_\theta(x)\neq 0$, so pointwise losses explicitly act on the prompt bias $b_\theta(x)$, not just on relative rankings. We spell this out concretely for ASFT in the appendix.
> > > > > >
> > > > > >
> > > > > > Putting this together, Appendix G shows that:
> > > > > >
> > > > > > - **(i)** prompt-specific bias is well defined for a broad scalar-score family and is generically adjustable by the model;
> > > > > > - **(ii)** pairwise objectives have zero first-order sensitivity to uniform per-prompt shifts $G_\theta(x)=0$, so the loss itself does not “ask” the model to change $b_\theta(x)$;
> > > > > > - **(iii)** pointwise objectives generically induce a nonzero $G_\theta(x)$, structurally coupling them to prompt-bias modification.
> > > > > >
> > > > > > This theoretical picture matches both our controlled experiment (where pointwise methods underperform in the intermediate capacity regime when they spend capacity on bias removal, Appendix F) and our ICC analysis on real data (Figure 7), where pairwise methods tend to preserve more prompt-level variance while pointwise methods reduce it, in line with the mechanism above.

---

### Author Response · Authors · 2025-11-19

We thank all reviewers for their time and thoughtful feedback. We have provided detailed answers to your specific questions in the individual threads below. In the revised paper, we have incorporated the following changes to reinforce our core contributions:

1. **Expanded and Reorganized Related Work:** We have moved this section to precede the Conclusion and enriched the discussion to include the suggested references. This better positions our contribution as a fundamental unification *across* algorithmic families (pairwise vs. pointwise), providing a baseline that is complementary to fine-grained improvements *within* the pairwise paradigm.
2. **Refined Motivation (Section 3.1.1):** We have sharpened the wording to rigorously distinguish between the **SFT loss term** and the explicit **SFT training stage**.
3. **Explicit Scope Definition:** We have clarified that our analysis focuses on **offline alignment with static binary preferences**. This isolates the ranking objective as the primary performance driver, orthogonal to confounding factors found in online or complex feedback settings.

We believe these updates solidify the paper’s core claim and improve the clarity of our unified framework.

---

### Meta-Review · Area_Chair_NrrT · 2026-01-06

**Summary:**

This work presents a comparison and analysis of Direct Alignment Algorithms - a class of methods for aligning LLMs without an explicit reward model.  To address this, they propose a unified training protocol for DAAs that makes it possible to compare different methods fairly. The authors conduct experiments across various models (like Llama or Qwen) and sizes (3-14B). They provide evidence that observed performance gaps arise from the interactions between each objective and prompt-specific data biases.

**Reviewer Concerns:**

1. The main concern lies in the novelty of the work. In addition, it has not been compared with more recent work and shows the difference. More clarification about the contribution would be helpful.
2. Limited Generalization Across Domains and Model Scales Although the paper reports results across several tasks and models, it remains unclear how well the method can generalize to larger models and more domains.

**Reviewer Scores:**

The paper received mixed ratings initially. The reviewers think the paper is well-written, and the empirical results on the current settings are good. After the rebuttal, the two main concerns still hold: the novelty of the work and the generalization ability of the method. The reviewers and the authors have not reached an agreement. Based on the current status, AC made the decision of rejection.

---

### Decision · Program_Chairs · 2026-01-26

Reject